# Fair Class-Incremental Learning using Sample Weighting

## Abstract

Model fairness is becoming important in class-incremental learning for Trustworthy AI. While accuracy has been a central focus in class-incremental learning, fairness has been relatively understudied. However, naïvely using all the samples of the current task for training results in *unfair catastrophic forgetting* for certain sensitive groups including classes. We theoretically analyze that forgetting occurs if the average gradient vector of the current task data is in an "opposite direction" compared to the average gradient vector of a sensitive group, which means their inner products are negative. We then propose a *fair class-incremental learning* framework that adjusts the training weights of current task samples to change the direction of the average gradient vector and thus reduce the forgetting of underperforming groups and achieve fairness. For various group fairness measures, we formulate optimization problems to minimize the overall losses of sensitive groups while minimizing the disparities among them. We also show the problems can be solved with linear programming and propose an efficient Fairness-aware Sample Weighting (FSW) algorithm. Experiments show that FSW achieves better accuracy-fairness tradeoff results than state-of-the-art approaches on real datasets.

## 1. Introduction

Trustworthy AI is becoming critical in various continual learning applications including autonomous vehicles, personalized recommendations, healthcare monitoring, and more (Liu et al., 2021; Kaur et al., 2023). In particular, it is important to improve model fairness along with accuracy when developing models incrementally in dynamic environments. Unfair model predictions have the potential to undermine the trust and safety in human-related automated

systems, especially as observed frequently in the context of continual learning. There are largely three continual learning scenarios (van de Ven & Tolias, 2019): task-incremental, domain-incremental, and class-incremental learning where the task, domain, or class may change over time, respectively. In this paper, we focus on class-incremental learning, where the objective is to incrementally learn new classes as they appear.

The main challenge of class-incremental learning is to learn new classes of data, while not forgetting previously-learned classes (Belouadah et al., 2021; Lange et al., 2022). If we simply fine-tune the model on the new classes, the model will gradually forget about the previously-learned classes. This phenomenon called catastrophic forgetting (McCloskey & Cohen, 1989; Kirkpatrick et al., 2016) may easily occur in real-world scenarios where the model needs to continuously learn new classes. We cannot stop learning new classes to avoid this forgetting either. Instead, we need to have a balance between learning new information and retaining previously-learned knowledge, which is called the stability-plasticity dilemma (Abraham & Robins, 2005; Mermillod et al., 2013; Kim & Han, 2023).

In this paper, we solve the problem of *fair class-incremental learning* where the goal is to satisfy various notions of fairness among sensitive groups including classes in addition to classifying accurately. In some scenarios, the class itself can be considered a sensitive attribute, especially in classification tasks where a model produces biased predictions toward a specific group of classes (Truong et al., 2023). In continual learning, unfair forgetting may occur if the current task data has similar characteristics to previous data, but belongs to different sensitive groups including classes, which negatively affects the performance on the previous data during training. Despite the importance of the problem, the existing research (Chowdhury & Chaturvedi, 2023; Truong et al., 2023) is still nascent and has limitations in terms of technique or scope (see Sec. 2). In comparison, we support fairness more generally in class-incremental learning by satisfying various notions of group fairness for sensitive groups including classes.

We demonstrate how unfair forgetting can occur on a synthetic dataset with two attributes $(x_1, x_2)$, and one true label $y$ as shown in Fig. 1a. We sample data for each class

[1]Anonymous Institution, Anonymous City, Anonymous Region, Anonymous Country. Correspondence to: Anonymous Author <anon.email@domain.com>.

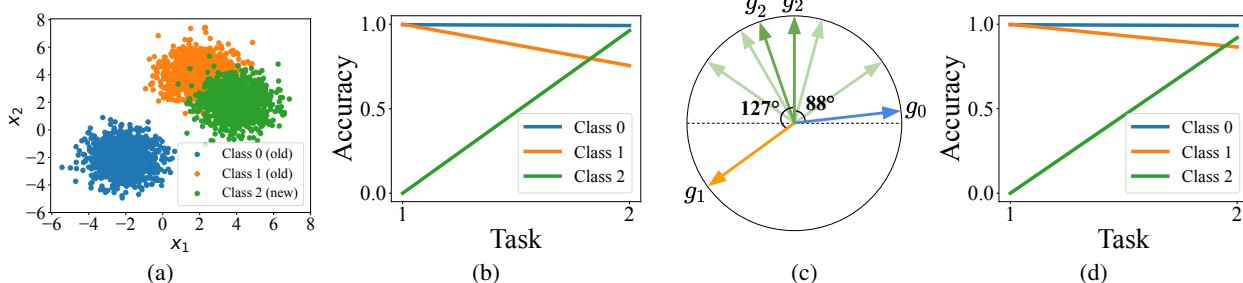

Figure 1: (a) A synthetic dataset for class-incremental learning. (b) After training on Classes 0 and 1, training on Class 2 results in unfair forgetting for Class 1 only. (c) The reason is that the average gradient vector of Class 2, $g_2$, is more than $90°$ apart from Class 1's $g_1$, which means the model is being trained in an opposite direction. Our method adjusts $g_2$ to $g_2^*$ through sample weighting to be closer to $g_1$, but not too far from the original $g_2$. (d) As a result, the unfair forgetting is mitigated while minimally sacrificing accuracy for Class 2.

from three different normal distributions: $(x_1, x_2)|y = 0 \sim \mathcal{N}([-2; -2], [1; 1])$, $(x_1, x_2)|y = 1 \sim \mathcal{N}([2; 4], [1; 1])$, and $(x_1, x_2)|y = 2 \sim \mathcal{N}([4; 2], [1; 1])$. Note that each data distribution can also be defined as a sensitive group with a sensitive attribute $z$. To simulate class-incremental learning, we introduce data for Class 0 (blue) and Class 1 (orange) in Task 1, followed by Class 2 (green) data in Task 2, where Class 2's data is similar to Class 1's data. We observe that this setting frequently occurs in real datasets, where different classes of data exhibit similar features or characteristics, as shown in Sec. B.1. We assume a data replay setting where only a small amount of previous data from Classes 0 and 1 are stored and utilized together when training on Class 2 data. After training the model for Task 1, we observe how the model accuracies on the three classes change when training for Task 2 in Fig. 1b. As the accuracy on Class 2 improves, there is a catastrophic forgetting of Class 1 only, which leads to unfairness.

To analytically understand the unfair forgetting, we project the average gradient vector for each class data on a 2-dimensional space in Fig. 1c. Here $g_0$, $g_1$, and $g_2$ represent the average gradient vectors of the samples of Classes 0, 1, and 2, respectively. We observe that $g_2$ is $127°$ apart from $g_1$, but $88°$ from $g_0$, which means that the inner products $\langle g_2, g_1 \rangle$ and $\langle g_2, g_0 \rangle$ are negative and close to 0, respectively. In Sec. 3.1, we theoretically show that a negative inner product between average gradient vectors of current and previous data results in higher loss for the previous data as the model is being updated in an opposite direction and identify a sufficient condition for unfair forgetting. As a result, Class 1's accuracy decreases, while Class 0's accuracy remains stable.

Our solution to mitigate unfair forgetting is to adjust the average gradient vector of the current task data by weighting its samples. The light-green vectors in Fig. 1c are the gradient vectors of individual samples from Class 2, and by weighting them we can adjust $g_2$ to $g_2^*$ to make the inner product

with $g_1$ less negative. At the same time, we do not want $g_2^*$ to be too different from $g_2$ and lose accuracy. In Sec. 3.2, we formalize this idea using the weighted average gradient vector of the current task data. We then optimize the sample weights such that unfair forgetting and accuracy reduction over sensitive groups including classes are both minimized. We show this optimization can be solved with linear programming and propose our efficient Fairness-aware Sample Weighting (FSW) algorithm. Fig. 1d shows how using FSW mitigates the unfair forgetting between Classes 0 and 1 without harming Class 2's accuracy much. Our framework supports the group fairness measures equal error rate (Venkatasubramanian, 2019), equalized odds (Hardt et al., 2016), and demographic parity (Feldman et al., 2015) and can be potentially extended to other measures.

In our experiments, we show that FSW achieves better fairness and competitive accuracy compared to state-of-the-art baselines on various image, text, and tabular datasets. The benefits come from assigning different training weights to the current task samples with accuracy and fairness in mind.

**Summary of Contributions:** (1) We theoretically analyze how unfair catastrophic forgetting can occur in class-incremental learning; (2) We formulate optimization problems for mitigating the unfairness for various group fairness measures and propose an efficient fairness-aware sample weighting algorithm, FSW; (3) We demonstrate how FSW outperforms state-of-the-art baselines in terms of fairness with comparable accuracy on various datasets.

## 2. Related Work

Class-incremental learning is a challenging type of continual learning where a model continuously learns new tasks, each composed of new disjoint classes, and the goal is to minimize catastrophic forgetting (Mai et al., 2022; Masana et al., 2023). Data replay techniques (Lopez-Paz & Ranzato, 2017; Chaudhry et al., 2019b) store a small portion of previous data in a buffer to utilize for training and is widely

used with other techniques including knowledge distillation, model rectification, and dynamic networks (see more details in Sec. C). Simple buffer sample selection methods such as random or herding-based approaches (Rebuffi et al., 2017) are also commonly used as well. There are also more advanced gradient-based sample selection techniques like GSS (Aljundi et al., 2019) and OCS (Yoon et al., 2022) that manage buffer data to have samples with diverse and representative gradient vectors. All these works do not consider fairness and simply assume that the entire incoming data is used for model training, which may result in unfair forgetting as we show in our experiments.

Model fairness research mitigates bias by ensuring that a model's performance is equitable across different sensitive groups, thereby preventing discrimination based on race, gender, age, or other sensitive attributes (Mehrabi et al., 2022). Existing model fairness techniques can be categorized as pre-processing, in-processing, and post-processing (see more details in Sec. C). In addition, there are other techniques that assign adaptive weights for samples to improve fairness (Chai & Wang, 2022; Jung et al., 2023). However, most of these techniques assume that the training data is given all at once, which may not be realistic. There are techniques for fairness-aware active learning (Anahideh et al., 2022; Pang et al., 2024; Tae et al., 2024), in which the training data evolves with the acquisition of samples. However, these techniques store all labeled data and use them for training, which is impractical in continual learning settings.

A recent study addresses model fairness in class-incremental learning where there is a risk of disproportionally forgetting previously-learned sensitive groups including classes, leading to unfairness across different groups. A recent study (He, 2024) addresses the dual imbalance problem involving both inter-task and intra-task imbalance by reweighting gradients. However, the bias is not only caused by the data imbalance, but also by the inherent or acquired characteristics of data itself (Mehrabi et al., 2022; Angwin et al., 2022). CLAD (Xu et al., 2024) first discovers imbalanced forgetting between classes caused by conflicts in representation and proposes a class-aware disentanglement technique to improve accuracy. Among the fairness-aware techniques, FaIRL (Chowdhury & Chaturvedi, 2023) supports group fairness metrics like demographic parity for continual learning, but proposes a representation learning method that does not directly optimize the given fairness measure and thus has limitations in improving fairness as we show in experiments. FairCL (Truong et al., 2023) also addresses fairness in a continual learning setup, but only focuses on resolving the imbalanced class distribution based on the number of pixels of each class in an image for semantic segmentation tasks. In comparison, we support fairness more generally in class-incremental learning by satisfying multiple notions of group fairness for sensitive groups including classes.

## 3. Framework

In this section, we first theoretically analyze unfair forgetting using gradient vectors of sensitive groups and the current task data. Next, we propose sample weighting to mitigate unfairness by adjusting the average gradient vector of the current task data and provide an efficient algorithm.

**Notations** In class-incremental learning, a model incrementally learns new current task data along with previous buffer data using data replay. Suppose we train a model to incrementally learn $L$ tasks $\{T_1, T_2, \ldots, T_L\}$ over time, and there are $N$ classes in each task as $C^{T_l} = \{C_1^{T_l}, C_2^{T_l}, \ldots, C_N^{T_l}\}$ with no overlapping classes between different tasks (i.e., $C^{T_{l_1}} \cap C^{T_{l_2}} = \emptyset$ if $l_1 \neq l_2$). After learning the $l^{th}$ task $T_l$, we would like the model to remember all $(l-1) \cdot N$ previous task classes and an additional $N$ current task classes. We assume the buffer has a fixed size of $M$ samples. For $L$ tasks, we allocate $m = M/L$ samples of buffer data per task. If each task consists of $N$ classes, then we allocate $m/N = M/(L \cdot N)$ samples of buffer data per class (Chaudhry et al., 2019a; Mirzadeh et al., 2020; Chaudhry et al., 2021). Each task $T_l = \{d_i = (X_i, y_i)\}_{i=1}^k$ is composed of feature-label pairs where a feature $X_i \in \mathbb{R}^d$ and a true label $y_i \in \mathbb{R}^c$. We also use $\mathcal{M}_l = \{d_j = (X_j, y_j)\}_{j=1}^m$ to represent the buffer data for each previous $l^{th}$ task $T_l$. We assume the buffer data per task is small, i.e., $m \ll k$ (Chaudhry et al., 2019b).

When defining fairness for class-incremental learning, we utilize sensitive groups including classes. According to the fairness literature, sensitive groups are divided by sensitive attributes like gender and race. For example, if the sensitive attribute is gender, the sensitive groups can be Male and Female. Similarly, the classes can also be perceived as sensitive groups, with the class itself serving as the sensitive attribute. Since we would like to support any sensitive group in a class-incremental setting, we use the following unifying notations: (1) if the sensitive groups are classes, then they form the set $G_y = \{(X, y) \in \mathcal{D} : y = y, y \in \mathbb{Y}\}$ where $\mathcal{D}$ is a dataset, y is a class attribute, and $\mathbb{Y}$ is the set of y; (2) if we are using sensitive attributes in addition to classes, we can further divide the classes into the set $G_{y,z} = \{(X, y, z) \in \mathcal{D} : y = y, z = z, y \in \mathbb{Y}, z \in \mathbb{Z}\}$ where z is a sensitive attribute, and $\mathbb{Z}$ is the set of z.

### 3.1. Unfair Forgetting

Catastrophic forgetting occurs when a model adapts to a new task and exhibits a drastic decrease in performance on previously-learned tasks (Parisi et al., 2019). We take inspiration from GEM (Lopez-Paz & Ranzato, 2017), which theoretically analyzes catastrophic forgetting by utilizing the angle between gradient vectors of data. If the inner products of gradient vectors for previous tasks and the current task are negative (i.e., $90° <$ angle $\leq 180°$), the loss of previous tasks increases after learning the current task. Catastrophic

forgetting thus occurs when the gradient vectors of different tasks point in opposite directions. Intuitively, the opposite gradient vectors update the model parameters in conflicting directions, leading to forgetting while learning.

Using the notion of catastrophic forgetting, we propose theoretical results for unfair forgetting:

**Lemma 3.1.** *Denote $G$ as a sensitive group of data composed of features $X$ and true labels $y$. Also, denote $f_\theta^{l-1}$ as a previous model and $f_\theta$ as the updated model after training on the current task $T_l$. Let $\ell$ be any differentiable loss function (e.g., cross-entropy loss), and $\eta$ be a learning rate. Then, the loss of the sensitive group of data after training with a current task sample $d_i \in T_l$ is approximated as:*

$$\tilde{\ell}(f_\theta, G) = \ell(f_\theta^{l-1}, G) - \eta \nabla_\theta \ell(f_\theta^{l-1}, G)^\top \nabla_\theta \ell(f_\theta^{l-1}, d_i), \tag{1}$$

*where $\tilde{\ell}(f_\theta, G)$ is the approximated average loss between model predictions $f_\theta(X)$ and true labels $y$, whereas $\ell(f_\theta^{l-1}, G)$ is the exact average loss, $\nabla_\theta \ell(f_\theta^{l-1}, G)$ is the average gradient vector for the samples in the group $G$, and $\nabla_\theta \ell(f_\theta^{l-1}, d_i)$ is the gradient vector for a sample $d_i$, each with respect to the previous model $f_\theta^{l-1}$.*

The proof is in Sec. A.1. We employ first-order Taylor series approximation for the proof, which is widely used in the continual learning literature, by assuming that the loss function is locally linear in small optimization steps and considering the first-order term as the cause of catastrophic forgetting (Lopez-Paz & Ranzato, 2017; Aljundi et al., 2019; Lee et al., 2019). We empirically find that the approximation error is large when a new task begins because new samples with unseen classes are introduced. However, the error gradually becomes quite small as the number of epochs increases while training a model for the task, as in Sec. B.2.

To define fairness in class-incremental learning with the approximated loss, we adopt the definition of approximate fairness that considers a model to be fair if it has approximately the same loss on the positive class, independent of the group membership (Donini et al., 2018). In this paper, we compute fairness measures based on the disparity between approximated cross-entropy losses, which are derived from Lemma 3.1 using gradients. The following proposition shows how using the cross-entropy loss disparity can effectively approximate common group fairness metrics such as equalized odds and demographic parity (see Sec. A.2 and Sec. B.15 for more justification of the loss function and an alternative, respectively).

**Proposition 3.2.** *(From Roh et al. (2021; 2023); Shen et al. (2022)) Using the cross-entropy loss disparity to measure fairness is empirically verified to provide reasonable proxies for common group fairness metrics.*

Using Lemma 3.1 and Proposition 3.2, the following theorem suggests a sufficient condition for unfair forgetting.

Intuitively, if a training sample's gradient is in an opposite direction to the average gradient of an underperforming group, but not for an overperforming group, the training causes more unfairness between the two groups.

**Theorem 3.3.** *Let $\ell$ be the cross-entropy loss and we denote $G_1$ and $G_2$ as the overperforming and underperforming sensitive groups of data, and $d_i$ as a training sample that satisfy the following conditions: $\ell(f_\theta^{l-1}, G_1) < \ell(f_\theta^{l-1}, G_2)$ while $\nabla_\theta \ell(f_\theta^{l-1}, G_1)^\top \nabla_\theta \ell(f_\theta^{l-1}, d_i) > 0$ and $\nabla_\theta \ell(f_\theta^{l-1}, G_2)^\top \nabla_\theta \ell(f_\theta^{l-1}, d_i) < 0$. Then $|\tilde{\ell}(f_\theta, G_1) - \tilde{\ell}(f_\theta, G_2)| > |\ell(f_\theta^{l-1}, G_1) - \ell(f_\theta^{l-1}, G_2)|$.*

The proof is in Sec. A.1. The result shows that the loss disparity between the two groups could become larger after training on the current task sample, which leads to worse fairness. This theorem can be extended to when we have a set of current task samples $T_l = \{d_i = (X_i, y_i)\}_{i=1}^k$ where we can replace $\nabla_\theta \ell(f_\theta^{l-1}, d_i)$ with $\frac{1}{|T_l|} \sum_{d_i \in T_l} \nabla_\theta \ell(f_\theta^{l-1}, d_i)$. If the average gradient vector of the current task data satisfies the derived sufficient condition, training with all of the current task samples using equal weights could thus result in unfair catastrophic forgetting.

### 3.2. Sample Weighting for Unfairness Mitigation

To mitigate unfairness, we propose sample weighting as a way to suppress samples that negatively impact fairness and promote samples that help. Finding the weights is not trivial as there can be many sensitive groups. Given training weights $\mathbf{w}_l \in [0, 1]^{|T_l|}$ for the samples in the current task data, the approximated loss of a group $G$ after training is:

$$\tilde{\ell}(f_\theta, G) = \ell(f_\theta^{l-1}, G) -$$
$$\eta \nabla_\theta \ell(f_\theta^{l-1}, G)^\top \left( \frac{1}{|T_l|} \sum_{d_i \in T_l} \mathbf{w}_l^i \nabla_\theta \ell(f_\theta^{l-1}, d_i) \right),$$

where $\mathbf{w}_l^i$ is a training weight for the current task sample $d_i$. We then formulate an optimization problem to find the weights such that both loss and unfairness are minimized. Here we define $\mathbb{Y}$ as the set of all classes and $\mathbb{Y}_c$ as the set of classes in the current task. We represent accuracy as the average loss over the current task data and minimize the cost function $L_{acc} = \tilde{\ell}(f_\theta, G_{\mathbb{Y}_c}) = \frac{1}{|\mathbb{Y}_c|} \sum_{y \in \mathbb{Y}_c} \tilde{\ell}(f_\theta, G_y) = \frac{1}{|\mathbb{Y}_c||\mathbb{Z}|} \sum_{y \in \mathbb{Y}_c, z \in \mathbb{Z}} \tilde{\ell}(f_\theta, G_{y,z})$. For fairness, the cost function $L_{fair}$ depends on the group fairness measure as we explain below. We then minimize $L_{fair} + \lambda L_{acc}$ where $\lambda$ is a hyperparameter that balances fairness and accuracy.

**Equal Error Rate (EER)** This measure (Venkatasubramanian, 2019) is defined as $\Pr(\hat{y} \neq y_1 | y = y_1) = \Pr(\hat{y} \neq y_2 | y = y_2)$ for $y_{1,2} \in \mathbb{Y}$, where $\hat{y}$ is the predicted class, and $y$ is the true class. We define the cost function for EER as the average absolute deviation of the class loss, consistent with the definition of group fairness metrics: $L_{EER} = \frac{1}{|\mathbb{Y}|} \sum_{y \in \mathbb{Y}} |\tilde{\ell}(f_\theta, G_y) - \tilde{\ell}(f_\theta, G_{\mathbb{Y}})|$. The entire optimization problem is:

$$\min_{\mathbf{w}_l} \frac{1}{|\mathbb{Y}|} \sum_{y \in \mathbb{Y}} |\tilde{\ell}(f_\theta, G_y) - \tilde{\ell}(f_\theta, G_{\mathbb{Y}})| +$$

$$\lambda \frac{1}{|\mathbb{Y}_c|} \sum_{y \in \mathbb{Y}_c} \tilde{\ell}(f_\theta, G_y), \tag{2}$$

$$\text{where } \tilde{\ell}(f_\theta, G_y) = \ell(f_\theta^{l-1}, G_y) -$$

$$\eta \nabla_\theta \ell(f_\theta^{l-1}, G_y)^\top \left( \frac{1}{|T_l|} \sum_{d_i \in T_l} \mathbf{w}_l^i \nabla_\theta \ell(f_\theta^{l-1}, d_i) \right)$$

**Equalized Odds (EO)** This measure (Hardt et al., 2016) is satisfied when sensitive groups have the same accuracy, i.e., $\Pr(\hat{y} = y | y = y, z = z_1) = \Pr(\hat{y} = y | y = y, z = z_2)$ for $y \in \mathbb{Y}$ and $z_{1,2} \in \mathbb{Z}$. We design the cost function for EO as $L_{EO} = \frac{1}{|\mathbb{Y}||\mathbb{Z}|} \sum_{y \in \mathbb{Y}, z \in \mathbb{Z}} |\tilde{\ell}(f_\theta, G_{y,z}) - \tilde{\ell}(f_\theta, G_y)|$, and the entire optimization problem is:

$$\min_{\mathbf{w}_l} \frac{1}{|\mathbb{Y}||\mathbb{Z}|} \sum_{y \in \mathbb{Y}, z \in \mathbb{Z}} |\tilde{\ell}(f_\theta, G_{y,z}) - \tilde{\ell}(f_\theta, G_y)| +$$

$$\lambda \frac{1}{|\mathbb{Y}_c||\mathbb{Z}|} \sum_{y \in \mathbb{Y}_c, z \in \mathbb{Z}} \tilde{\ell}(f_\theta, G_{y,z}), \tag{3}$$

$$\text{where } \tilde{\ell}(f_\theta, G_{y,z}) = \ell(f_\theta^{l-1}, G_{y,z}) -$$

$$\eta \nabla_\theta \ell(f_\theta^{l-1}, G_{y,z})^\top \left( \frac{1}{|T_l|} \sum_{d_i \in T_l} \mathbf{w}_l^i \nabla_\theta \ell(f_\theta^{l-1}, d_i) \right)$$

**Demographic Parity (DP)** This measure (Feldman et al., 2015) is satisfied by minimizing the difference in positive prediction rates between sensitive groups. Here, we extend the notion of demographic parity to the multi-class setting (Alabdulmohsin et al., 2022; Denis et al., 2023), i.e., $\Pr(\hat{y} = y | z = z_1) = \Pr(\hat{y} = y | z = z_2)$ for $y \in \mathbb{Y}$ and $z_{1,2} \in \mathbb{Z}$. In the binary setting of $\mathbb{Y} = \mathbb{Z} = \{0, 1\}$, a sufficient condition for demographic parity is suggested using the loss multiplied by the ratios of sensitive groups (Roh et al., 2021). By extending the setting to multi-class, we derive a sufficient condition for demographic parity as follows: $\frac{m_{y,z_1}}{m_{*,z_1}} \ell(f_\theta, G_{y,z_1}) = \frac{m_{y,z_2}}{m_{*,z_2}} \ell(f_\theta, G_{y,z_2})$ where $m_{y,z} := |\{i : \mathbf{y}_i = y, \mathbf{z}_i = z\}|$ and $m_{*,z} := |\{i : \mathbf{z}_i = z\}|$. The proof is in Sec. A.3. Let us define $\ell'(f_\theta, G_{y,z}) = \frac{m_{y,z}}{m_{*,z}} \ell(f_\theta, G_{y,z})$ and $\ell'(f_\theta, G_y) = \frac{1}{|\mathbb{Z}|} \sum_{n=1}^{|\mathbb{Z}|} \frac{m_{y,z_n}}{m_{*,z_n}} \ell(f_\theta, G_{y,z_n})$. We then define the cost function for DP using the sufficient condition as $L_{DP} = \frac{1}{|\mathbb{Y}||\mathbb{Z}|} \sum_{y \in \mathbb{Y}, z \in \mathbb{Z}} |\tilde{\ell}'(f_\theta, G_{y,z}) - \tilde{\ell}'(f_\theta, G_y)|$. The entire optimization problem is:

$$\min_{\mathbf{w}_l} \frac{1}{|\mathbb{Y}||\mathbb{Z}|} \sum_{y \in \mathbb{Y}, z \in \mathbb{Z}} |\tilde{\ell}'(f_\theta, G_{y,z}) - \tilde{\ell}'(f_\theta, G_y)| +$$

$$\lambda \frac{1}{|\mathbb{Y}_c||\mathbb{Z}|} \sum_{y \in \mathbb{Y}_c, z \in \mathbb{Z}} \tilde{\ell}(f_\theta, G_{y,z}), \tag{4}$$

$$\text{where } \tilde{\ell}(f_\theta, G_{y,z}) = \ell(f_\theta^{l-1}, G_{y,z}) -$$

$$\eta \nabla_\theta \ell(f_\theta^{l-1}, G_{y,z})^\top \left( \frac{1}{|T_l|} \sum_{d_i \in T_l} \mathbf{w}_l^i \nabla_\theta \ell(f_\theta^{l-1}, d_i) \right)$$

---

**Algorithm 1** Fair Class-Incremental Learning

**Input:** Current task data $T_l$, previous buffer data $\mathcal{M} = \{\mathcal{M}_1, \ldots, \mathcal{M}_{l-1}\}$, previous model $f_\theta^{l-1}$, loss function $\ell$, learning rate $\eta$, hyperparameters $\{\alpha, \lambda, \tau\}$, and fairness measure $F$

1: **for** each epoch **do**
2:    $\mathbf{w}_l^* = \textbf{FSW}(T_l, \mathcal{M}, f_\theta^{l-1}, \ell, \alpha, \lambda, F)$
3:    $g_{curr} = \frac{1}{|T_l|} \sum_{d_i \in T_l} \mathbf{w}_l^{*i} \nabla_\theta \ell(f_\theta^{l-1}, d_i)$
4:    $g_{prev} = \nabla_\theta \ell(f_\theta^{l-1}, \mathcal{M})$
5:    $\theta \leftarrow \theta - \eta(g_{curr} + \tau g_{prev})$
6: **end for**
7: $\mathcal{M}_l = \textit{Buffer Sample Selection}(T_l)$
8: $\mathcal{M} \leftarrow \mathcal{M} \cup \mathcal{M}_l$

---

To find the optimal sample weights for the current task data considering both model accuracy and fairness, we first transform the defined optimization problems of Eq. 2, 3, and 4 into the form of linear programming (LP) problems.

**Theorem 3.4.** *The fairness-aware optimization problems (Eq. 2, 3, and 4) can be transformed into the form of linear programming (LP) problems.*

The loss of each group can be approximated as a linear function, as described in Lemma 3.1. This result implies that the optimization problems, consisting of the loss of each group, can likewise be transformed into LP problems. The comprehensive proof is in Sec. A.4. We solve the LP problems using linear optimization solvers (e.g., CPLEX (Cplex, 2009)). As we add the average loss of the current task data in Eq. 2, 3, and 4 as a regularization term, the optimal sample weights do not indicate a severely shifted distribution.

### 3.3. Algorithm

We describe the overall process of fair class-incremental learning in Alg. 1. For the recently arrived current task data, we first perform our fairness-aware sample weighting (FSW) to assign training weights that can help learn new knowledge of the current task while retaining accurate and fair memories of previous tasks (Step 2). Next, we train the model using the current task data with its corresponding weights and stored buffer data of previous tasks (Steps 3–5), where $\eta$ is a learning rate, and $\tau$ is a hyperparameter to balance between them during training. The sample weights are computed once at the beginning of each epoch, and they are applied to all batches for computational efficiency (Killamsetty et al., 2021a;b). This procedure is repeated until the model converges (Steps 1–5). Before moving on to the next task, we employ buffer sample selection to store a small data subset for the current task (Steps 7–8). Buffer sample selection can also be done with consideration for fairness, but our experimental observations indicate that selecting representative and diverse samples for the buffer, as previous studies have shown, results in better accuracy and also fairness performance. We thus employ a simple random sampling technique for the buffer sample selection in our framework.

**Algorithm 2** Fairness-aware Sample Weighting (FSW)

**Input:** Current task data $T_l = \{d_1, \ldots, d_k\}$, previous buffer data $\mathcal{M} = \cup_{y \in \mathbb{Y} - \mathbb{Y}_c, z \in \mathbb{Z}} G_{y,z}$, previous model $f_\theta^{l-1}$, loss function $\ell$, hyperparameters $\{\alpha, \lambda\}$, and fairness measure $F$

**Output:** Optimal training weights $\mathbf{w}_l^*$ for current task data

1: $\ell_G = [\ell(f_\theta^{l-1}, G_{1,1}), \ldots, \ell(f_\theta^{l-1}, G_{|\mathbb{Y}|,|\mathbb{Z}|})]$
2: $g_G = [\nabla_\theta \ell(f_\theta^{l-1}, G_{1,1}), \ldots, \nabla_\theta \ell(f_\theta^{l-1}, G_{|\mathbb{Y}|,|\mathbb{Z}|})]$
3: $g_d = [\nabla_\theta \ell(f_\theta^{l-1}, d_1), \ldots, \nabla_\theta \ell(f_\theta^{l-1}, d_k)]$
4: **switch** $F$ **do**
5:    **case** EER: $\mathbf{w}_l^* \leftarrow$ Solve Eq. 2
6:    **case** EO: $\mathbf{w}_l^* \leftarrow$ Solve Eq. 3
7:    **case** DP: $\mathbf{w}_l^* \leftarrow$ Solve Eq. 4
8: **return** $\mathbf{w}_l^*$

Alg. 2 shows the fairness-aware sample weighting (FSW) algorithm for the current task data. We first divide both the previous buffer data and the current task data into groups based on each class and sensitive attribute. Next, we compute the average loss and gradient vectors for each group (Steps 1–2), and individual gradient vectors for the current task data (Step 3). To compute gradient vectors, we use the last layer approximation, which only considers the gradients of the model's last layer, that is efficient and known to be reasonable (Katharopoulos & Fleuret, 2018; Ash et al., 2020; Mirzasoleiman et al., 2020). We then solve linear programming to find the optimal sample weights for a user-defined target fairness measure such as EER (Step 5), EO (Step 6), and DP (Step 7). We use CPLEX as a linear optimization solver that employs an efficient simplex-based algorithm. Since the gradient norm of the current task data is significantly larger than that of the buffer data, we utilize normalized gradients to update the loss of each group and replace the learning rate parameter $\eta$ with a hyperparameter $\alpha$ in the equations. Finally, we return the weights for the current task samples to be used during training (Step 8).

Training with FSW theoretically guarantees model convergence under the assumptions that the training loss is Lipschitz continuous and strongly convex, and that a proper learning rate is used (Killamsetty et al., 2021a; Chai & Wang, 2022; Lu et al., 2020). The computational complexity of FSW is quadratic to the number of current task samples, as CPLEX generally has quadratic complexity with respect to the number of variables when solving LP problems (Bixby, 2002). However, our empirical results show that for about twelve thousand current task samples, the time to solve an LP problem is a few seconds, which leads to a few minutes of overall runtime for MNIST datasets (see Sec. B.3 for details). Since we focus on continual offline training of large batches or separate tasks, rather than online learning, the overhead is manageable enough to deploy updated models in real-world applications. If the task size becomes too large, clustering similar samples and assigning weights to the clusters, rather than samples, could be a solution to reduce the computational overhead.

Table 1: Experimental settings for the five datasets.

| Dataset | Size | #Features | #Classes | #Tasks |
|---|---|---|---|---|
| MNIST | 60K | $28 \times 28$ | 10 | 5 |
| FMNIST | 60K | $28 \times 28$ | 10 | 5 |
| Biased MNIST | 60K | $3 \times 28 \times 28$ | 10 | 5 |
| DRUG | 1.3K | 12 | 6 | 3 |
| BiasBios | 253K | $128 \times 768$ | 25 | 5 |

## 4. Experiments

In this section, we construct experiments on our FSW and address the following research questions: **RQ1** How well can FSW mitigate the unfair forgetting that occurs in class-incremental learning with better accuracy-fairness tradeoff? **RQ2** How does FSW weight the samples? **RQ3** Can FSW be further integrated with fair post-processing techniques?

### 4.1. Experiment Settings

**Metrics**. We evaluate all methods using accuracy and fairness metrics as in the fair continual learning literature (Chowdhury & Chaturvedi, 2023; Truong et al., 2023).

• Average Accuracy. We denote $A_l = \frac{1}{l} \sum_{t=1}^{l} a_{l,t}$ as the accuracy at the $l^{th}$ task, where $a_{l,t}$ is the accuracy of the $t^{th}$ task after learning the $l^{th}$ task. We measure accuracy for each task and then take the average across all tasks to produce the final average accuracy, denoted as $\overline{A_l} = \frac{1}{L} \sum_{l=1}^{L} A_l$ where $L$ represents the total number of tasks.

• Average Fairness. We measure fairness for each task and then take the average across all tasks to produce the final average fairness. We use one of three measures: (1) *EER disparity*, which computes the average difference in test error rates among classes: $\frac{1}{|\mathbb{Y}|} \sum_{y \in \mathbb{Y}} |\Pr(\hat{y} \neq y | y = y) - \Pr(\hat{y} \neq y)|$; (2) *EO disparity*, which computes the average difference in accuracy among sensitive groups for all classes: $\frac{1}{|\mathbb{Y}||\mathbb{Z}|} \sum_{y \in \mathbb{Y}, z \in \mathbb{Z}} |\Pr(\hat{y} = y | y = y, z = z) - \Pr(\hat{y} = y | y = y)|$; and (3) *DP disparity*, which computes the average difference in class prediction ratios among sensitive groups for all classes: $\frac{1}{|\mathbb{Y}||\mathbb{Z}|} \sum_{y \in \mathbb{Y}, z \in \mathbb{Z}} |\Pr(\hat{y} = y | z = z) - \Pr(\hat{y} = y)|$. For all measures, low disparity is desirable.

**Datasets**. We use a total of five datasets as shown in Table 1. We first utilize commonly used benchmarks for continual image classification tasks, which include MNIST and Fashion-MNIST (FMNIST). Here we regard the class as the sensitive attribute and evaluate fairness with EER disparity. We also use multi-class fairness benchmark datasets that have sensitive attributes (Xu et al., 2020; Putzel & Lee, 2022; Churamani et al., 2023; Denis et al., 2023): Biased MNIST, Drug Consumption (DRUG), and BiasBios. We consider background color as the sensitive attribute for Biased MNIST, and gender for DRUG and BiasBios. We then use EO and DP disparity to evaluate fairness. We also consider using other datasets in the fairness field, but they

Table 2: Accuracy and fairness results with respect to (1) EER disparity, where class is considered the sensitive attribute for MNIST and FMNIST datasets, and (2) EO disparity, where background color or gender are the sensitive attributes for Biased MNIST, DRUG, and BiasBios datasets (see DP disparity results in Sec. B.7). We compare FSW with four types of baselines: naïve (*Joint Training* and *Fine Tuning*), state-of-the-art (*iCaRL*, *WA*, and *CLAD*), sample selection (*GSS* and *OCS*), and fairness-aware (*FaIRL*) methods. We mark the best and second-best results with **bold** and underline, respectively, excluding the naïve methods. Due to the excessive time required to run *OCS* on BiasBios, we are not able to measure the results.

| Methods | MNIST | | FMNIST | | Biased MNIST | | DRUG | | BiasBios | |
|---|---|---|---|---|---|---|---|---|---|---|
| | Acc. | EER Disp. | Acc. | EER Disp. | Acc. | EO Disp. | Acc. | EO Disp. | Acc. | EO Disp. |
| Joint Training | $.989_{\pm.000}$ | $.003_{\pm.000}$ | $.921_{\pm.002}$ | $.024_{\pm.002}$ | $.944_{\pm.002}$ | $.108_{\pm.003}$ | $.442_{\pm.015}$ | $.179_{\pm.052}$ | $.823_{\pm.002}$ | $.076_{\pm.001}$ |
| Fine Tuning | $.455_{\pm.000}$ | $.326_{\pm.000}$ | $.451_{\pm.000}$ | $.325_{\pm.000}$ | $.449_{\pm.001}$ | $.016_{\pm.002}$ | $.357_{\pm.009}$ | $.125_{\pm.034}$ | $.420_{\pm.001}$ | $.028_{\pm.002}$ |
| iCaRL | $.918_{\pm.005}$ | $.048_{\pm.003}$ | $\mathbf{.852}_{\pm.002}$ | $\underline{.047}_{\pm.001}$ | $.802_{\pm.008}$ | $.365_{\pm.021}$ | $\mathbf{.444}_{\pm.025}$ | $.190_{\pm.017}$ | $\mathbf{.829}_{\pm.002}$ | $.084_{\pm.003}$ |
| WA | $.911_{\pm.007}$ | $.052_{\pm.006}$ | $.809_{\pm.005}$ | $.088_{\pm.003}$ | $\mathbf{.916}_{\pm.002}$ | $.140_{\pm.004}$ | $.408_{\pm.022}$ | $.134_{\pm.029}$ | $.796_{\pm.003}$ | $.076_{\pm.001}$ |
| CLAD | $.835_{\pm.016}$ | $.099_{\pm.016}$ | $.782_{\pm.018}$ | $.118_{\pm.022}$ | $.871_{\pm.012}$ | $.198_{\pm.022}$ | $.410_{\pm.026}$ | $.114_{\pm.043}$ | $.799_{\pm.003}$ | $.074_{\pm.002}$ |
| GSS | $.889_{\pm.010}$ | $.080_{\pm.009}$ | $.732_{\pm.021}$ | $.149_{\pm.019}$ | $.809_{\pm.005}$ | $.325_{\pm.017}$ | $\underline{.426}_{\pm.010}$ | $.167_{\pm.038}$ | $\underline{.808}_{\pm.003}$ | $.081_{\pm.002}$ |
| OCS | $\mathbf{.929}_{\pm.002}$ | $\underline{.040}_{\pm.003}$ | $.799_{\pm.008}$ | $.109_{\pm.007}$ | $.824_{\pm.007}$ | $.331_{\pm.013}$ | $.406_{\pm.024}$ | $.142_{\pm.003}$ | – | – |
| FaIRL | $.558_{\pm.060}$ | $.273_{\pm.018}$ | $.531_{\pm.032}$ | $.289_{\pm.019}$ | $.411_{\pm.012}$ | $\mathbf{.118}_{\pm.011}$ | $.354_{\pm.011}$ | $\mathbf{.060}_{\pm.021}$ | $.400_{\pm.060}$ | $\mathbf{.055}_{\pm.020}$ |
| **FSW** | $\underline{.925}_{\pm.004}$ | $\mathbf{.032}_{\pm.005}$ | $.824_{\pm.006}$ | $\mathbf{.039}_{\pm.006}$ | $.909_{\pm.004}$ | $.119_{\pm.007}$ | $.406_{\pm.014}$ | $\underline{.077}_{\pm.010}$ | $.808_{\pm.002}$ | $\underline{.072}_{\pm.001}$ |

are unsuitable for class-incremental learning experiments because either there are only two classes, or it is difficult to apply group fairness metrics. See Sec. B.4 for more details.

**Models and Hyperparameters**. Following the experimental setups of Chaudhry et al. (2019a); Mirzadeh et al. (2020), we use a two-layer MLP with each 256 neurons for the MNIST, FMNIST, Biased MNIST, and DRUG datasets. For BiasBios dataset, we use a pre-trained BERT language model (Devlin et al., 2019). For our buffer storage, we store 32 samples per sensitive group for all experiments. For the hyperparameters $\alpha$, $\lambda$, and $\tau$ used in our algorithms, we perform cross-validation with a sequential grid search to find their optimal parameters. See Sec. B.5 for more details.

**Baselines**. We compare *FSW* with several baselines, including *iCaRL* (Rebuffi et al., 2017), *WA* (Zhao et al., 2020), *CLAD* (Xu et al., 2024), *GSS* (Aljundi et al., 2019) and *OCS* (Yoon et al., 2022). In particular, we consider *FaIRL* (Chowdhury & Chaturvedi, 2023) as the first fairness paper for continual learning. To obtain upper and lower bound performance, we included *Joint Training* and *Fine Tuning*, which have access to all previous data and no access to previous data, respectively. See Sec. B.6 for more details.

### 4.2. Accuracy and Fairness Results

To answer **RQ1**, we compare FSW against other baselines on the five datasets with respect to accuracy and corresponding fairness metrics as shown in Table 2 and Sec. B.7. The Pareto front on MNIST and Biased MNIST is represented in Fig. 2. For any method, we store a fixed number of samples per task in a buffer, which may not be identical to its original setup, but necessary for a fair comparison. The detailed sequential performance and accuracy-fairness tradeoff results are shown in Sec. B.8 and Sec. B.9, respectively. Additional results, including variations in buffer size, are in Sec. B.10.

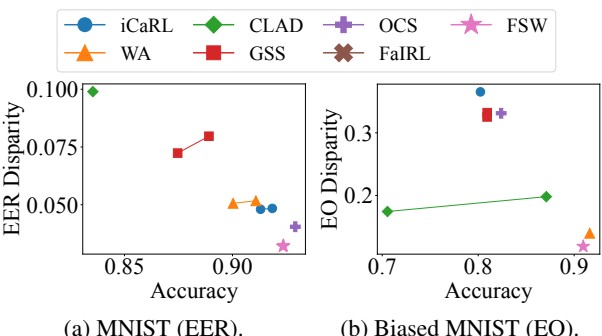

(a) MNIST (EER).  (b) Biased MNIST (EO).

Figure 2: Tradeoff results between accuracy and fairness on the MNIST and Biased MNIST datasets.

Overall, FSW achieves better accuracy-fairness tradeoff results compared to the baselines for all the datasets. For Biased MNIST, DRUG, and BiasBios, although FSW does not achieve the best performance in either accuracy or fairness, FSW shows the best fairness results among the baselines with similar accuracies (e.g., *iCaRL*, *WA*, *CLAD*, *GSS*, and *OCS*) and thus has the best accuracy-fairness tradeoff. We observe that FSW also improves model accuracy while enhancing the performance of underperforming groups for fairness. The state-of-the-art method, *iCaRL*, generally achieves high accuracy with low EER disparity results. However, since *iCaRL* uses a nearest-mean-of-exemplars approach for its classification model, the predictions are significantly affected by sensitive attribute values, resulting in high disparities for EO. The fairness-aware method *FaIRL* leverages an adversarial debiasing framework combined with a rate-distortion function, but the method loses significant accuracy because training the feature encoder and discriminator together is unstable. In comparison, FSW explicitly utilizes approximated loss and fairness measures to adjust the training weights for the current task samples, which leads to much better model accuracy and fairness.

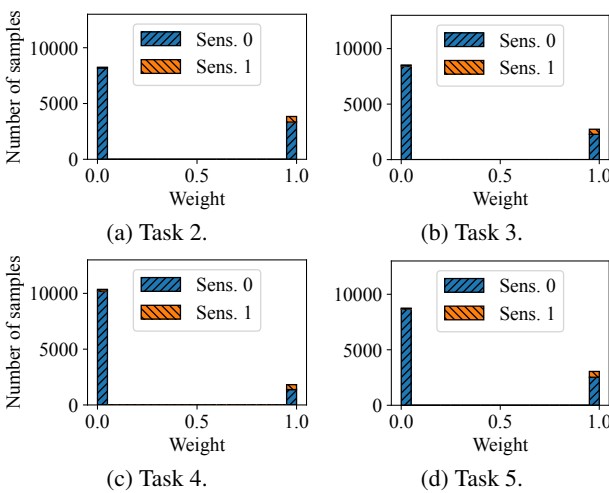

(a) Task 2.     (b) Task 3.

(c) Task 4.     (d) Task 5.

Figure 3: Distribution of sample weights for EO in sequential tasks of the Biased MNIST dataset.

### 4.3. Sample Weighting Analysis

To answer **RQ2**, we analyze how our FSW algorithm weights the current task samples at each task using the Biased MNIST dataset results shown in Fig. 3. The results for the other datasets are similar and shown in Sec. B.11. As the acquired sample weights may change with epochs during training, we show the average weight distribution of sensitive groups over all epochs. Since FSW is not applied to the first task, where the model is trained with only the current task data, we present results starting from the second task. Note that the acquired sample weights are mostly close to 0 or 1 in practice, but they are not strictly binary. See Sec. B.12 for more details.

Unlike naïve methods that use all the current task data with equal training weights, FSW assigns higher weights on average to the underperforming group (Sensitive group 1 in Fig. 3) compared to the overperforming group (Sensitive group 0 in Fig. 3). The weights are computed by considering complex forgetting relationships between sensitive groups, which differs from simply assigning higher weights to underperforming groups. We also observe that FSW assigns zero weight to a significant number of samples, indicating that relatively less data is used for training. This weighting approach provides an additional advantage in enabling efficient model training while retaining accuracy and fairness.

### 4.4. Ablation Study

To show the effectiveness of FSW on accuracy and fairness, we perform an ablation study comparing the performance of using FSW versus using all current task samples for training with equal weights. The results for the MNIST and Biased MNIST datasets are shown in Table 3. The results for DP disparity and other datasets, which are similar, can be found in Sec. B.13. As a result, applying sample weighting to the current task data is necessary to improve fairness while maintaining comparable accuracy.

Table 3: Accuracy and fairness results on the MNIST and Biased MNIST datasets with or without FSW.

| Methods | MNIST | | Biased MNIST | |
|---|---|---|---|---|
| | Acc. | EER Disp. | Acc. | EO Disp. |
| W/o FSW | $.912_{\pm.004}$ | $.051_{\pm.005}$ | $\mathbf{.910}_{\pm\mathbf{.003}}$ | $.126_{\pm.005}$ |
| **FSW** | $\mathbf{.925}_{\pm\mathbf{.004}}$ | $\mathbf{.032}_{\pm\mathbf{.005}}$ | $.909_{\pm.004}$ | $\mathbf{.119}_{\pm\mathbf{.007}}$ |

Table 4: Accuracy and fairness (DP disparity) results when combining fair post-processing technique ($\epsilon$-fair) with continual learning methods (*iCaRL*, *OCS*, and FSW).

| Methods | Biased MNIST | | DRUG | |
|---|---|---|---|---|
| | Acc. | DP Disp. | Acc. | DP Disp. |
| iCaRL | $.802_{\pm.008}$ | $.015_{\pm.001}$ | $\mathbf{.444}_{\pm\mathbf{.025}}$ | $.093_{\pm.009}$ |
| OCS | $.824_{\pm.007}$ | $.035_{\pm.003}$ | $.393_{\pm.017}$ | $.053_{\pm.012}$ |
| **FSW** | $.904_{\pm.004}$ | $.008_{\pm.001}$ | $.405_{\pm.013}$ | $.043_{\pm.004}$ |
| iCaRL – $\epsilon$-fair | $.944_{\pm.008}$ | $.006_{\pm.002}$ | $.427_{\pm.018}$ | $.026_{\pm.004}$ |
| OCS – $\epsilon$-fair | $\mathbf{.952}_{\pm\mathbf{.003}}$ | $.032_{\pm.004}$ | $.384_{\pm.009}$ | $.051_{\pm.002}$ |
| **FSW – $\epsilon$-fair** | $.906_{\pm.006}$ | $\mathbf{.005}_{\pm\mathbf{.001}}$ | $.405_{\pm.013}$ | $\mathbf{.021}_{\pm\mathbf{.004}}$ |

### 4.5. Integrating FSW with Fair Post-processing

To answer **RQ3**, we emphasize the extensibility of FSW by showing how it can be combined with a post-processing method to further improve fairness. We integrate FSW and other existing continual learning methods with the state-of-the-art fair post-processing technique in multi-class tasks, $\epsilon$-fair (Denis et al., 2023), as shown in Table 4 and Sec. B.14. Since $\epsilon$-fair only supports DP, we only show DP results in the Biased MNIST and DRUG datasets. We mark the best and second-best results with bold and underline, respectively, regardless of the application of post-processing. Overall, combining the fair post-processing technique can further improve fairness without degrading accuracy much. In addition, FSW still shows a better accuracy-fairness tradeoff with the combination of the fair post-processing technique, compared to existing continual learning methods.

## 5. Conclusion

We proposed FSW, a fairness-aware sample weighting algorithm for class-incremental learning. Unlike conventional class-incremental learning, we demonstrated how training with all the current task data using equal weights may result in unfair catastrophic forgetting. We theoretically showed that the average gradient vector of the current task data should not be solely in the opposite direction of the average gradient vector of a sensitive group to avoid unfair forgetting. We then proposed FSW as a solution to adjust the average gradient vector of the current task data, thereby achieving better accuracy-fairness tradeoff results. FSW supports various group fairness measures by converting the optimization problem into a linear program. In our experiments, FSW outperformed other baselines in terms of fairness while having comparable accuracy across various datasets with different domains. Future work will focus on generalizing to multiple sensitive attributes, as discussed in Sec. D.

**Impact Statement** We believe our work contributes to advancing the field of Trustworthy AI by addressing the critical yet understudied problem of fairness in class-incremental learning. While existing approaches have primarily focused on mitigating catastrophic forgetting to improve accuracy, we identify and tackle the issue of unfair catastrophic forgetting that disproportionately affects certain sensitive groups, including classes. Our Fairness-aware Sample Weighting (FSW) algorithm effectively balances accuracy and fairness by adjusting training weights to align gradient updates.

The implications of our work are significant for real-world applications where fairness is paramount, such as healthcare, autonomous systems, and personalized recommendations. Our framework not only enhances trust in continually learning models by reducing biases, but also provides a scalable and generalizable solution that supports diverse fairness measures. By addressing both technical challenges and ethical considerations, our work serves as an important step toward developing fairer, more transparent, and responsible AI systems.

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

# A. Appendix – Theory

## A.1. Theoretical Analysis of Unfairness in Class-Incremental Learning

Continuing from Sec. 3.1, we prove the lemma on the updated loss of a group of data after learning the current task data.

**Lemma A.1** (Restated from Lemma 3.1). *Denote $G$ as a sensitive group of data composed of features $X$ and true labels $y$. Also, denote $f_\theta^{l-1}$ as a previous model and $f_\theta$ as the updated model after training on the current task $T_l$. Let $\ell$ be any differentiable loss function (e.g., cross-entropy loss), and $\eta$ be a learning rate. Then, the loss of the sensitive group of data after training with a current task sample $d_i \in T_l$ is approximated as follows:*

$$\tilde{\ell}(f_\theta, G) = \ell(f_\theta^{l-1}, G) - \eta \nabla_\theta \ell(f_\theta^{l-1}, G)^\top \nabla_\theta \ell(f_\theta^{l-1}, d_i),$$

*where $\tilde{\ell}(f_\theta, G)$ is the approximated average loss between model predictions $f_\theta(X)$ and true labels $y$, whereas $\ell(f_\theta^{l-1}, G)$ is the exact average loss, $\nabla_\theta \ell(f_\theta^{l-1}, G)$ is the average gradient vector for the samples in the group $G$, and $\nabla_\theta \ell(f_\theta^{l-1}, d_i)$ is the gradient vector for a sample $d_i$, each with respect to the previous model $f_\theta^{l-1}$.*

*Proof.* We update the model using gradient descent with the current task sample $d_i \in T_l$ and learning rate $\eta$ as follows:

$$\theta = \theta^{l-1} - \eta \nabla_\theta \ell(f_\theta^{l-1}, d_i).$$

Using the Taylor series approximation,

$$\begin{aligned}
\tilde{\ell}(f_\theta, G) &= \ell(f_\theta^{l-1}, G) + \nabla_\theta \ell(f_\theta^{l-1}, G)^\top (\theta - \theta^{l-1}) \\
&= \ell(f_\theta^{l-1}, G) + \nabla_\theta \ell(f_\theta^{l-1}, G)^\top (-\eta \nabla_\theta \ell(f_\theta^{l-1}, d_i)) \\
&= \ell(f_\theta^{l-1}, G) - \eta \nabla_\theta \ell(f_\theta^{l-1}, G)^\top \nabla_\theta \ell(f_\theta^{l-1}, d_i).
\end{aligned}$$

If we update the model using all the current task data $T_l$, the equation is formulated as $\tilde{\ell}(f_\theta, G) = \ell(f_\theta^{l-1}, G) - \eta \nabla_\theta \ell(f_\theta^{l-1}, G)^\top \nabla_\theta \ell(f_\theta^{l-1}, T_l)$. Therefore, if the average gradient vectors of the sensitive group and the current task data have opposite directions, i.e., $\nabla_\theta \ell(f_\theta^{l-1}, G)^\top \nabla_\theta \ell(f_\theta^{l-1}, T_l) < 0$, learning the current task data increases the loss of the sensitive group data and finally leads to catastrophic forgetting. □

We next derive a sufficient condition for unfair forgetting.

**Theorem A.2** (Restated from Theorem 3.3). *Let $\ell$ be the cross-entropy loss and we denote $G_1$ and $G_2$ as the over-performing and underperforming groups of data, and $d_i$ as a training sample that satisfy the following conditions: $\ell(f_\theta^{l-1}, G_1) < \ell(f_\theta^{l-1}, G_2)$ while $\nabla_\theta \ell(f_\theta^{l-1}, G_1)^\top \nabla_\theta \ell(f_\theta^{l-1}, d_i) > 0$ and $\nabla_\theta \ell(f_\theta^{l-1}, G_2)^\top \nabla_\theta \ell(f_\theta^{l-1}, d_i) < 0$. Then $|\tilde{\ell}(f_\theta, G_1) - \tilde{\ell}(f_\theta, G_2)| > |\ell(f_\theta^{l-1}, G_1) - \ell(f_\theta^{l-1}, G_2)|$.*

*Proof.* Using the derived equation in the lemma A.1 $\tilde{\ell}(f_\theta, G) = \ell(f_\theta^{l-1}, G) - \eta \nabla_\theta \ell(f_\theta^{l-1}, G)^\top \nabla_\theta \ell(f_\theta^{l-1}, d_i)$, we compute the disparity of losses between the two groups $G_1$ and $G_2$ after the model update as follows:

$$\begin{aligned}
|\tilde{\ell}(f_\theta, G_1) - \tilde{\ell}(f_\theta, G_2)| &= |(\ell(f_\theta^{l-1}, G_1) - \eta \nabla_\theta \ell(f_\theta^{l-1}, G_1)^\top \nabla_\theta \ell(f_\theta^{l-1}, d_i)) - \\
&\quad (\ell(f_\theta^{l-1}, G_2) - \eta \nabla_\theta \ell(f_\theta^{l-1}, G_2)^\top \nabla_\theta \ell(f_\theta^{l-1}, d_i))| \\
&= |(\ell(f_\theta^{l-1}, G_1) - \ell(f_\theta^{l-1}, G_2)) - \\
&\quad \eta (\nabla_\theta \ell(f_\theta^{l-1}, G_1)^\top \nabla_\theta \ell(f_\theta^{l-1}, d_i) - \nabla_\theta \ell(f_\theta^{l-1}, G_2)^\top \nabla_\theta \ell(f_\theta^{l-1}, d_i))|.
\end{aligned}$$

Since $\ell(f_\theta^{l-1}, G_1) < \ell(f_\theta^{l-1}, G_2)$, it leads to $\ell(f_\theta^{l-1}, G_1) - \ell(f_\theta^{l-1}, G_2) < 0$. Next, the two assumptions of $\nabla_\theta \ell(f_\theta^{l-1}, G_1)^\top \nabla_\theta \ell(f_\theta^{l-1}, d_i) > 0$ and $\nabla_\theta \ell(f_\theta^{l-1}, G_2)^\top \nabla_\theta \ell(f_\theta^{l-1}, d_i) < 0$ make $-\eta(\nabla_\theta \ell(f_\theta^{l-1}, G_1)^\top \nabla_\theta \ell(f_\theta^{l-1}, d_i) - \nabla_\theta \ell(f_\theta^{l-1}, G_2)^\top \nabla_\theta \ell(f_\theta^{l-1}, d_i)) < 0$. Since the two terms in the absolute value equation are both negative,

$$\begin{aligned}
|\tilde{\ell}(f_\theta, G_1) - \tilde{\ell}(f_\theta, G_2)| &= |\ell(f_\theta^{l-1}, G_1) - \ell(f_\theta^{l-1}, G_2)| + \\
&\quad |-\eta(\nabla_\theta \ell(f_\theta^{l-1}, G_1)^\top \nabla_\theta \ell(f_\theta^{l-1}, d_i) - \nabla_\theta \ell(f_\theta^{l-1}, G_2)^\top \nabla_\theta \ell(f_\theta^{l-1}, d_i))| \\
&> |\ell(f_\theta^{l-1}, G_1) - \ell(f_\theta^{l-1}, G_2)|.
\end{aligned}$$

We finally have $|\tilde{\ell}(f_\theta, G_1) - \tilde{\ell}(f_\theta, G_2)| > |\ell(f_\theta^{l-1}, G_1) - \ell(f_\theta^{l-1}, G_2)|$, which implies that fairness deteriorates after training on the current task data. □

## A.2. From Cross-Entropy Loss to Group Fairness Metrics

Continuing from Sec. 3.1, we explain how to approximate the group fairness metrics using cross-entropy loss disparity. Existing works (Shen et al., 2022; Roh et al., 2021; 2023) empirically verified that using the cross-entropy loss disparity can provide reasonable proxies for common group fairness metrics such as equalized odds (EO) and demographic parity (DP) disparity. In addition, we theoretically describe how minimizing the cost function for EO using the cross-entropy loss disparity (i.e., $L_{EO} = \frac{1}{|\mathbb{Y}||\mathbb{Z}|} \sum_{y \in \mathbb{Y}, z \in \mathbb{Z}} |\ell(f_\theta, G_{y,z}) - \ell(f_\theta, G_y)|$ where $\ell$ is a cross-entropy loss) leads to ensuring EO disparity. (Shen et al., 2022) theoretically and empirically showed that using cross-entropy loss instead of the 0-1 loss (i.e., $\mathbf{1}(\mathbf{y} \neq \hat{\mathbf{y}})$ where $\mathbf{1}(\cdot)$ is an indicator function, which is equivalent to the probability of correct prediction) can still capture EO disparity in binary classification. We now prove how applying the cross-entropy loss disparity for EO can be extended to multi-class classification as follows:

Let $m_{y,z}$ be the size of a sensitive group (i.e., $m_{y,z} := |\{i : \mathbf{y}_i = y, \mathbf{z}_i = z\}|$) and $\mathbb{Y}$ be a set of all classes y. Also, let

$\begin{pmatrix} \vdots \\ \mathbf{y}_i^j \\ \vdots \end{pmatrix}$ be the one-hot encoding vector of $\mathbf{y}_i$. Similarly, $\hat{\mathbf{y}}_i$ is a predicted label and $\begin{pmatrix} \vdots \\ \hat{\mathbf{y}}_i^j \\ \vdots \end{pmatrix}$ denotes a probability distribution

for each label of the sample $i$. Then, the cross-entropy loss for a sensitive group $G_{y,z}$ can be transformed as follows:

$$\ell(f_\theta, G_{y,z}) = -\frac{1}{m_{y,z}} \sum_{i=1}^{m_{y,z}} \left( \sum_{j=1}^{|\mathbb{Y}|} \mathbf{y}_i^j \cdot \log(\hat{\mathbf{y}}_i^j) \right)$$

$$= -\frac{1}{m_{y,z}} \sum_{i=1}^{m_{y,z}} \log(\hat{\mathbf{y}}_i^y).$$

Since $\hat{\mathbf{y}}_i^y$ is equivalent to $p(\hat{\mathbf{y}}_i = y)$ and we are measuring a loss for the sensitive group (y $= y$, z $= z$), $\ell(f_\theta, G_{y,z}) = -\frac{1}{m_{y,z}} \sum_i \log(p(\hat{\mathbf{y}}_i))$ is an unbiased estimator of $-\log p(\hat{\mathbf{y}}|\mathbf{y} = y, \mathbf{z} = z)$. Likewise, $\ell(f_\theta, G_y)$ is an unbiased estimator of $-\log p(\hat{\mathbf{y}}|\mathbf{y} = y)$ and our cost function becomes equivalent to $\left| \log \frac{p(\hat{\mathbf{y}}|\mathbf{y}=y)}{p(\hat{\mathbf{y}}|\mathbf{y}=y,\mathbf{z}=z)} \right|$. Since $\frac{p(\hat{\mathbf{y}}|\mathbf{y}=y)}{p(\hat{\mathbf{y}}|\mathbf{y}=y,\mathbf{z}=z)} = 1$ for all $y$, $z$ implies $\hat{Y} \perp\!\!\!\perp Z \mid Y$, we conclude that minimizing the cost function for EO can satisfy equalized odds.

We next perform experiments to evaluate how well the cost function for EO approximates EO disparity (i.e., $\frac{1}{|\mathbb{Y}||\mathbb{Z}|} \sum_{y \in \mathbb{Y}, z \in \mathbb{Z}} |\Pr(\hat{\mathbf{y}} = y|\mathbf{y} = y, \mathbf{z} = z) - \Pr(\hat{\mathbf{y}} = y|\mathbf{y} = y)|$) on the Biased MNIST dataset as shown in Fig. 4. Although the scales of the two metrics are different, the simultaneous movements of these two trends suggest that our cost function is effective in satisfying equalized odds.

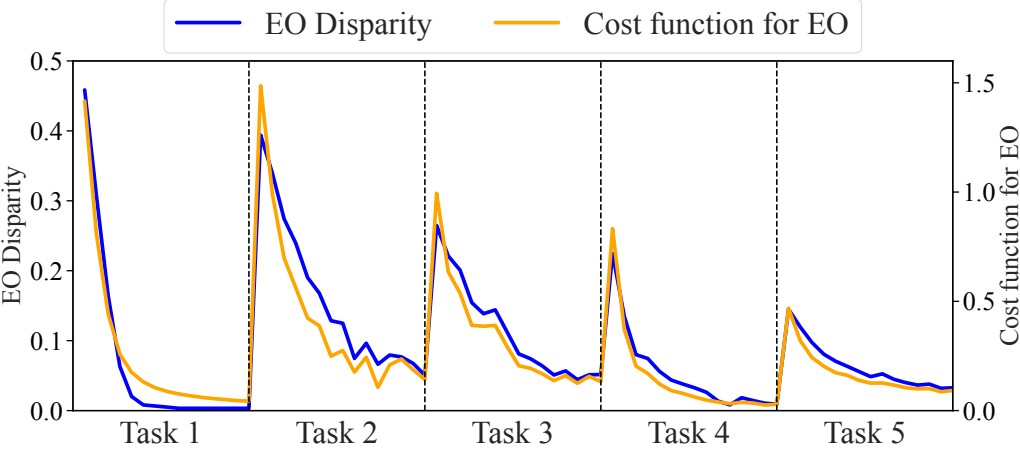

Figure 4: Comparison of EO disparity and cost function for EO during training on the Biased MNIST dataset. We train a model for 15 epochs per task.

## A.3. Derivation of a Sufficient Condition for Demographic Parity in the Multi-Class Setting

Continuing from Sec. 3.2, we derive a sufficient condition for satisfying demographic parity in the multi-class setting.

**Proposition A.3.** *In the multi-class setting, $\frac{m_{y,z_1}}{m_{*,z_1}}\ell(f_\theta, G_{y,z_1}) = \frac{m_{y,z_2}}{m_{*,z_2}}\ell(f_\theta, G_{y,z_2})$ where $m_{y,z} := |\{i : y_i = y, z_i = z\}|$ and $m_{*,z} := |\{i : z_i = z\}|$ for $y \in \mathbb{Y}$ and $z_1, z_2 \in \mathbb{Z}$ can serve as a sufficient condition for demographic parity.*

*Proof.* In the multi-class setting, we can extend the definition of demographic parity as $\Pr(\hat{y} = y|z = z_1) = \Pr(\hat{y} = y|z = z_2)$ for $y \in \mathbb{Y}$ and $z_1, z_2 \in \mathbb{Z}$. The term $\Pr(\hat{y} = y|z = z)$ can be decomposed as follows: $\Pr(\hat{y} = y|z = z) = \Pr(\hat{y} = y, y = y|z = z) + \sum_{y_n \neq y} \Pr(\hat{y} = y, y = y_n|z = z)$. Without loss of generality, we set $z_1 = 0$ and $z_2 = 1$. Then the definition of demographic parity in the multi-class setting now becomes

$$\Pr(\hat{y} = y, y = y|z = 0) + \sum_{y_n \neq y} \Pr(\hat{y} = y, y = y_n|z = 0)$$

$$= \Pr(\hat{y} = y, y = y|z = 1) + \sum_{y_n \neq y} \Pr(\hat{y} = y, y = y_n|z = 1).$$

The term $\Pr(\hat{y} = y, y = y|z = 0)$ can be represented with the 0-1 loss as follows:

$$\Pr(\hat{y} = y, y = y|z = 0) = \frac{\Pr(\hat{y} = y, y = y, z = 0)}{\Pr(z = 0)}$$

$$= \frac{\Pr(\hat{y} = y|y = y, z = 0)\Pr(y = y, z = 0)}{\Pr(z = 0)}$$

$$= \frac{1}{m_{*,0}} \sum_{i:y_i=y,z_i=0} (1 - \mathbb{1}(y_i \neq \hat{y}_i)).$$

Similarly, $\Pr(\hat{y} = y, y = y_n|z = 0)$ for $y_n \neq y$ can be transformed as follows:

$$\Pr(\hat{y} = y, y = y_n|z = 0) = \frac{\Pr(\hat{y} = y, y = y_n, z = 0)}{\Pr(z = 0)}$$

$$= \frac{\Pr(\hat{y} = y|y = y_n, z = 0)\Pr(y = y_n, z = 0)}{\Pr(z = 0)}$$

$$= \frac{1}{m_{*,0}} \sum_{j:y_j=y_n,z_j=0} \mathbb{1}(y_j \neq \hat{y}_j).$$

By applying the same technique to $\Pr(\hat{y} = y, y = y|z = 1)$ and $\Pr(\hat{y} = y, y = y_n|z = 1)$, we have the 0-1 loss-based definition of demographic parity:

$$\frac{1}{m_{*,0}} \sum_{i:y_i=y,z_i=0} (1 - \mathbb{1}(y_i \neq \hat{y}_i)) + \sum_{i:y_i\neq y} \frac{1}{m_{*,0}} \sum_{j:y_j=y_i,z_j=0} \mathbb{1}(y_j \neq \hat{y}_j)$$

$$= \frac{1}{m_{*,1}} \sum_{i:y_i=y,z_i=1} (1 - \mathbb{1}(y_i \neq \hat{y}_i)) + \sum_{i:y_i\neq y} \frac{1}{m_{*,1}} \sum_{j:y_j=y_i,z_j=1} \mathbb{1}(y_j \neq \hat{y}_j).$$

Since the 0-1 loss is not differentiable, it is not suitable to approximate the updated loss using gradients as in Eq. 1. We thus approximate the 0-1 loss to a standard loss function $\ell$ (e.g., cross-entropy loss),

$$\frac{1}{m_{*,0}} \sum_{i:y_i=y,z_i=0} -\ell(f_\theta, d_i) + \sum_{i:y_i\neq y} \frac{1}{m_{*,0}} \sum_{j:y_j=y_i,z_j=0} \ell(f_\theta, d_j)$$

$$= \frac{1}{m_{*,1}} \sum_{i:y_i=y,z_i=1} -\ell(f_\theta, d_i) + \sum_{i:y_i\neq y} \frac{1}{m_{*,1}} \sum_{j:y_j=y_i,z_j=1} \ell(f_\theta, d_j),$$

where $\ell(f_\theta, d_j)$ is the loss between the model prediction $f_\theta(d_j)$ and the true label $y_j$. By replacing $\sum_{i:y_i=y,z_i=z} \ell(f_\theta, d_i) = m_{y,z}\ell(f_\theta, G_{y,z})$,

$$\frac{m_{y,0}}{m_{*,0}}(-\ell(f_\theta, G_{y,0})) + \sum_{i:y_i\neq y} \frac{m_{y_i,0}}{m_{*,0}}\ell(f_\theta, G_{y_i,0}) = \frac{m_{y,1}}{m_{*,1}}(-\ell(f_\theta, G_{y,1})) + \sum_{i:y_i\neq y} \frac{m_{y_i,1}}{m_{*,1}}\ell(f_\theta, G_{y_i,1}).$$

To satisfy the constraint for all $y \in \mathbb{Y}$, the corresponding terms on the left-hand side and the right-hand side of the equation should be equal, i.e., $\frac{m_{y,0}}{m_{*,0}} \ell(f_\theta, G_{y,0}) = \frac{m_{y,1}}{m_{*,1}} \ell(f_\theta, G_{y,1})$. In general, we derive a sufficient condition for demographic parity as $\frac{m_{y,z_1}}{m_{*,z_1}} \ell(f_\theta, G_{y,z_1}) = \frac{m_{y,z_2}}{m_{*,z_2}} \ell(f_\theta, G_{y,z_2})$. Note that the number of samples in sensitive groups (e.g., $m_{*,z}$ and $m_{y,z}$) is derived from the definition of demographic parity, which is independent of sample weights. $\qquad\square$

### A.4. LP Formulation of Fairness-aware Optimization Problems

Continuing from Sec. 3.2, we prove Theorem 3.4, which implies that fairness-aware optimization problems can be transformed into linear programming problems. This transformation is made possible by using Lemma A.4, which suggests that minimizing the sum of absolute values with linear terms can be transformed into a linear programming form.

**Lemma A.4.** *The following optimization problem can be reformulated into a linear programming form. Note that in the following equation, y and z refer to arbitrary variables, not to the label or sensitive attribute, respectively.*

$$\min_{\mathbf{x}} \sum_{i=1}^{n} |y_i| + z_i$$

$$s.t. \quad y_i = a_i - \mathbf{b}_i^\top \mathbf{x}, \quad z_i = c_i - \mathbf{d}_i^\top \mathbf{x}$$
$$a_i, c_i, y_i, z_i \in \mathbb{R}, \quad \mathbf{b}_i, \mathbf{d}_i \in \mathbb{R}^{m \times 1} \quad \forall i \in \{1, \ldots, n\}, \quad \mathbf{x} \in [0,1]^{m \times 1}.$$

*Proof.* The transformation for minimizing the sum of absolute values was introduced in (Ferguson & Sargent, 1958; McCarl & Spreen, 1997; Asghari et al., 2022). Note that considering the additional affine term does not affect the flow of the proof. We first substitute $y_i$ for $y_i^+ - y_i^-$ where both $y_i^+$ and $y_i^-$ are nonnegative. Then, the optimization problem becomes

$$\min_{\mathbf{x}} \sum_{i=1}^{n} |y_i^+ - y_i^-| + z_i$$

$$\text{s.t.} \quad y_i^+ - y_i^- = a_i - \mathbf{b}_i^\top \mathbf{x}, \quad z_i = c_i - \mathbf{d}_i^\top \mathbf{x}, \quad y_i^+ - y_i^- = y_i$$
$$y_i^+, y_i^- \in \mathbb{R}^+, \quad a_i, c_i, y_i, z_i \in \mathbb{R}, \quad \mathbf{b}_i, \mathbf{d}_i \in \mathbb{R}^{m \times 1} \quad \forall i \in \{1, \ldots, n\}, \quad \mathbf{x} \in [0,1]^{m \times 1}.$$

This problem is still nonlinear. However, the absolute value terms can be simplified when either $y_i^+$ or $y_i^-$ equals to zero (i.e., $y_i^+ y_i^- = 0$), as the consequent absolute value reduces to zero plus the other term. Then, the absolute value term can be written as the sum of two variables,

$$|y_i^+ - y_i^-| = |y_i^+| + |y_i^-| = y_i^+ + y_i^- \quad \text{if} \quad y_i^+ y_i^- = 0.$$

By using the assumption, the formulation becomes

$$\min_{\mathbf{x}} \sum_{i=1}^{n} y_i^+ + y_i^- + z_i$$

$$\text{s.t.} \quad y_i^+ - y_i^- = a_i - \mathbf{b}_i^\top \mathbf{x}, \quad z_i = c_i - \mathbf{d}_i^\top \mathbf{x}, \quad y_i^+ - y_i^- = y_i, \quad \underline{y_i^+ y_i^- = 0}$$
$$y_i^+, y_i^- \in \mathbb{R}^+, \quad a_i, c_i, y_i, z_i \in \mathbb{R}, \quad \mathbf{b}_i, \mathbf{d}_i \in \mathbb{R}^{m \times 1} \quad \forall i \in \{1, \ldots, n\}, \quad \mathbf{x} \in [0,1]^{m \times 1}$$

with the underlined condition added. However, this condition can be dropped. Assume there exist $y_i^+$ and $y_i^-$, which do not satisfy $y_i^+ y_i^- = 0$. When $y_i^+ \geq y_i^- > 0$, there exists a better solution $(y_i^+ - y_i^-, 0)$ instead of $(y_i^+, y_i^-)$, which satisfies all the conditions, but has a smaller objective function value $y_i^+ - y_i^- + 0 + z_i < y_i^+ + y_i^- + z_i$. For the case of $y_i^- > y_i^+ > 0$, a solution $(0, y_i^- - y_i^+)$ works better for similar reasons. Thus, the minimization automatically leads to $y_i^+ y_i^- = 0$, and the underlined nonlinear constraint becomes unnecessary. Consequently, the final formulation becomes this linear problem:

$$\min_{\mathbf{x}} \sum_{i=1}^{n} y_i^+ + y_i^- + z_i$$

$$\text{s.t.} \quad y_i^+ - y_i^- = a_i - \mathbf{b}_i^\top \mathbf{x}, \quad z_i = c_i - \mathbf{d}_i^\top \mathbf{x}, \quad y_i^+ - y_i^- = y_i$$
$$y_i^+, y_i^- \in \mathbb{R}^+, \quad a_i, c_i, y_i, z_i \in \mathbb{R}, \quad \mathbf{b}_i, \mathbf{d}_i \in \mathbb{R}^{m \times 1} \quad \forall i \in \{1, \ldots, n\}, \quad \mathbf{x} \in [0,1]^{m \times 1}.$$

$\qquad\square$

Applying Lemma A.4, we next prove Theorem 3.4. By using the result of Theorem 3.4, we show that the fairness-aware optimization problems, where the objective function includes both fairness ($L_{fair}$) and accuracy ($L_{acc}$) losses, can be transformed into linear programming (LP) problems.

**Theorem A.5** (Restated from Theorem 3.4). *The fairness-aware optimization problems (Eq. 2, 3, and 4) can be transformed into the form of linear programming (LP) problems.*

*Proof.* For every update of the model, the corresponding loss of each group can be approximated linearly in the same way as in Sec. A.1: $\tilde{\ell}(f_\theta, G) = \ell(f_\theta^{l-1}, G) - \eta \nabla_\theta \ell(f_\theta^{l-1}, G)^\top \nabla_\theta \ell(f_\theta^{l-1}, T_l)$. With a technique of sample weighting for the current task data, $\nabla_\theta \ell(f_\theta^{l-1}, T_l)$ can be changed as $\frac{1}{|T_l|} \sum_{d_i \in T_l} \mathbf{w}_l^i \nabla_\theta \ell(f_\theta^{l-1}, d_i)$ where $\mathbf{w}_l^i$ represents a training weight for the current task sample $d_i$.

We believe that this transformation is natural and valid, as models are generally updated using the average gradient of training data, formulated as $\frac{1}{|T_l|} \sum_{d_i \in T_l} \nabla_\theta \ell(f_\theta^{l-1}, d_i)$, and a training weight is additionally assigned to each sample for weighting. Here, $|T_l|$ is the number of samples in the current task data, and this is independent of the fairness notions considered. Note that if the normalization coefficient $\frac{1}{|T_l|}$ is replaced with $\frac{1}{\sum \mathbf{w}_l^i}$, the revised equation cannot handle the case where all weights are zero. Also, our revised optimization problems of Eq. 2, 3, and 4 would no longer be linear programs.

Thus, $\tilde{\ell}(f_\theta, G)$ can be rewritten as follows:

$$\tilde{\ell}(f_\theta, G) = \ell(f_\theta^{l-1}, G) - \eta \nabla_\theta \ell(f_\theta^{l-1}, G)^\top \left( \frac{1}{|T_l|} \sum_{d_i \in T_l} \mathbf{w}_l^i \nabla_\theta \ell(f_\theta^{l-1}, d_i) \right)$$

$$= \ell(f_\theta^{l-1}, G) - \frac{\eta}{|T_l|} \nabla_\theta \ell(f_\theta^{l-1}, G)^\top \begin{bmatrix} \cdots & \nabla_\theta \ell(f_\theta^{l-1}, d_i) & \cdots \end{bmatrix} \begin{bmatrix} \vdots \\ \mathbf{w}_l^i \\ \vdots \end{bmatrix}$$

$$= a_G - \mathbf{b}_G^\top \mathbf{w},$$

where $a_G := \ell(f_\theta^{l-1}, G)$ and $\mathbf{b}_G := \frac{\eta}{|T_l|} \begin{bmatrix} \cdots & \nabla_\theta \ell(f_\theta^{l-1}, d_i) & \cdots \end{bmatrix}^\top \nabla_\theta \ell(f_\theta^{l-1}, G)$ are a constant and a vector with

constants, respectively, and $\mathbf{w} := \begin{bmatrix} \vdots \\ w_l^i \\ \vdots \end{bmatrix}$ is a variable where $w_l^i \in [0, 1]$.

*Case* 1. If target fairness measure is EER ($L_{fair} = L_{EER}$),

$$L_{EER} + \lambda L_{acc} = \frac{1}{|\mathbb{Y}|} \sum_{y \in \mathbb{Y}} |\tilde{\ell}(f_\theta, G_y) - \tilde{\ell}(f_\theta, G_\mathbb{Y})| + \lambda \frac{1}{|\mathbb{Y}_c|} \sum_{y \in \mathbb{Y}_c} \tilde{\ell}(f_\theta, G_y)$$

$$= \frac{1}{|\mathbb{Y}|} \sum_{y \in \mathbb{Y}} |(a_{G_y} - a_{G_\mathbb{Y}}) - (\mathbf{b}_{G_y} - \mathbf{b}_{G_\mathbb{Y}})^\top \mathbf{w}| + \lambda \frac{1}{|\mathbb{Y}_c|} \sum_{y \in \mathbb{Y}_c} (a_{G_y} - \mathbf{b}_{G_y}^\top \mathbf{w}).$$

*Case* 2. If target fairness measure is EO ($L_{fair} = L_{EO}$),

$$L_{EO} + \lambda L_{acc} = \frac{1}{|\mathbb{Y}||\mathbb{Z}|} \sum_{y \in \mathbb{Y}, z \in \mathbb{Z}} |\tilde{\ell}(f_\theta, G_{y,z}) - \tilde{\ell}(f_\theta, G_y)| + \lambda \frac{1}{|\mathbb{Y}_c||\mathbb{Z}|} \sum_{y \in \mathbb{Y}_c, z \in \mathbb{Z}} \tilde{\ell}(f_\theta, G_{y,z})$$

$$= \frac{1}{|\mathbb{Y}||\mathbb{Z}|} \sum_{y \in \mathbb{Y}, z \in \mathbb{Z}} |(a_{G_{y,z}} - a_{G_y}) - (\mathbf{b}_{G_{y,z}} - \mathbf{b}_{G_y})^\top \mathbf{w}| +$$

$$\lambda \frac{1}{|\mathbb{Y}_c||\mathbb{Z}|} \sum_{y \in \mathbb{Y}_c, z \in \mathbb{Z}} (a_{G_{y,z}} - \mathbf{b}_{G_{y,z}}^\top \mathbf{w}).$$

*Case* 3. If target fairness measure is DP ($L_{fair} = L_{DP}$),

$$L_{DP} + \lambda L_{acc} = \frac{1}{|\mathbb{Y}||\mathbb{Z}|} \sum_{y \in \mathbb{Y}, z \in \mathbb{Z}} |\tilde{\ell}'(f_\theta, G_{y,z}) - \tilde{\ell}'(f_\theta, G_y)| + \lambda \frac{1}{|\mathbb{Y}_c||\mathbb{Z}|} \sum_{y \in \mathbb{Y}_c, z \in \mathbb{Z}} \tilde{\ell}(f_\theta, G_{y,z})$$

$$= \frac{1}{|\mathbb{Y}||\mathbb{Z}|} \sum_{y \in \mathbb{Y}, z \in \mathbb{Z}} |(a'_{G_{y,z}} - a'_{G_y}) - (\mathbf{b}'_{G_{y,z}} - \mathbf{b}'_{G_y})^\top \mathbf{w}| +$$

$$\lambda \frac{1}{|\mathbb{Y}_c||\mathbb{Z}|} \sum_{y \in \mathbb{Y}_c, z \in \mathbb{Z}} (a_{G_{y,z}} - \mathbf{b}_{G_{y,z}}^\top \mathbf{w}),$$

where $a'_{G_{y,z}} := \frac{m_{y,z}}{m_{*,z}} a_{G_{y,z}}$, $a'_{G_y} := \sum_{z \in \mathbb{Z}} \frac{m_{y,z}}{m_{*,z}} a_{G_{y,z}}$, $\mathbf{b}'_{G_{y,z}} := \frac{m_{y,z}}{m_{*,z}} \mathbf{b}_{G_{y,z}}$, $\mathbf{b}'_{G_y} := \sum_{z \in \mathbb{Z}} \frac{m_{y,z}}{m_{*,z}} \mathbf{b}_{G_{y,z}}$.

Since $a_G$ and $\mathbf{b}_G$ are composed of constant values, each equation above can be reformulated to a linear programming form by applying the above lemma. □

## B. Appendix – Experiments

### B.1. T-SNE Results for Real Datasets

Continuing from Sec. 1, we provide t-SNE results for real datasets to show that data overlapping between different classes also occurs in real scenarios, similar to the synthetic dataset results depicted in Fig. 1a. Using t-SNE, we project the high-dimensional data of the MNIST, FMNIST, Biased MNIST, and DRUG datasets into a lower-dimensional 2D space with $x_1$ and $x_2$, as shown in Fig. 5. Since BiasBios is a text dataset that requires pre-trained embeddings to represent the data, we do not include the t-SNE results for it. In the MNIST dataset, the images with labels of 3 (red), 5 (brown), and 8 (yellow) exhibit similar characteristics and overlap, but belong to different classes. As another example, in the FMNIST dataset, the images of the classes 'Sandal' (brown), 'Sneaker' (gray), and 'Ankel boot' (sky-blue) also have similar characteristics and overlap.

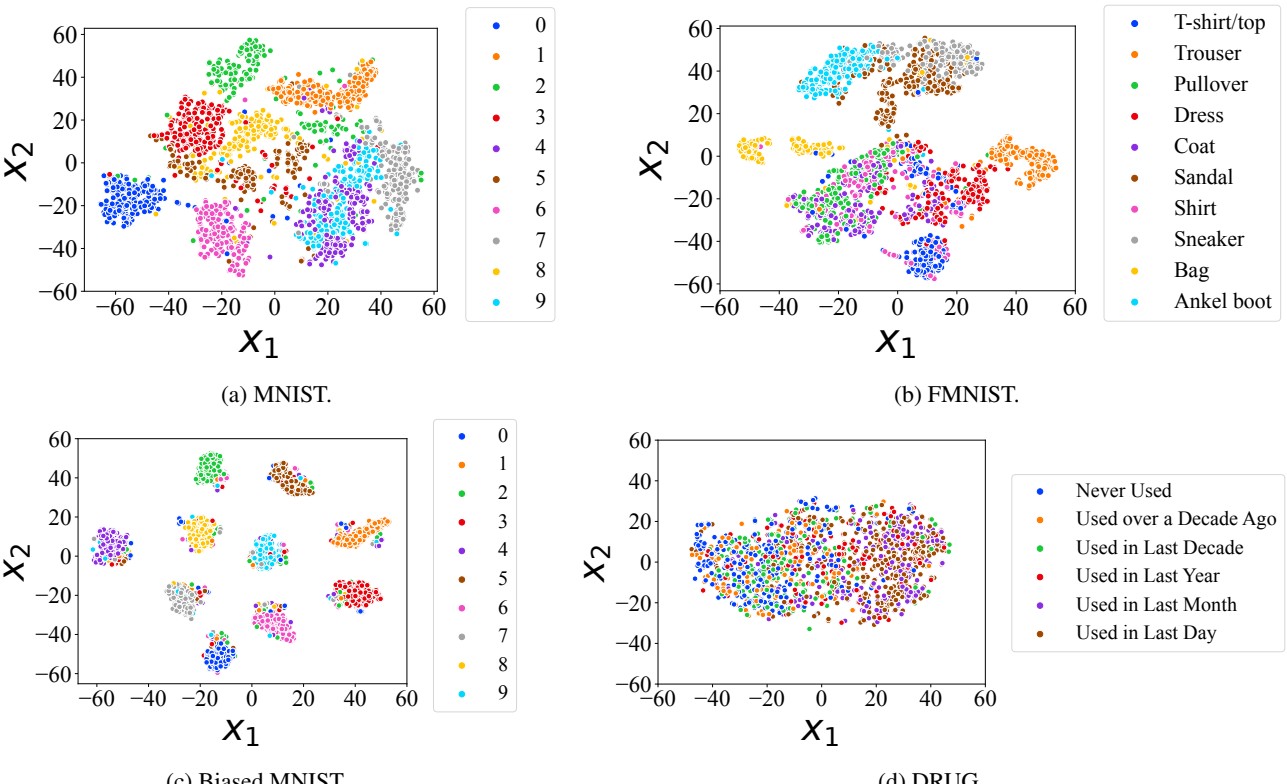

(a) MNIST.

(b) FMNIST.

(c) Biased MNIST.

(d) DRUG.

Figure 5: t-SNE results for the MNIST, FMNIST, Biased MNIST, and DRUG datasets.

## B.2. Approximation Error of Taylor Series

Continuing from Sec. 3.1, we provide empirical approximation errors between true losses and approximated losses derived from first-order Taylor series on the MNIST and Biased MNIST datasets as shown in Fig. 6. For each task, we train the model for 5 epochs and 15 epochs on the MNIST and Biased MNIST datasets, respectively. The approximation error is large when a new task begins because new samples with unseen classes are introduced. However, the error gradually decreases as the number of epochs increases while training a model for the task.

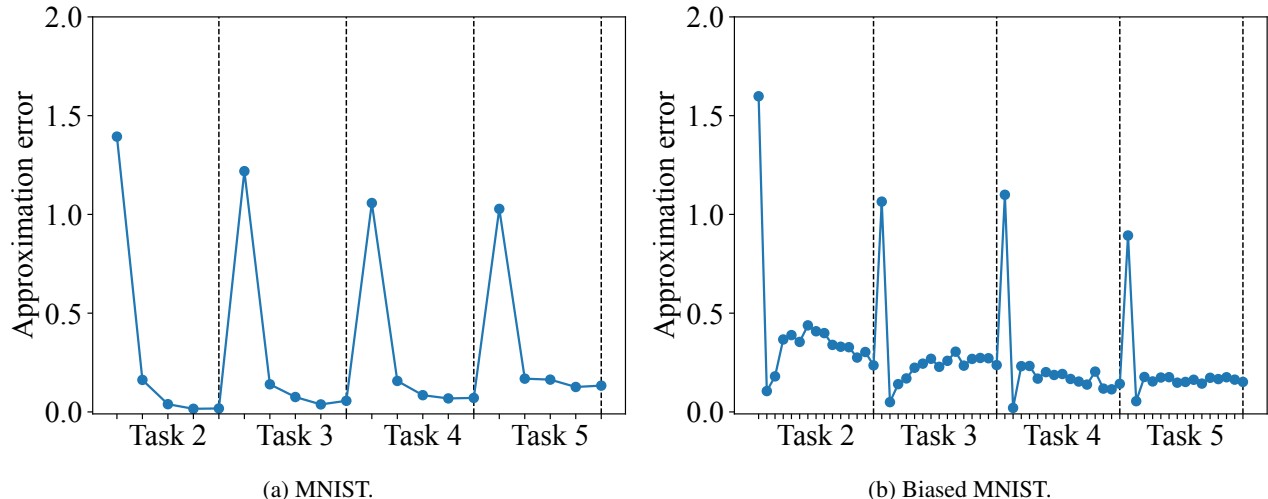

(a) MNIST.  (b) Biased MNIST.

Figure 6: Absolute errors between true losses and approximated losses derived from first-order Taylor series while training a model.

## B.3. Computational Complexity and Runtime Results of FSW

Continuing from Sec. 3.3, we provide computational complexity and overall runtime results of FSW using the MNIST and Biased MNIST datasets as shown in Fig. 7 and Fig. 8. Our empirical results show that for about twelve thousand current-task samples, the time to solve an LP problem is a few seconds for the MNIST dataset as shown in Fig. 7. By applying the log-log regression model to the results in Fig. 7, the computational complexity of solving LP at each epoch is $\mathcal{O}(|T_l|^{1.642})$ where $|T_l|$ denotes the number of current task samples. We note that this complexity can be quadratic in the worst case. If the task size becomes too large, we believe that clustering similar samples and assigning weights to the clusters, rather than samples, could be a solution to reduce the computational overhead. In Fig. 8, we compute the overall runtime of FSW divided into three steps: Gradient Computation, CPLEX Computation, and Model Training.

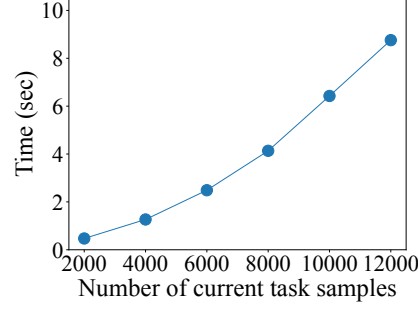

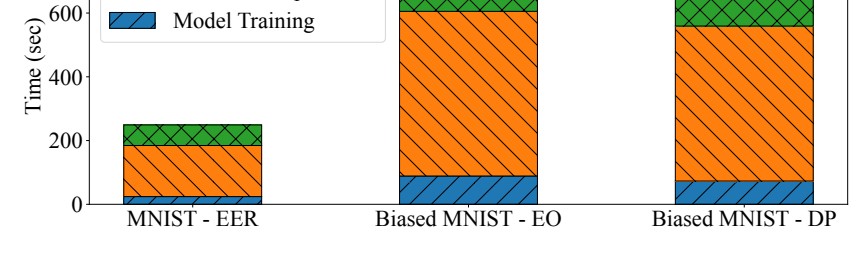

Figure 7: Runtime results of solving a single LP problem in FSW using CPLEX for the MNIST dataset.

Figure 8: Overall runtime results of our framework on all tasks for three datasets: MNIST–EER, Biased MNIST–EO, and Biased MNIST–DP.

**B.4. More Details on Datasets**

Continuing from Sec. 4.1, we provide more details of the two datasets using the class as the sensitive attribute and the three datasets with separate sensitive attributes. For datasets with a total of $C$ classes, we divide the datasets into $L$ sequences of tasks where each task consists of $C/L$ classes, and assume that task boundaries are available (van de Ven & Tolias, 2019). We also consider using standard benchmark datasets in the fairness field, but they are unsuitable for class-incremental learning experiments either because there are only two classes (e.g., COMPAS (Angwin et al., 2016), AdultCensus (Kohavi, 1996), and Jigsaw (cjadams, 2019)), or because it is difficult to apply group fairness metrics. For instance, in the case of CelebA (Liu et al., 2015), each person is considered a class, making the sensitive attribute dependent on the true label.

- **MNIST** (LeCun et al., 1998): The MNIST dataset is a standard benchmark for evaluating the performance of machine learning models, especially in image classification tasks. The dataset is a collection of grayscale images of handwritten digits ranging from 0 to 9, each measuring 28 pixels in width and 28 pixels in height. The dataset consists of 60,000 training images and 10,000 test images. We configure a class-incremental learning setup, where a total of 10 classes are evenly distributed across 5 tasks, with 2 classes per task. We assume the class itself is the sensitive attribute.

- **Fashion-MNIST (FMNIST)** (Xiao et al., 2017): The Fashion-MNIST dataset is a specialized variant of the original MNIST dataset, designed for the classification of various clothing items into 10 distinct classes. The classes include 'T-shirt/top', 'Trouser', 'Pullover', 'Dress', 'Coat', 'Sandal', 'Shirt', 'Sneaker', 'Bag', and 'Ankle boot'. The dataset consists of grayscale images with dimensions of 28 pixels by 28 pixels including 60,000 training images and 10,000 test images. We configure a class-incremental learning setup, where a total of 10 classes are evenly distributed across 5 tasks, with 2 classes per task. We assume the class itself is the sensitive attribute.

- **Biased MNIST** (Bahng et al., 2020): The Biased MNIST dataset is a modified version of the MNIST dataset that introduces bias by incorporating background colors highly correlated with the digits. We select 10 distinct background colors and assign one to each digit from 0 to 9. For the training images, each digit is assigned the selected background color with a probability of 0.95, or one of the other colors at random with a probability of 0.05. For the test images, the background color of each digit is assigned from the selected color or other random colors with equal probability of 0.5. The dataset consists of 60,000 training images and 10,000 test images. We configure a class-incremental learning setup, where a total of 10 classes are evenly distributed across 5 tasks, with 2 classes per task. We set the background color as the sensitive attribute and consider two sensitive groups: the origin color and other random colors for each digit.

- **Drug Consumption (DRUG)** (Fehrman et al., 2017): The Drug Consumption dataset contains information about the usage of various drugs by individuals and correlates it with different demographic and personality traits. The dataset includes records for 1,885 respondents, each with 12 attributes including NEO-FFI-R, BIS-11, ImpSS, level of education, age, gender, country of residence, and ethnicity. We split the dataset into the ratio of 70/30 for training and testing. All input attributes are originally categorical, but we quantify them as real values for training. Participants were questioned about their use of 18 drugs, and our task is to predict cannabis usage. The label variable contains six classes: 'Never Used', 'Used over a Decade Ago', 'Used in Last Decade', 'Used in Last Year', 'Used in Last Month', and 'Used in Last Day'. We configure a class-incremental learning setup, where a total of 6 classes are distributed across 3 tasks, with 2 classes per task. We set gender as the sensitive attribute and consider two sensitive groups: male and female.

- **BiasBios** (De-Arteaga et al., 2019): The BiasBios dataset is a benchmark designed to explore and evaluate bias in natural language processing models, particularly in the context of profession classification from bios. The dataset consists of short textual biographies collected from online sources, labeled with one of the 28 profession classes, such as 'professor', 'nurse', or 'software engineer'. The dataset includes gender annotations, which makes it suitable for studying biases related to gender. The dataset contains approximately 350k biographies where 253k are for training and 97k for testing. We configure a class-incremental learning setup using the 25 most-frequent professions, where a total of 25 classes are distributed across 5 tasks, with 5 classes per task. As the number of samples for each class varies significantly, we arrange the classes in descending order based on their size (Chowdhury & Chaturvedi, 2023). We set gender as the sensitive attribute and consider two sensitive groups: male and female.

## B.5. More Details on Models and Hyperparemeters

Continuing from Sec. 4.1, we provide more details on experimental settings. We implement FSW using Python and PyTorch. To solve the fairness-aware optimization problems and find optimal sample weights, we use CPLEX, a high-performance optimization solver developed by IBM that specializes in solving linear programming (LP) problems. For training, we use an SGD optimizer with momentum 0.9 and a batch size of 64 for all experiments. We also set the initial learning rate and the number of epochs for each dataset as follows: For the MNIST, FMNIST, Biased MNIST, and DRUG datasets, we train both our model and baselines with initial learning rates of [0.001, 0.01, 0.1], for 5, 5, 15, and 25 epochs, respectively. For the BiasBios dataset, we use learning rates of [0.00002, 0.0001, 0.001] for 10 epochs and set the maximum token length to 128. For hyperparameters, we perform cross-validation with a grid search for $\alpha \in \{0.0005, 0.001, 0.002, 0.01\}$, $\lambda \in \{0.1, 0.5, 1\}$, and $\tau \in \{1, 2, 5, 10\}$. We employ single-head evaluation where a final layer of the model is shared for all tasks (Farquhar & Gal, 2018; Chaudhry et al., 2018). All evaluations are performed on separate test sets and repeated with five random seeds. We write the average and standard deviation of performance results and run experiments on Intel Xeon Silver 4114 CPUs and NVIDIA RTX A6000 GPUs.

## B.6. More Details on Baselines

Continuing from Sec. 4.1, we provide more details on baselines. In the continual learning literature (Aljundi et al., 2019; Yoon et al., 2022), it is natural for all the baselines to be continual learning methods. We compare our algorithm with the following baselines categorized into four types:

- **Naïve methods**: *Joint Training* assumes access to all the data of previous classes for training and thus has an upper-bound performance; *Fine Tuning* trains a model using only new classes of data without access to previous data and thus has a lower-bound performance.

- **State-of-the-art methods**: *iCaRL* (Rebuffi et al., 2017) performs herding-based buffer selection and representation learning using additional knowledge distillation loss; *WA* (Zhao et al., 2020) is a model rectification method designed to correct the bias in the last fully-connected layer of the model. *WA* uses weight aligning techniques to align the norms of the weight vectors over classes; *CLAD* (Xu et al., 2024) is a representation learning method that disentangles the representation interference between old and new classes.

- **Sample selection methods**: *GSS* (Aljundi et al., 2019) and *OCS* (Yoon et al., 2022) are gradient-based sample selection methods. *GSS* selects a buffer with diverse gradients of samples; *OCS* uses gradient-based similarity, diversity, and affinity scores to rank and select samples for both current and buffer data.

- **Fairness-aware methods**: *FaIRL* (Chowdhury & Chaturvedi, 2023) performs fair representation learning by controlling the rate-distortion function of representations. *FairCL* (Truong et al., 2023) addresses fairness in semantic segmentation tasks arising from the imbalanced class distribution of pixels, but we consider this problem to be unrelated from ours to add the method as a baseline.

## B.7. More Results on Accuracy and Fairness

Continuing from Sec. 4.2, we compare FSW with other baselines with respect to EER, EO, and DP disparity as shown in Tables 5, 6, and 7, respectively. Due to the excessive time (>5 days) required to run *OCS* on BiasBios, we are not able to measure the results. Overall, FSW achieves better accuracy-fairness tradeoff results compared to the baselines for all the datasets. The state-of-the-art method, *iCaRL*, generally achieves high accuracy with low EER disparity results. However, since *iCaRL* uses a nearest-mean-of-exemplars approach for its classification model, the predictions are significantly affected by sensitive attribute values, resulting in large EO and DP disparity. Although *WA* also performs well, the method adjusts the model weights for the current task classes as a whole, which leads to an unfair forgetting of sensitive groups and unstable results. The closest work to FSW is *CLAD*, which disentangles the representations of new classes and a fixed proportion of conflicting old classes to mitigate imbalanced forgetting across classes. However, the proportion of conflicts may vary by task in practice, limiting *CLAD*'s ability to achieve group fairness. While the two sample selection methods *GSS* and *OCS* store diverse and representative samples in the buffer, these methods sometimes result in an imbalance in the number of buffer samples across sensitive groups. The fairness-aware method *FaIRL* leverages an adversarial debiasing framework combined with a rate-distortion function, but the method loses significant accuracy because training the feature encoder and discriminator together is unstable.

Table 5: Accuracy and fairness results on the MNIST and FMNIST datasets with respect to EER disparity, where the class is the sensitive attribute. We compare FSW with four types of baselines: naïve (*Joint Training* and *Fine Tuning*), state-of-the-art (*iCaRL*, *WA*, and *CLAD*), sample selection (*GSS* and *OCS*), and fairness-aware (*FaIRL*) methods. We mark the best and second best results with **bold** and underline, respectively.

| Methods | MNIST | | FMNIST | |
|---|---|---|---|---|
| | Acc. | EER Disp. | Acc. | EER Disp. |
| Joint Training | $.989_{\pm.000}$ | $.003_{\pm.000}$ | $.921_{\pm.002}$ | $.024_{\pm.002}$ |
| Fine Tuning | $.455_{\pm.000}$ | $.326_{\pm.000}$ | $.451_{\pm.000}$ | $.325_{\pm.000}$ |
| iCaRL | $.918_{\pm.005}$ | $.048_{\pm.003}$ | $\mathbf{.852}_{\pm\mathbf{.002}}$ | $\underline{.047}_{\pm.001}$ |
| WA | $.911_{\pm.007}$ | $.052_{\pm.006}$ | $.809_{\pm.005}$ | $.088_{\pm.003}$ |
| CLAD | $.835_{\pm.016}$ | $.099_{\pm.016}$ | $.782_{\pm.018}$ | $.118_{\pm.022}$ |
| GSS | $.889_{\pm.010}$ | $.080_{\pm.009}$ | $.732_{\pm.021}$ | $.149_{\pm.019}$ |
| OCS | $\mathbf{.929}_{\pm\mathbf{.002}}$ | $\underline{.040}_{\pm.003}$ | $.799_{\pm.008}$ | $.109_{\pm.007}$ |
| FaIRL | $.558_{\pm.060}$ | $.273_{\pm.018}$ | $.531_{\pm.032}$ | $.289_{\pm.019}$ |
| **FSW** | $\underline{.925}_{\pm.004}$ | $\mathbf{.032}_{\pm\mathbf{.005}}$ | $\underline{.824}_{\pm.006}$ | $\mathbf{.039}_{\pm\mathbf{.006}}$ |

Table 6: Accuracy and fairness results on the Biased MNIST, DRUG, and BiasBios datasets with respect to EO disparity, where background color is the sensitive attribute for Biased MNIST, and gender for DRUG and BiasBios, respectively. Due to the excessive time (>5 days) required to run *OCS* on BiasBios, we are not able to measure the results and mark them as '–'. The other settings are the same as in Table 5.

| Methods | Biased MNIST | | DRUG | | BiasBios | |
|---|---|---|---|---|---|---|
| | Acc. | EO Disp. | Acc. | EO Disp. | Acc. | EO Disp. |
| Joint Training | $.944_{\pm.002}$ | $.108_{\pm.003}$ | $.442_{\pm.015}$ | $.179_{\pm.052}$ | $.823_{\pm.002}$ | $.076_{\pm.001}$ |
| Fine Tuning | $.449_{\pm.001}$ | $.016_{\pm.002}$ | $.357_{\pm.009}$ | $.125_{\pm.034}$ | $.420_{\pm.001}$ | $.028_{\pm.002}$ |
| iCaRL | $.802_{\pm.008}$ | $.365_{\pm.021}$ | $\mathbf{.444}_{\pm\mathbf{.025}}$ | $.190_{\pm.017}$ | $\mathbf{.829}_{\pm\mathbf{.002}}$ | $.084_{\pm.003}$ |
| WA | $\mathbf{.916}_{\pm\mathbf{.002}}$ | $.140_{\pm.004}$ | $.408_{\pm.022}$ | $.134_{\pm.029}$ | $.796_{\pm.003}$ | $.076_{\pm.001}$ |
| CLAD | $.871_{\pm.012}$ | $.198_{\pm.022}$ | $.410_{\pm.026}$ | $.114_{\pm.043}$ | $.799_{\pm.003}$ | $.074_{\pm.002}$ |
| GSS | $.809_{\pm.005}$ | $.325_{\pm.017}$ | $\underline{.426}_{\pm.010}$ | $.167_{\pm.038}$ | $\underline{.808}_{\pm.003}$ | $.081_{\pm.002}$ |
| OCS | $.824_{\pm.007}$ | $.331_{\pm.013}$ | $.406_{\pm.024}$ | $.142_{\pm.030}$ | – | – |
| FaIRL | $.411_{\pm.012}$ | $\mathbf{.118}_{\pm\mathbf{.011}}$ | $.354_{\pm.011}$ | $\mathbf{.060}_{\pm\mathbf{.021}}$ | $.400_{\pm.060}$ | $\mathbf{.055}_{\pm\mathbf{.020}}$ |
| **FSW** | $\underline{.909}_{\pm.004}$ | $\underline{.119}_{\pm.007}$ | $.406_{\pm.014}$ | $\underline{.077}_{\pm.010}$ | $\underline{.808}_{\pm.002}$ | $\underline{.072}_{\pm.001}$ |

Table 7: Accuracy and fairness results on the Biased MNIST, DRUG, and BiasBios datasets with respect to DP disparity. The other settings are the same as in Table 6.

| Methods | Biased MNIST | | DRUG | | BiasBios | |
|---|---|---|---|---|---|---|
| | Acc. | DP Disp. | Acc. | DP Disp. | Acc. | DP Disp. |
| Joint Training | $.944_{\pm.002}$ | $.006_{\pm.001}$ | $.442_{\pm.015}$ | $.090_{\pm.020}$ | $.823_{\pm.002}$ | $.021_{\pm.000}$ |
| Fine Tuning | $.449_{\pm.001}$ | $.017_{\pm.008}$ | $.357_{\pm.009}$ | $.102_{\pm.013}$ | $.420_{\pm.001}$ | $.028_{\pm.002}$ |
| iCaRL | $.802_{\pm.008}$ | $.015_{\pm.001}$ | $\mathbf{.444}_{\pm\mathbf{.025}}$ | $.093_{\pm.009}$ | $\mathbf{.829}_{\pm\mathbf{.002}}$ | $\underline{.022}_{\pm.000}$ |
| WA | $\mathbf{.916}_{\pm\mathbf{.002}}$ | $\underline{.009}_{\pm.001}$ | $.408_{\pm.022}$ | $.067_{\pm.013}$ | $.796_{\pm.003}$ | $\underline{.022}_{\pm.000}$ |
| CLAD | $.871_{\pm.012}$ | $\underline{.013}_{\pm.001}$ | $\underline{.410}_{\pm.026}$ | $.069_{\pm.019}$ | $.799_{\pm.003}$ | $\underline{.022}_{\pm.000}$ |
| GSS | $.809_{\pm.005}$ | $.039_{\pm.003}$ | $.392_{\pm.022}$ | $.065_{\pm.015}$ | $.808_{\pm.003}$ | $.023_{\pm.000}$ |
| OCS | $.824_{\pm.007}$ | $.035_{\pm.003}$ | $.393_{\pm.017}$ | $.053_{\pm.012}$ | – | – |
| FaIRL | $.411_{\pm.012}$ | $.026_{\pm.008}$ | $.354_{\pm.011}$ | $\mathbf{.040}_{\pm\mathbf{.008}}$ | $.400_{\pm.060}$ | $\mathbf{.015}_{\pm\mathbf{.002}}$ |
| **FSW** | $\underline{.904}_{\pm.004}$ | $\mathbf{.008}_{\pm\mathbf{.001}}$ | $.405_{\pm.013}$ | $\underline{.043}_{\pm.004}$ | $\underline{.809}_{\pm.003}$ | $\underline{.022}_{\pm.000}$ |

## B.8. More Results on Sequential Accuracy and Fairness

Continuing from Sec. 4.2, we present the sequential performance results for each task as shown in Fig. 9–Fig. 16. Due to the excessive time required to run *OCS* on BiasBios, we are not able to measure the results.

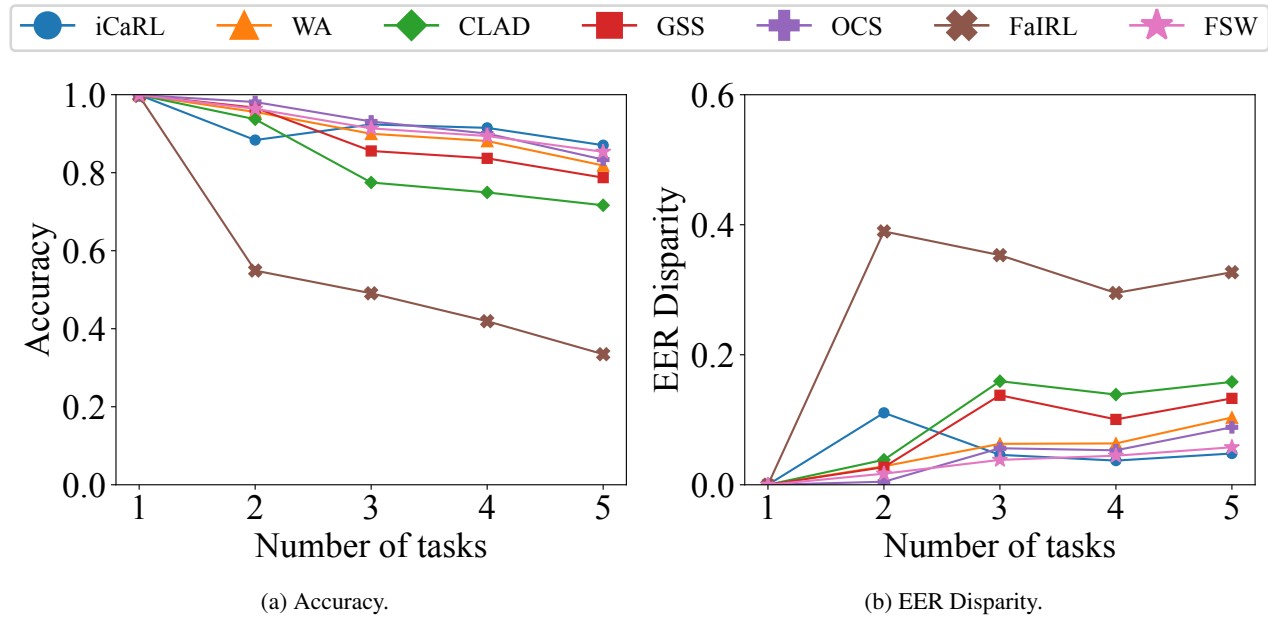

(a) Accuracy.  (b) EER Disparity.

Figure 9: Sequential accuracy and fairness (EER) results on the MNIST dataset.

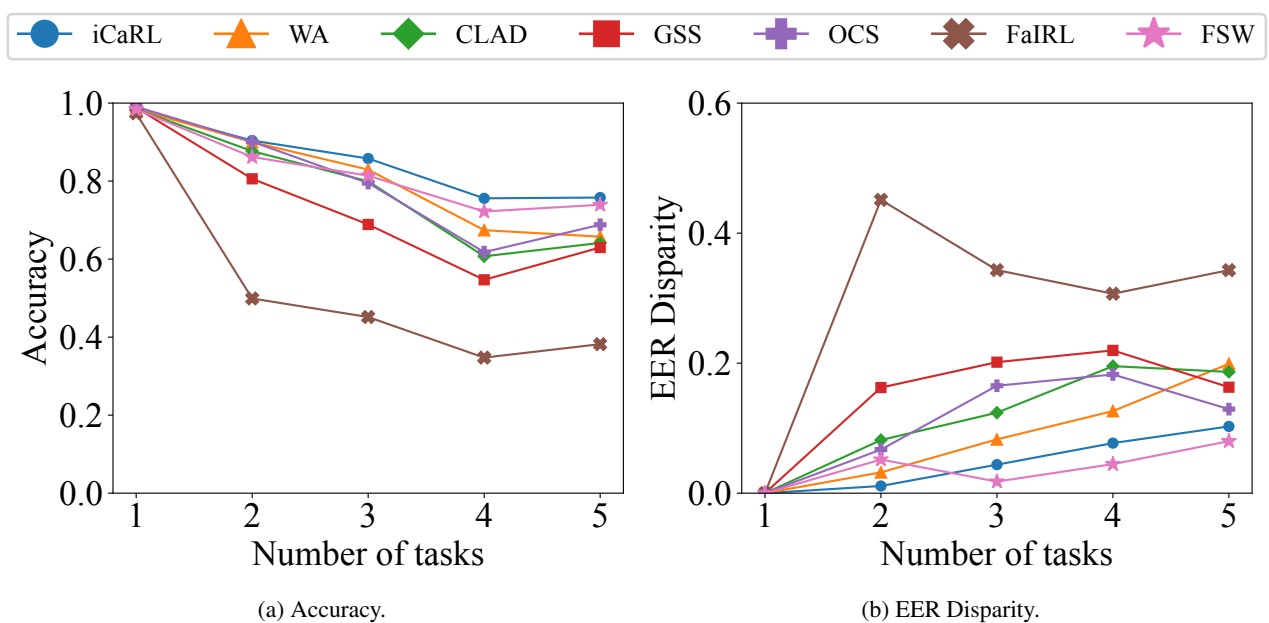

(a) Accuracy.  (b) EER Disparity.

Figure 10: Sequential accuracy and fairness (EER) results on the FMNIST dataset.

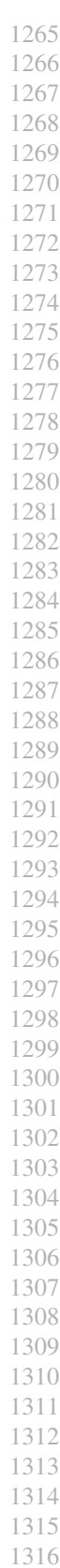

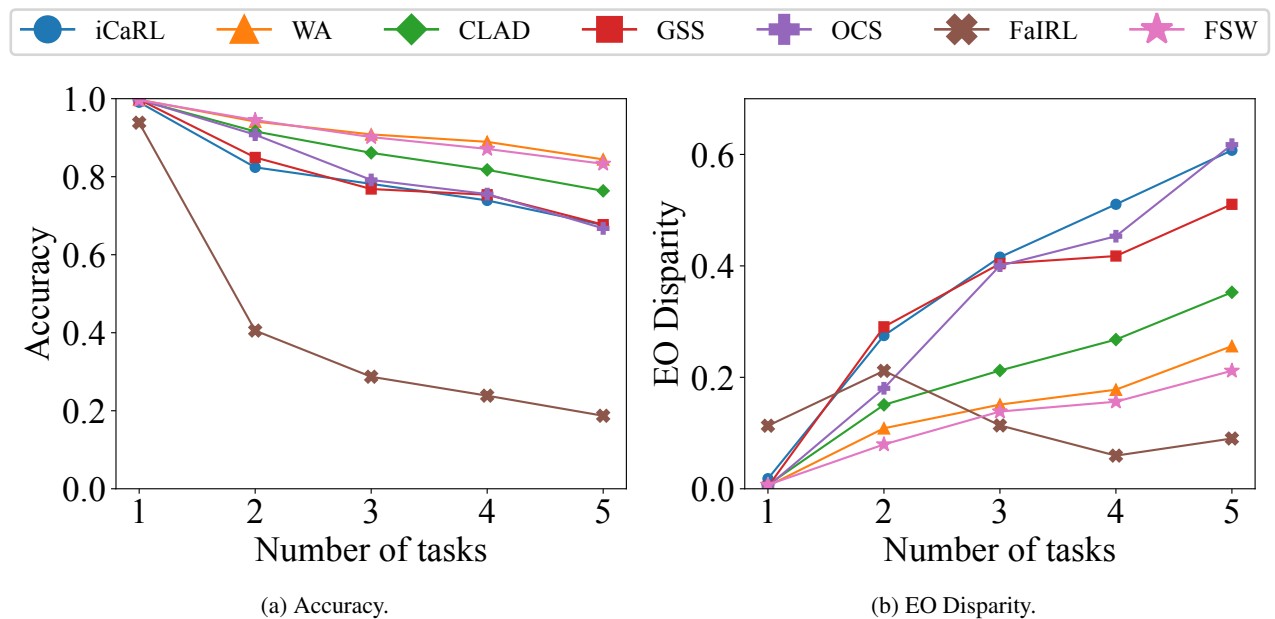

(a) Accuracy.

(b) EO Disparity.

Figure 11: Sequential accuracy and fairness (EO) results on the Biased MNIST dataset.

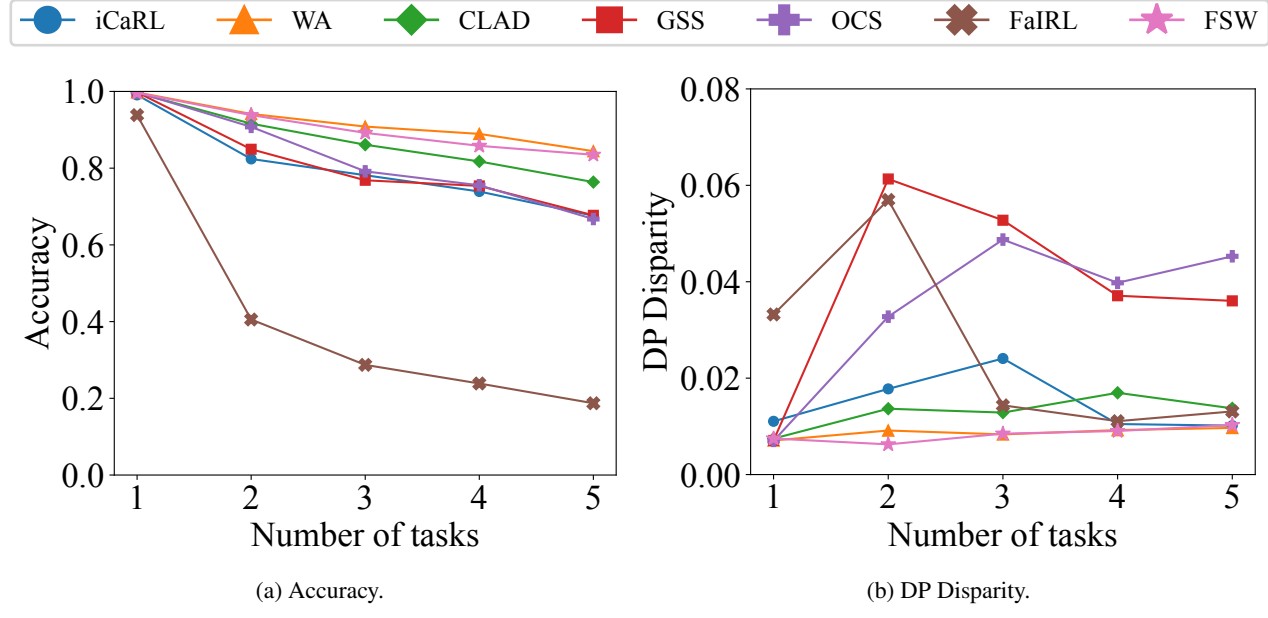

(a) Accuracy.

(b) DP Disparity.

Figure 12: Sequential accuracy and fairness (DP) results on the Biased MNIST dataset.

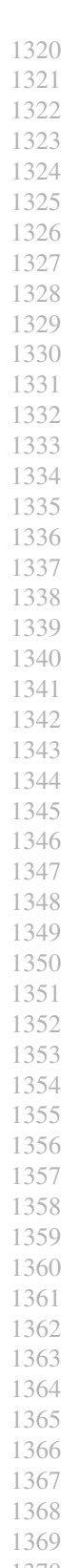

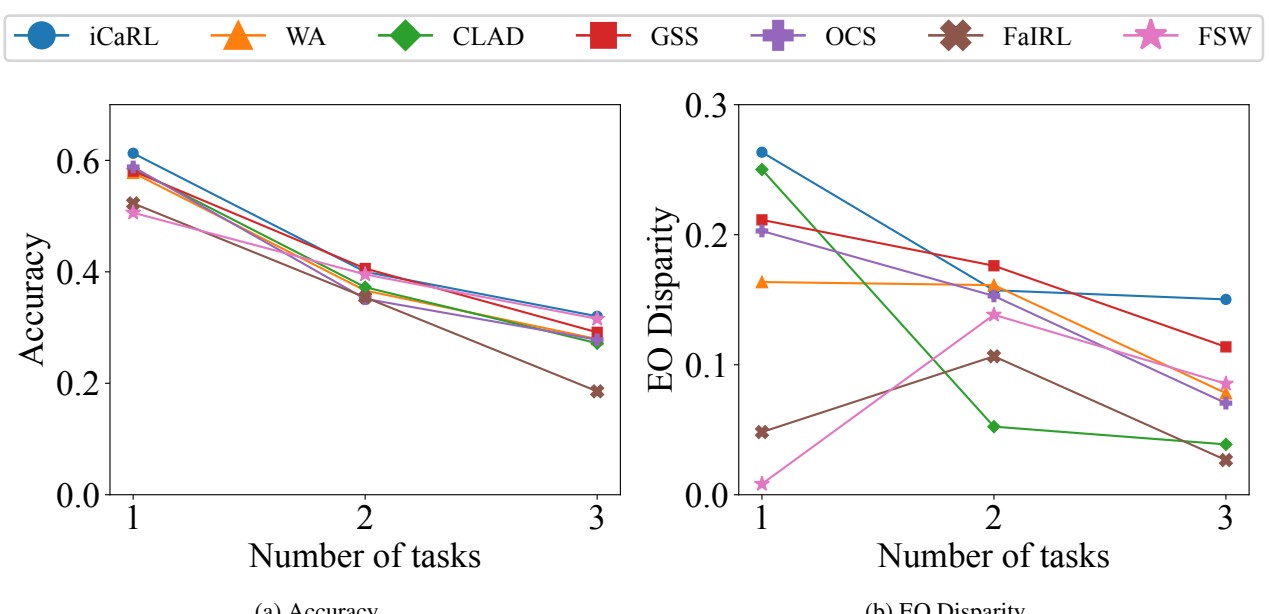

(a) Accuracy.

(b) EO Disparity.

Figure 13: Sequential accuracy and fairness (EO) results on the DRUG dataset.

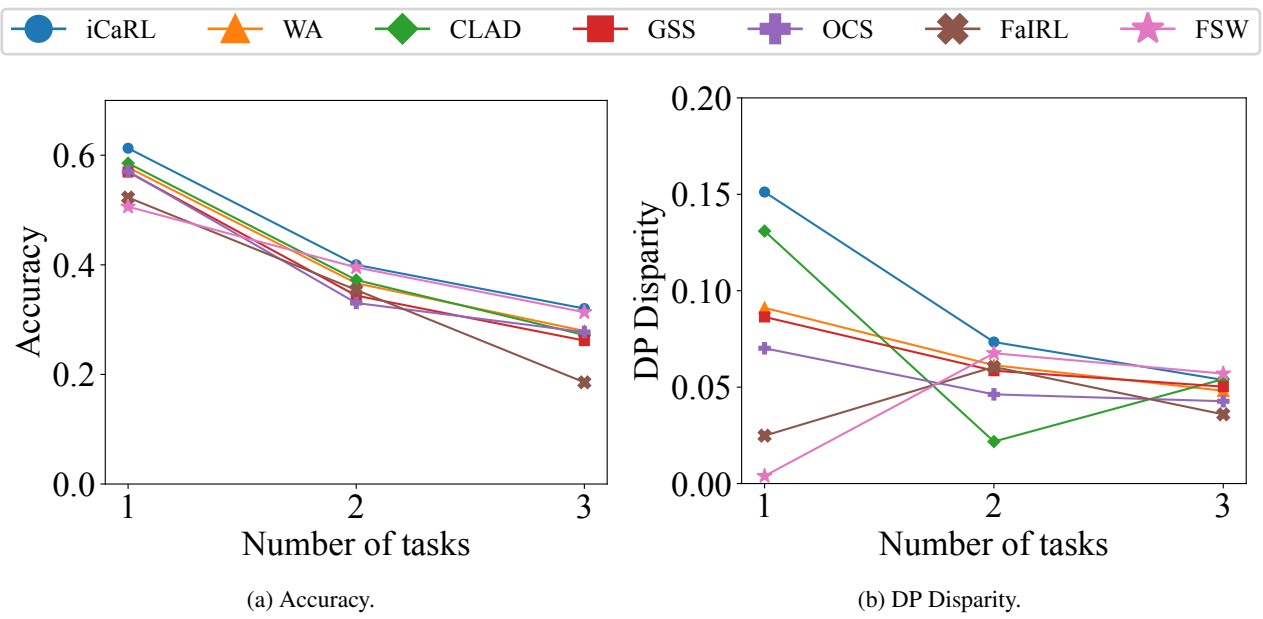

(a) Accuracy.

(b) DP Disparity.

Figure 14: Sequential accuracy and fairness (DP) results on the DRUG dataset.

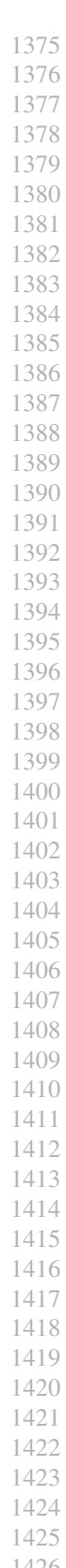

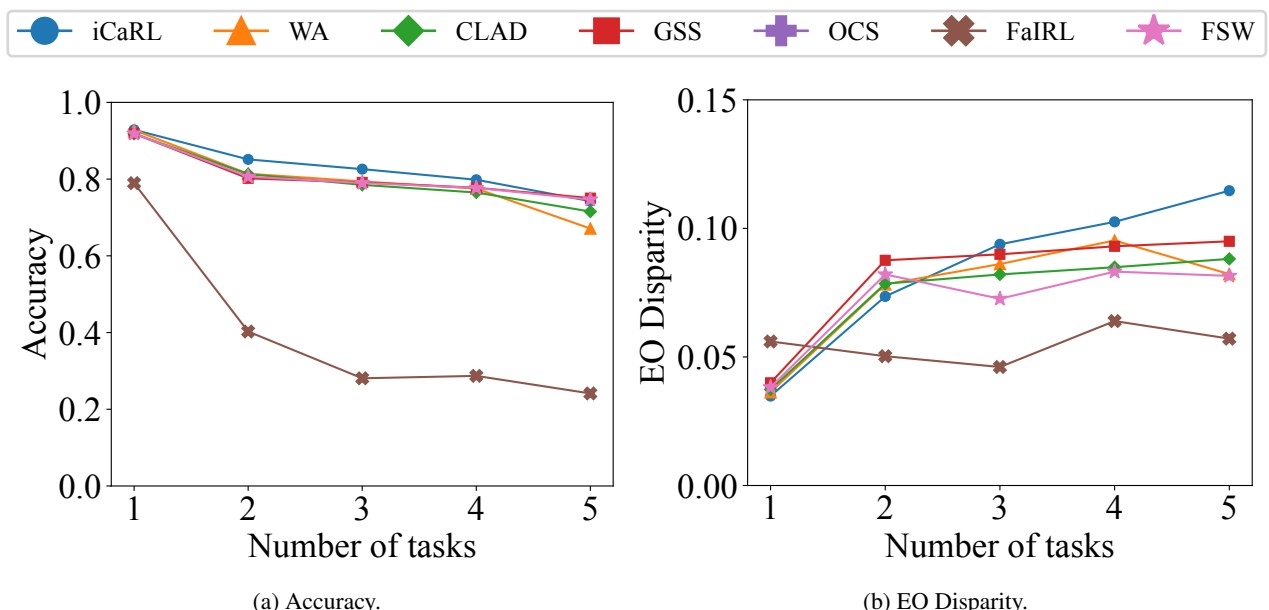

(a) Accuracy.

(b) EO Disparity.

Figure 15: Sequential accuracy and fairness (EO) results on the BiasBios dataset.

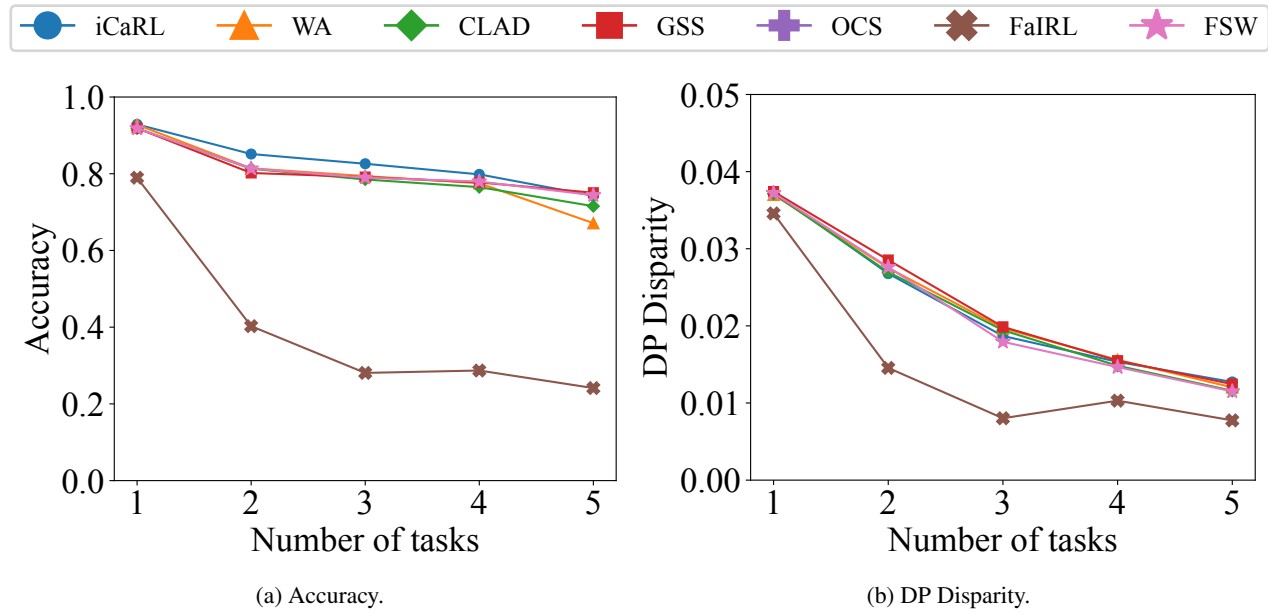

(a) Accuracy.

(b) DP Disparity.

Figure 16: Sequential accuracy and fairness (DP) results on the BiasBios dataset.

## B.9. More Results on Tradeoff between Accuracy and Fairness

Continuing from Sec. 4.2, we evaluate the tradeoff between accuracy and fairness of FSW with other baselines as shown in Fig. 17–Fig. 20. FSW in the figures represents the result for different values of $\lambda$, a hyperparameter that balances fairness and accuracy. Since other baselines do not have a balancing parameter, we select Pareto-optimal points from all search spaces, where a Pareto-optimal point is defined as a point for which there does not exist another point with both higher accuracy and lower fairness disparity. The figures show FSW positioned in the lower right corner of the graph, indicating better accuracy-fairness tradeoff results compared to other baselines. Due to the excessive time required to run *OCS* on BiasBios, we are not able to measure the results.

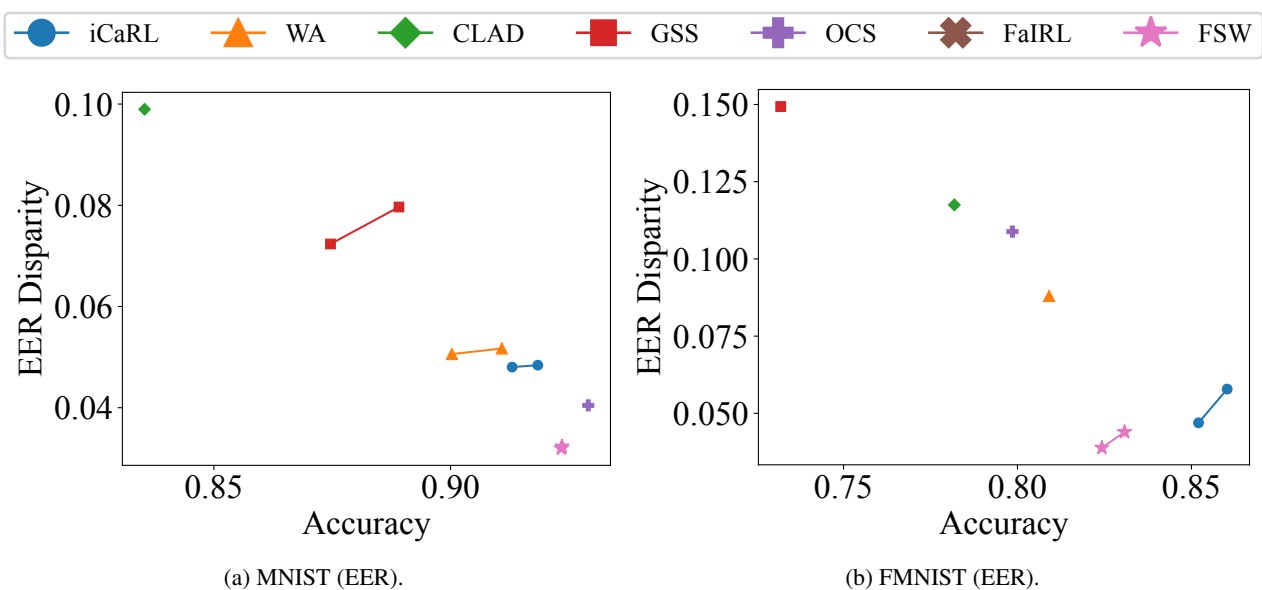

(a) MNIST (EER).

(b) FMNIST (EER).

Figure 17: Tradeoff results between accuracy and fairness (EER) on the MNIST and FMNIST datasets.

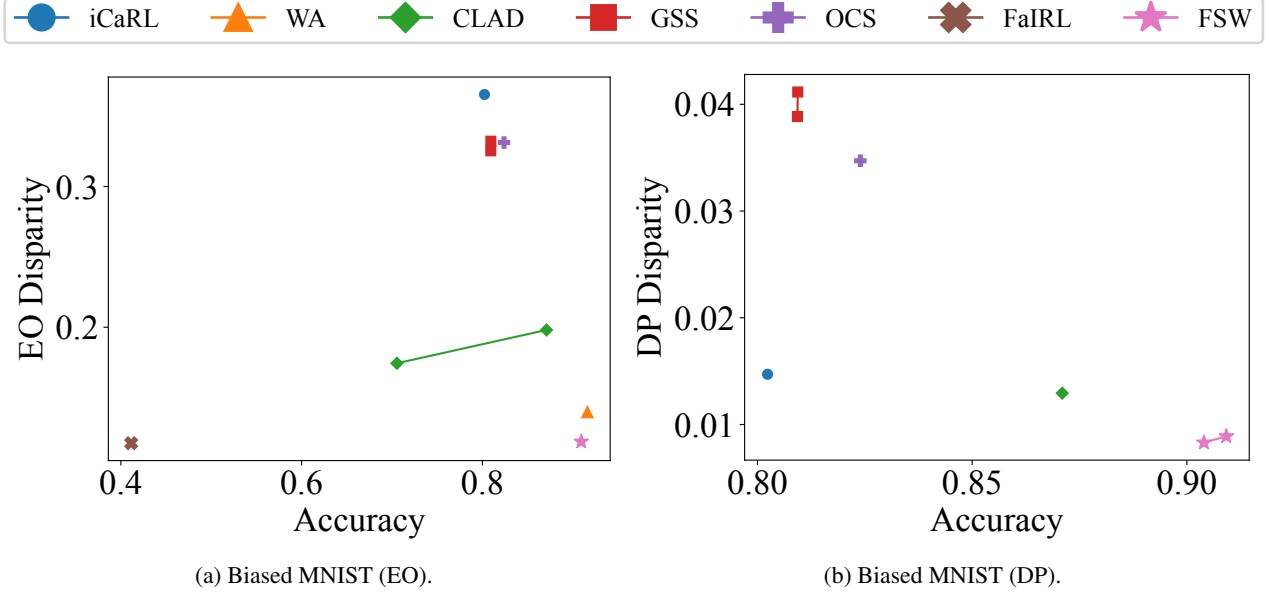

(a) Biased MNIST (EO).

(b) Biased MNIST (DP).

Figure 18: Tradeoff results between accuracy and fairness (EO and DP) on the Biased MNIST dataset.

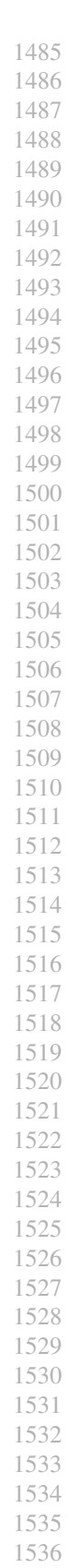

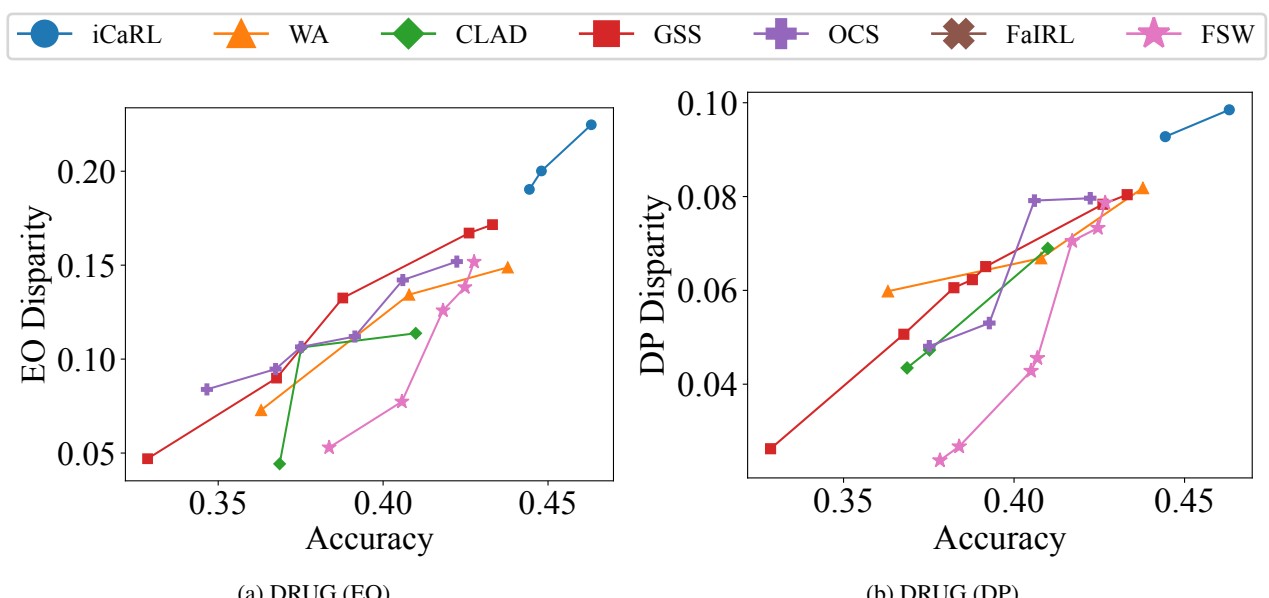

(a) DRUG (EO).

(b) DRUG (DP).

Figure 19: Tradeoff results between accuracy and fairness (EO and DP) on the DRUG dataset.

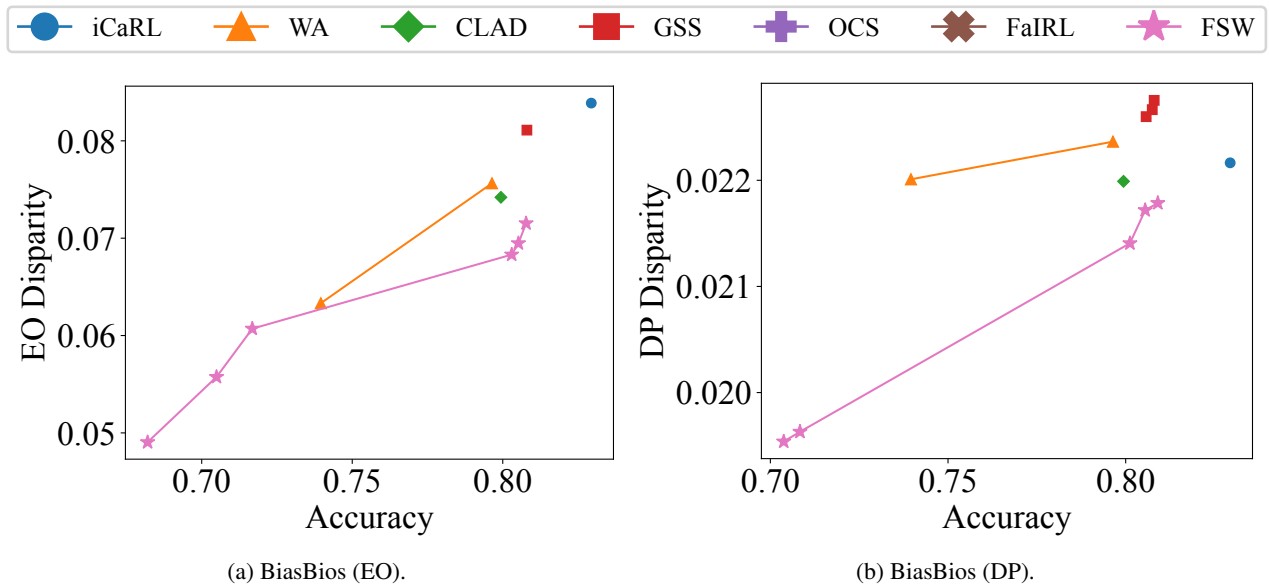

(a) BiasBios (EO).

(b) BiasBios (DP).

Figure 20: Tradeoff results between accuracy and fairness (EO and DP) on the BiasBios dataset.

## B.10. More Results of FSW when Varying the Buffer Size

Continuing from Sec. 4.2, we have additional experimental results of FSW on the MNIST and Biased MNIST datasets when varying the buffer size to 16, 32, 64, and 128 per sensitive group as shown in Fig. 21. As the buffer size increases, both accuracy and fairness performances improve. In addition, we compute the number of current task data assigned with non-zero weights as shown in Fig. 22, and there is no clear relationship between the buffer size and weights.

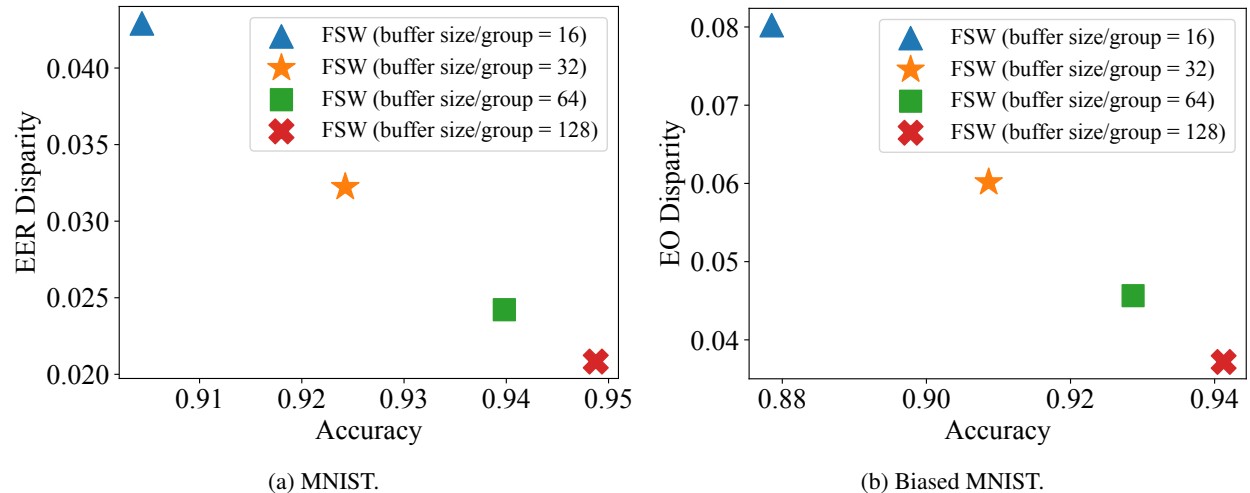

(a) MNIST.  (b) Biased MNIST.

Figure 21: Accuracy and fairness results of FSW when varying the buffer size on the MNIST and Biased MNIST datasets.

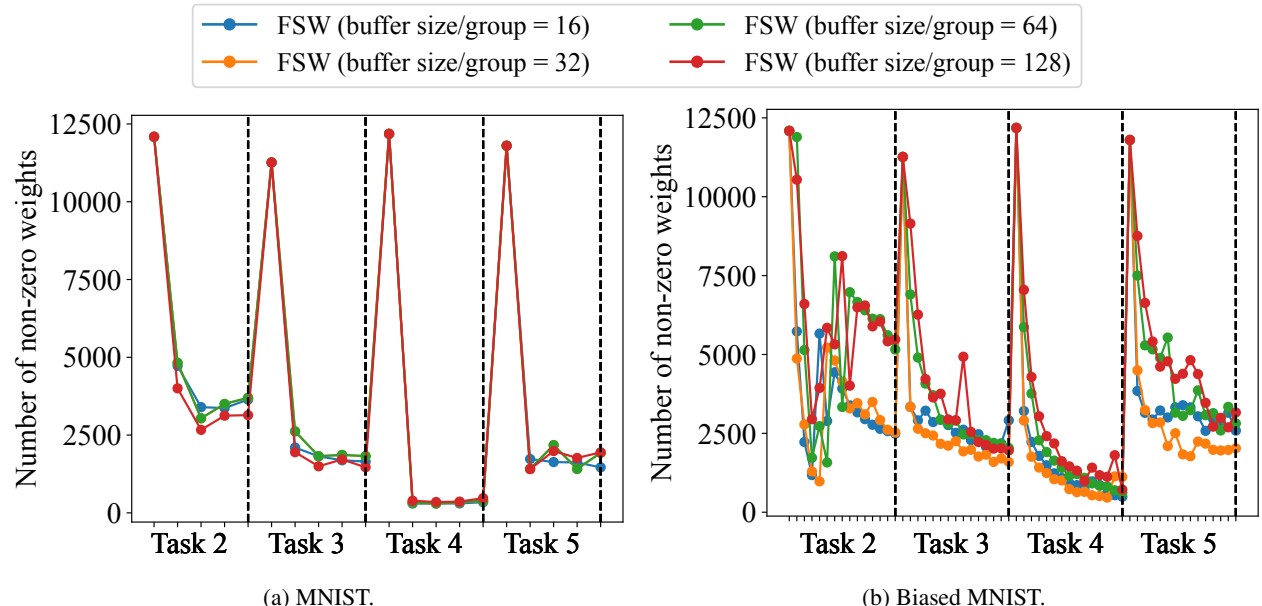

(a) MNIST.  (b) Biased MNIST.

Figure 22: Number of current task data assigned with non-zero weights when varying the buffer size on the MNIST and Biased MNIST datasets.

## B.11. More Results on Sample Weighting Analysis

Continuing from Sec. 4.3, we show more results from the sample weighting analysis for all sequential tasks of each dataset, as shown in the figures below (Fig. 23–Fig. 30). We compute the number of samples for weights in sensitive groups including classes. For each task, we show the average weight distribution over all epochs, as sample weights may change during each epoch of training. Since FSW is not applied to the first task, where the model is trained with only the current task data, we present the results starting from the second task.

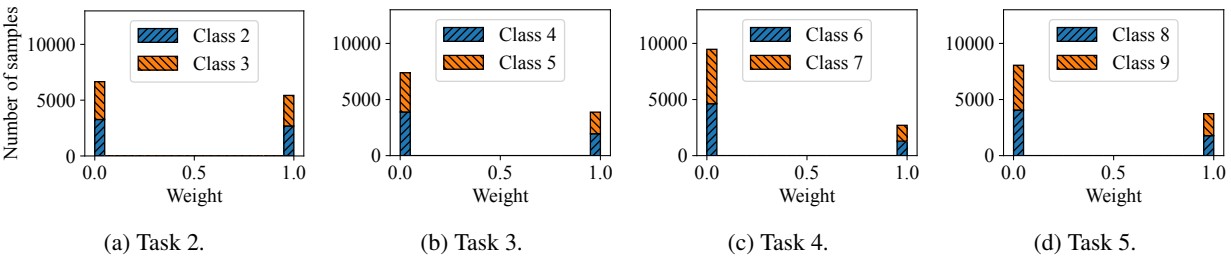

|  |  |  |  |
|---|---|---|---|
| (a) Task 2. | (b) Task 3. | (c) Task 4. | (d) Task 5. |

Figure 23: Distribution of sample weights for EER in sequential tasks of the MNIST dataset.

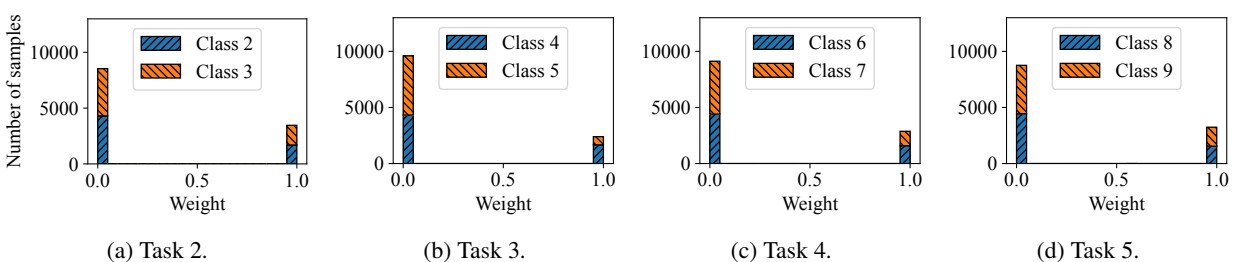

|  |  |  |  |
|---|---|---|---|
| (a) Task 2. | (b) Task 3. | (c) Task 4. | (d) Task 5. |

Figure 24: Distribution of sample weights for EER in sequential tasks of the FMNIST dataset.

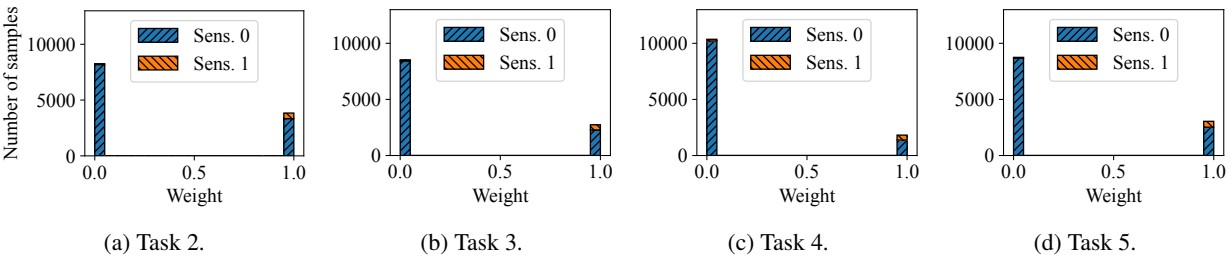

|  |  |  |  |
|---|---|---|---|
| (a) Task 2. | (b) Task 3. | (c) Task 4. | (d) Task 5. |

Figure 25: Distribution of sample weights for EO in sequential tasks of the Biased MNIST dataset.

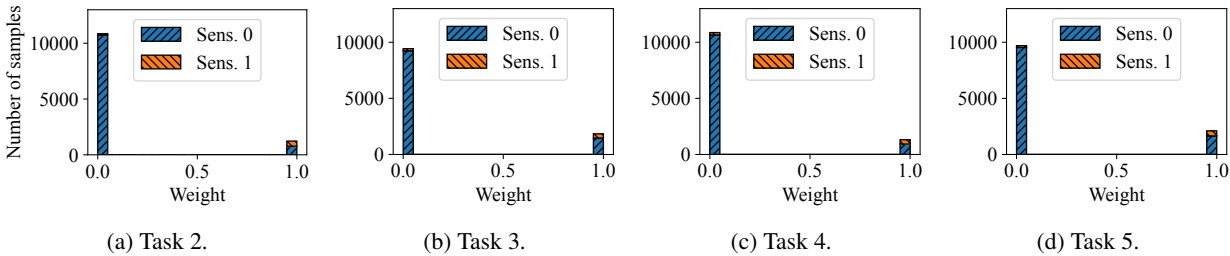

|  |  |  |  |
|---|---|---|---|
| (a) Task 2. | (b) Task 3. | (c) Task 4. | (d) Task 5. |

Figure 26: Distribution of sample weights for DP in sequential tasks of the Biased MNIST dataset.

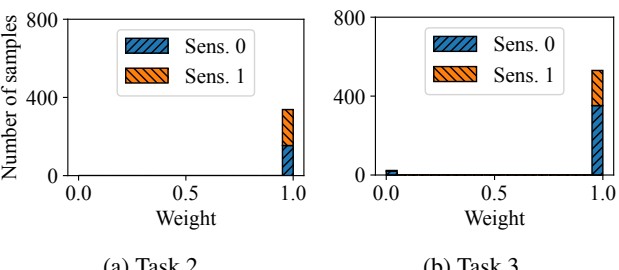

(a) Task 2.        (b) Task 3.

Figure 27: Distribution of sample weights for EO in sequential tasks of the DRUG dataset.

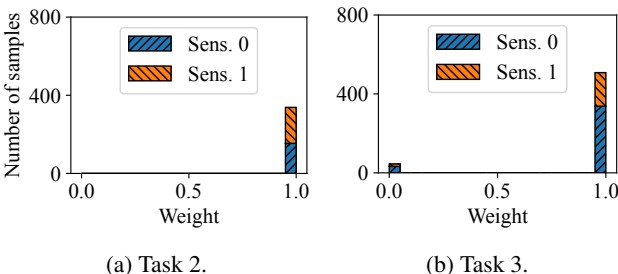

(a) Task 2.        (b) Task 3.

Figure 28: Distribution of sample weights for DP in sequential tasks of the DRUG dataset.

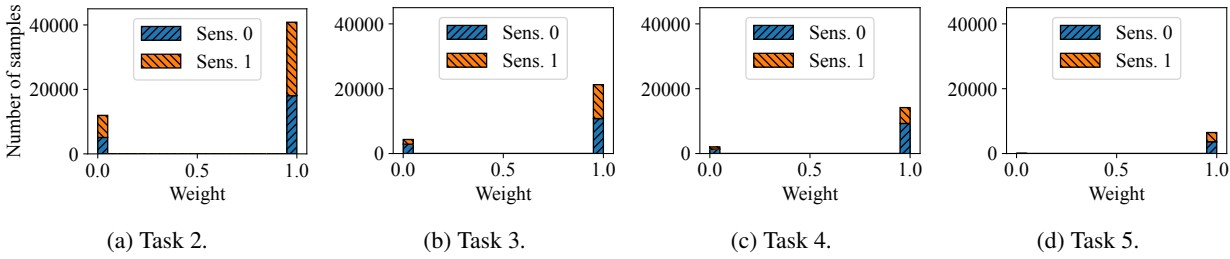

(a) Task 2.    (b) Task 3.    (c) Task 4.    (d) Task 5.

Figure 29: Distribution of sample weights for EO in sequential tasks of the BiasBios dataset.

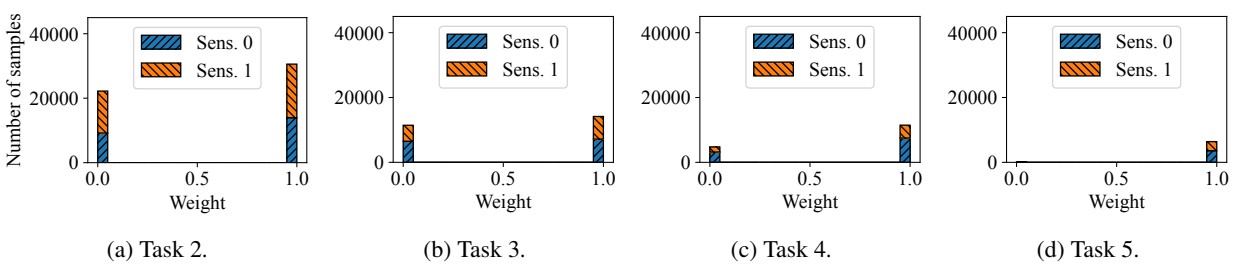

(a) Task 2.    (b) Task 3.    (c) Task 4.    (d) Task 5.

Figure 30: Distribution of sample weights for DP in sequential tasks of the BiasBios dataset.

## B.12. Binarity of the Sample Weight

Continuing from Sec. 4.3, the acquired sample weights are mostly close to 0 or 1, which are extreme values. However, there are also some values that do not lie at these boundaries (0 or 1). The average number of binary (0 or 1) and non-binary (not 0 or 1) samples weights are shown in Table 8. For the distribution of binary values, please refer to Fig. 23–Fig. 30 in Sec. B.11. We also emphasize that if we limit the solution of the optimization problem to binary, which is equivalent to sample selection, the problem would transform into a mixed-integer linear programming problem, which is NP-hard and cannot be solved efficiently.

Table 8: Average counts of binary (0 or 1) and non-binary (not 0 or 1) sample weights for each optimization task. Since we take averages of different tasks excluding the first task, the sum of (# Binary) and (# Non-binary) are not necessarily integers.

| Dataset (Metric) | MNIST (EER) | FMNIST (EER) | Biased MNIST (EO) | Biased MNIST (DP) |
|---|---|---|---|---|
| # Binary | 11830.8 | 11997.4 | 11831.9 | 11831.6 |
| # Non-binary | 3.0 | 2.6 | 1.9 | 2.1 |

| Dataset (Metric) | Drug (EO) | Drug (DP) | BiasBios (EO) | BiasBios (DP) |
|---|---|---|---|---|
| # Binary | 446.5 | 446.1 | 25274.7 | 25273.2 |
| # Non-binary | 0.0 | 0.4 | 2.0 | 3.5 |

## B.13. More Results on Ablation Study

Continuing from Sec. 4.4, we present additional results of the ablation study to demonstrate the contribution of FSW to the overall accuracy and fairness performance. The results are shown in Tables 9, 10, and 11.

Table 9: Accuracy and fairness results on the MNIST and FMNIST datasets, with respect to EER disparity with or without FSW.

| Methods | MNIST | | FMNIST | |
|---|---|---|---|---|
| | Acc. | EER Disp. | Acc. | EER Disp. |
| W/o FSW | $.912_{\pm.004}$ | $.051_{\pm.005}$ | $.810_{\pm.004}$ | $.092_{\pm.003}$ |
| **FSW** | $\mathbf{.925}_{\pm\mathbf{.004}}$ | $\mathbf{.032}_{\pm\mathbf{.005}}$ | $\mathbf{.824}_{\pm\mathbf{.006}}$ | $\mathbf{.039}_{\pm\mathbf{.006}}$ |

Table 10: Accuracy and fairness results on the Biased MNIST, DRUG, and BiasBios datasets with respect to EO disparity with or without FSW.

| Methods | Biased MNIST | | DRUG | | BiasBios | |
|---|---|---|---|---|---|---|
| | Acc. | EO Disp. | Acc. | EO Disp. | Acc. | EO Disp. |
| W/o FSW | $\mathbf{.910}_{\pm\mathbf{.003}}$ | $.126_{\pm.005}$ | $.402_{\pm.010}$ | $.080_{\pm.005}$ | $.806_{\pm.003}$ | $.073_{\pm.002}$ |
| **FSW** | $.909_{\pm.004}$ | $\mathbf{.119}_{\pm\mathbf{.007}}$ | $\mathbf{.406}_{\pm\mathbf{.014}}$ | $\mathbf{.077}_{\pm\mathbf{.010}}$ | $\mathbf{.808}_{\pm\mathbf{.002}}$ | $\mathbf{.072}_{\pm\mathbf{.001}}$ |

Table 11: Accuracy and fairness results on the Biased MNIST, DRUG, and BiasBios datasets with respect to DP disparity with or without FSW.

| Methods | Biased MNIST | | DRUG | | BiasBios | |
|---|---|---|---|---|---|---|
| | Acc. | DP Disp. | Acc. | DP Disp. | Acc. | DP Disp. |
| W/o FSW | $\mathbf{.910}_{\pm\mathbf{.003}}$ | $.009_{\pm.001}$ | $.402_{\pm.010}$ | $.044_{\pm.004}$ | $.805_{\pm.002}$ | $\mathbf{.022}_{\pm\mathbf{.000}}$ |
| **FSW** | $.904_{\pm.004}$ | $\mathbf{.008}_{\pm\mathbf{.001}}$ | $\mathbf{.405}_{\pm\mathbf{.013}}$ | $\mathbf{.043}_{\pm\mathbf{.004}}$ | $\mathbf{.809}_{\pm\mathbf{.003}}$ | $\mathbf{.022}_{\pm\mathbf{.000}}$ |

## B.14. More Results on Integrating FSW with a Fair Post-processing Method

Continuing from Sec. 4.5, we provide additional results on integrating continual learning methods with a fair post-processing method ($\epsilon$-fair) as shown in Table 12. Since $\epsilon$-fair only supports DP, we only show DP results. We mark the best and second-best results with bold and underline, respectively, regardless of the application of post-processing.

Table 12: Accuracy and fairness (DP disparity) results when combining fair post-processing techniques ($\epsilon$-fair) with continual learning methods (*iCaRL*, *WA*, *CLAD*, *GSS*, *OCS*, and FSW). Due to the excessive time ($>5$ days) required to run *OCS* on BiasBios, we are not able to measure the results and mark them as '–'.

| Methods | Biased MNIST | | DRUG | | BiasBios | |
|---|---|---|---|---|---|---|
| | Acc. | DP Disp. | Acc. | DP Disp. | Acc. | DP Disp. |
| iCaRL | $.802_{\pm.008}$ | $.015_{\pm.001}$ | $\mathbf{.444}_{\pm\mathbf{.025}}$ | $.093_{\pm.009}$ | $\mathbf{.829}_{\pm\mathbf{.002}}$ | $.022_{\pm.000}$ |
| WA | $.916_{\pm.002}$ | $.009_{\pm.001}$ | $.408_{\pm.022}$ | $.067_{\pm.013}$ | $.796_{\pm.003}$ | $.022_{\pm.000}$ |
| CLAD | $.871_{\pm.012}$ | $.013_{\pm.001}$ | $.410_{\pm.026}$ | $.069_{\pm.019}$ | $.799_{\pm.003}$ | $.022_{\pm.000}$ |
| GSS | $.809_{\pm.005}$ | $.039_{\pm.003}$ | $.392_{\pm.022}$ | $.065_{\pm.015}$ | $.808_{\pm.003}$ | $.023_{\pm.000}$ |
| OCS | $.824_{\pm.007}$ | $.035_{\pm.003}$ | $.393_{\pm.017}$ | $.053_{\pm.012}$ | – | – |
| **FSW** | $.904_{\pm.004}$ | $.008_{\pm.001}$ | $.405_{\pm.013}$ | $.043_{\pm.004}$ | $\underline{.809}_{\pm.003}$ | $.022_{\pm.000}$ |
| iCaRL – $\epsilon$-fair | $.944_{\pm.008}$ | $.006_{\pm.002}$ | $\underline{.427}_{\pm.018}$ | $.026_{\pm.004}$ | $.753_{\pm.002}$ | $.017_{\pm.000}$ |
| WA – $\epsilon$-fair | $\mathbf{.953}_{\pm\mathbf{.003}}$ | $\underline{.006}_{\pm.002}$ | $.404_{\pm.021}$ | $.044_{\pm.020}$ | $.708_{\pm.003}$ | $\mathbf{.016}_{\pm\mathbf{.000}}$ |
| CLAD – $\epsilon$-fair | $.924_{\pm.012}$ | $\underline{.006}_{\pm.002}$ | $.406_{\pm.027}$ | $.030_{\pm.010}$ | $.716_{\pm.004}$ | $\mathbf{.016}_{\pm\mathbf{.001}}$ |
| GSS – $\epsilon$-fair | $.938_{\pm.006}$ | $\underline{.006}_{\pm.002}$ | $.382_{\pm.014}$ | $.035_{\pm.017}$ | $.717_{\pm.005}$ | $\mathbf{.016}_{\pm\mathbf{.000}}$ |
| OCS – $\epsilon$-fair | $\underline{.952}_{\pm.003}$ | $.032_{\pm.004}$ | $.384_{\pm.009}$ | $.051_{\pm.002}$ | – | – |
| **FSW – $\epsilon$-fair** | $.906_{\pm.006}$ | $\mathbf{.005}_{\pm\mathbf{.001}}$ | $.405_{\pm.013}$ | $\mathbf{.021}_{\pm\mathbf{.004}}$ | $.723_{\pm.004}$ | $\mathbf{.016}_{\pm\mathbf{.000}}$ |

## B.15. Alternative Loss Function for Group Fairness Metrics

Continuing from Sec. 3.1, we use cross-entropy loss disparity to approximate group fairness metrics such as EER, EO, and DP disparity. Both theoretical and empirical results show that the cross-entropy loss disparity can effectively approximate these group fairness metrics, as discussed in Sec. A.2. However, the cross-entropy loss disparity is not the only possible type of loss for approximating the group fairness metrics; the disparity of other loss functions may yield better performance. Our method can be applied regardless of the loss definition if (1) the loss update process can be linearly approximated (as in Sec. A.1) and (2) the loss disparity promotes fairness (as in Sec. A.2).

To verify if FSW can also be effective with different loss function designs, we conduct simple experiments using hinge loss (i.e., $\sum_{j \neq y_i} \max(0, s_j - s_{y_i} + 1)$ where $y_i$ is the true integer label, and $s_j$ is the softmax output for label $j$) to approximate group fairness metrics in FSW. The results are shown in Tables 13, 14, and 15. Overall, both methods show comparable accuracy-fairness results, suggesting that FSW performs well regardless of the type of loss function used to approximate group fairness metrics. Here, we would like to note that the cross-entropy loss disparity is widely used and empirically verified as a reasonable proxy for capturing group fairness metrics (Shen et al., 2022; Roh et al., 2021; 2023; Gupta et al., 2024), which is why we use it, although we could also use other losses.

Table 13: Accuracy and fairness results on the MNIST and FMNIST datasets with respect to EER disparity. "FSW (hinge)" uses hinge loss, while "FSW" uses cross-entropy loss to approximate the group fairness metric.

| Methods | MNIST | | FMNIST | |
|---|---|---|---|---|
| | Acc. | EER Disp. | Acc. | EER Disp. |
| FSW | $\mathbf{.925}_{\pm\mathbf{.004}}$ | $.032_{\pm.005}$ | $.824_{\pm.006}$ | $\mathbf{.039}_{\pm\mathbf{.006}}$ |
| FSW (hinge) | $\mathbf{.925}_{\pm\mathbf{.003}}$ | $\mathbf{.030}_{\pm\mathbf{.006}}$ | $.825_{\pm.006}$ | $\mathbf{.039}_{\pm\mathbf{.005}}$ |

Table 14: Accuracy and fairness results on the Biased MNIST, DRUG, and BiasBios datasets with respect to EO disparity. The other settings are the same as in Table 13.

| Methods | Biased MNIST | | DRUG | | BiasBios | |
|---|---|---|---|---|---|---|
| | Acc. | EO Disp. | Acc. | EO Disp. | Acc. | EO Disp. |
| FSW | $\mathbf{.909}_{\pm\mathbf{.004}}$ | $\mathbf{.119}_{\pm\mathbf{.007}}$ | $\mathbf{.406}_{\pm\mathbf{.014}}$ | $\mathbf{.077}_{\pm\mathbf{.010}}$ | $\mathbf{.808}_{\pm\mathbf{.002}}$ | $.072_{\pm.001}$ |
| FSW (hinge) | $\mathbf{.909}_{\pm\mathbf{.004}}$ | $\mathbf{.119}_{\pm\mathbf{.006}}$ | $\mathbf{.406}_{\pm\mathbf{.014}}$ | $\mathbf{.077}_{\pm\mathbf{.010}}$ | $.807_{\pm.002}$ | $\mathbf{.071}_{\pm\mathbf{.002}}$ |

Table 15: Accuracy and fairness results on the Biased MNIST, DRUG, and BiasBios datasets with respect to DP disparity. The other settings are the same as in Table 13.

| Methods | Biased MNIST | | DRUG | | BiasBios | |
|---|---|---|---|---|---|---|
| | Acc. | DP Disp. | Acc. | DP Disp. | Acc. | DP Disp. |
| FSW | $.904_{\pm.004}$ | $.008_{\pm.001}$ | $.405_{\pm.013}$ | $.043_{\pm.004}$ | $.809_{\pm.003}$ | $.022_{\pm.000}$ |
| FSW (hinge) | $.904_{\pm.004}$ | $.008_{\pm.001}$ | $.405_{\pm.013}$ | $.043_{\pm.004}$ | $.807_{\pm.006}$ | $.022_{\pm.000}$ |

## C. Appendix – More Related Work

Continuing from Sec. 2, we discuss more related work.

Class-incremental learning is a challenging type of continual learning where a model continuously learns new tasks, each composed of new disjoint classes, and the goal is to minimize catastrophic forgetting (Mai et al., 2022; Masana et al., 2023). Data replay techniques (Lopez-Paz & Ranzato, 2017; Rebuffi et al., 2017; Chaudhry et al., 2019b) store a small portion of previous data in a buffer to utilize for training and are widely used with other techniques (Zhou et al., 2023a) including knowledge distillation (Rebuffi et al., 2017; Buzzega et al., 2020), model rectification (Wu et al., 2019; Zhao et al., 2020), and dynamic networks (Yan et al., 2021; Wang et al., 2022; Zhou et al., 2023b). Simple buffer sample selection methods such as random or herding-based approaches (Rebuffi et al., 2017) are also commonly used as well. There are also more advanced gradient-based sample selection techniques like GSS (Aljundi et al., 2019) and OCS (Yoon et al., 2022) that manage buffer data to have samples with diverse and representative gradient vectors. All these works do not consider fairness and simply assume that the entire incoming data is used for model training, which may result in unfair forgetting, as we show in our experiments.

Model fairness research mitigates bias by ensuring that a model's performance is equitable across different sensitive groups, thereby preventing discrimination based on race, gender, age, or other sensitive attributes (Mehrabi et al., 2022). Existing model fairness techniques can be categorized as pre-processing (Kamiran & Calders, 2011; Feldman et al., 2015; Calmon et al., 2017; Jiang & Nachum, 2020), in-processing (Agarwal et al., 2018; Zhang et al., 2018; Cotter et al., 2019; Roh et al., 2020), and post-processing (Hardt et al., 2016; Pleiss et al., 2017; Chzhen et al., 2019). In addition, there are other techniques that assign adaptive weights for samples to improve fairness (Chai & Wang, 2022; Jung et al., 2023). However, most of these techniques assume that the training data is given all at once, which may not be realistic. There are techniques for fairness-aware active learning (Anahideh et al., 2022; Pang et al., 2024; Tae et al., 2024), in which the training data evolves with the acquisition of samples. However, these techniques store all labeled data and use them for training, which is impractical in continual learning settings.

## D. Appendix – Future Work

Continuing from Sec. 5, we discuss future work.

### D.1. Generalization to Multiple sensitive attributes

FSW can be extended to tasks involving multiple sensitive attributes by defining a sensitive group as a combination of sensitive attributes. For instance, recall the loss for EO in a single sensitive attribute is $\frac{1}{|\mathbb{Y}||\mathbb{Z}|} \sum_{y\in\mathbb{Y}, z\in\mathbb{Z}} |\tilde{\ell}(f_\theta, G_{y,z}) - \tilde{\ell}(f_\theta, G_y)|$. This definition can be extended to the case of multiple sensitive attributes as $\frac{1}{|\mathbb{Y}||\mathbb{Z}_1||\mathbb{Z}_2|} \sum_{y\in\mathbb{Y}, z_1\in\mathbb{Z}_1, z_2\in\mathbb{Z}_2} |\tilde{\ell}(f_\theta, G_{y,z_1,z_2}) - \tilde{\ell}(f_\theta, G_y)|$. The new definition for multiple sensitive attributes allows the overall optimization problem to optimize both sensitive attributes simultaneously. The design above can also help prevent 'fairness gerrymandering' (Kearns et al., 2018), a situation where fairness is superficially achieved across multiple groups, but specific individuals or subgroups within those groups are systematically disadvantaged. This is achieved by minimizing all combinations of subgroups, thereby disrupting the potential for unfair prediction based on certain attribute combinations. However, having multiple loss functions may increase the complexity of optimization, and a more advanced loss function may need to be designed for multiple sensitive attributes. We leave the extension of this work to multiple sensitive attributes in future work.

