# OpenReview forum: "Fair Class-Incremental Learning using Sample Weighting"
_ICML.cc/2025/Conference — Submitted to ICML 2025_

### Official Review · Reviewer_1WVp · 2025-03-09

**Overall Recommendation:** 3

**Summary:**

This paper addresses the issue of ensuring fairness in class incremental learning. The authors identify a problem of unfair catastrophic forgetting, where previously learned information about certain sensitive groups (including classes) are disproportionately forgotten when new data is introduced. This work provides a novel approach for class incremental learning, ensuring that sensitive groups are not disproportionately forgotten while maintaining accuracy. The proposed FSW algorithm offers a practical solution for integrating fairness into continual learning settings.


Key Contributions:
- This paper demonstrates that unfair forgetting occurs when the average gradient vector of the current task data is in the opposite direction to the gradient of an underperforming group, resulting in a greater loss for certain groups, leading to biased learning.
- This paper introduces FSW algorithm which adjusts the training weights of current task samples to modify the gradient direction, thereby mitigating unfair forgetting.
- The proposed method supports various fairness measures, including EER, EO and DP.

Key Findings:
- The paper shows that training with equal weights for all current task data can lead to unfair forgetting.
- By adjusting the weights of current task samples, FSW reduces unfair forgetting and improves fairness without significantly sacrificing accuracy.


## Update after rebuttal
I appreciate the authors’ rebuttal. However, my concern regarding the relatively minor role of fairness in the overall contribution is not fully resolved. I therefore maintain my current recommendation.

**Claims And Evidence:**

- Unfair forgetting occurs when the gradient of the current task data is in the opposite direction of a sensitive group’s gradient.

  - The authors provide a theoretical analysis (Lemma 3.1 and Theorem 3.3) showing that when the inner product between these gradients is negative, the loss for the sensitive group increases. However, the analysis primarily focuses on sensitive classes, which is less common in fairness research compared to sensitive attributes (e.g., gender, race). Extending the analysis to sensitive attributes, a more prominent area in fairness, could have strengthened the paper's relevance to broader fairness applications.


- FSW reduces unfair forgetting by reweighting sample gradients.

    - The method is mathematically sound. Experimental results across multiple datasets show that FSW achieves competitive results compared to state-of-the-art baselines. However, the reported performance of FaIRL in Table 2 appears significantly worse than the results in its original paper. For instance, the accuracy for Biased MNIST (p=0.95) is 0.411 in this paper versus 0.7566 in the FaIRL paper. While the authors mentioned the instability of training the feature encoder and discriminator together in FaIRL, the large performance gap raises concerns about whether the baseline models represent their actual performance. In some cases, FaIRL performs worse than the naive "Fine Tuning" approach, which only trains on new data without access to previous task data, further questioning the validity of the baseline comparisons.


- Extensibility of FSW

    - In section 4.5, the authors combine FSW with a post-processing method, $\epsilon$-fair. However, the results in Table 4 do not strongly support that FSW outperforms baseline methods. At best, the results show a trade-off between accuracy and fairness, but they do not clearly demonstrate that FSW, $\epsilon$-fair is superior. In fact, the results suggest that iCaRL, despite not being designed with fairness in mind, might still be a competitive algorithm when combined with $\epsilon$-fair. This raises questions about whether FSW's fairness-aware design provides a significant advantage over fairness-agnostic methods when post-processing is applied.

**Essential References Not Discussed:**

The related works are essential to understanding the (context for) key contributions of the paper.

**Experimental Designs Or Analyses:**

The experimental design and analyses in the paper appear methodologically sound, but several considerations and potential limitations should be noted:

- The paper highlights extending FSW to multiple sensitive attributes as a future research direction. However, the current approach optimizes fairness for only a single sensitive attribute, which may not fully address real-world biases.

- The evaluation focuses solely on the fairness measure used in FSW. Including additional fairness metrics would provide a more comprehensive assessment and better illustrate FSW’s advantages over fairness-agnostic baselines.

- A comparison of computational costs across methods would be beneficial. Figure 8 in Appendix B.3 indicates that FSW significantly increases runtime, with training accounting for only about 10% of the total runtime. This added computational complexity may limit its practical applications. While the authors emphasize an offline learning setting rather than online learning, the high computational cost could diminish FSW’s benefits. For instance, if FSW requires excessive time, it may be more practical to simply retrain a model using all data (as done in the Joint Training baseline), particularly in an offline setting.

**Methods And Evaluation Criteria:**

The proposed methods and evaluation criteria generally make sense for the problem of unfair class forgetting in class incremental learning. However, the results would be more balanced and convincing if additional fairness measures were reported in the results tables, rather than only the measure optimized by FSW. Since most baseline models (except FaIRL) do not consider fairness, FSW is naturally expected to perform better in terms of fairness. Including multiple fairness metrics would provide a more comprehensive comparison and highlight whether FSW improves fairness across different criteria or only the one it optimizes.

**Other Comments Or Suggestions:**

The authors "only consider the gradient of the model's last layer." However, the referenced papers primarily discuss gradient norms and only state that "the variation of the gradient norm is mostly captured by the gradient of the loss function with respect to the pre-activation outputs of the last layer," which does not necessarily justify last layer gradient approximation. It would be helpful for the authors to clarify this choice and discuss any potential limitations it may introduce.

Since there are relative few methods specifically designed for fair class-incremental learning, it would be interesting to evaluate whether the proposed sample weighting approach improves catastrophic forgetting without considering fairness. For instance, one could modify the optimization objective to minimize the following:

This adjustment would allow for a direct comparison with other fairness-agnostic incremental learning methods, helping to isolate the impact of the sample weighting strategy on mitigating catastrophic forgetting.

**Other Strengths And Weaknesses:**

Strengths:

- The paper provides a strong theoretical analysis and conducts comprehensive experiments.

- It establishes a solid theoretical foundation for catastrophic forgetting, using gradient vector analysis to illustrate how sample weighting can mitigate it. This enhances the motivation behind the proposed FSW method for class-incremental learning.

- The proposed method is flexible and can be integrated with various fairness measures.

Weaknesses:

- While the authors incorporate fairness measures to find the optimal weights, the core contribution remains a weighting strategy to mitigate catastrophic forgetting, making the fairness aspect relatively minor.

- The performance of the proposed FSW is not entirely convincing. Table 12 shows that when post-processing is applied (iCaRL - $\epsilon$-fair), FSW performs worse in both accuracy and fairness on the Biased MNIST and DRUG datasets. On the BiasBios dataset, iCaRL achieves the same level of fairness while maintaining higher accuracy, despite not explicitly considering fairness, raising questions about FSW’s effectiveness in certain scenarios.

**Questions For Authors:**

1. In Section 3.3, the authors state that experiments indicate selecting representative and diverse samples for the buffer leads to better accuracy and fairness. Based on this, they choose a simple random sampling approach. However, this appears contradictory to the core idea of the paper — that samples should be carefully weighted, as equal weighting leads to unfairness and catastrophic forgetting. In the current approach, the samples that FSW assigns lower weights have an equal chance to be reintroduced into the buffer and later used to train the model on new classes. It would be valuable for the authors to provide some of the "experimental observations" that support the choice of random sampling for the buffer. Additionally, conducting experiments to assess the impact of different sampling strategies, such as using sample weights to determine selection probabilities, could further strengthen their findings.

2. In Appendix B.12, the authors demonstrate that most sample weights are binary, with only few falling between 0 and 1. This naturally raises the question: what would happen if samples with weights close to 0 were entirely excluded from training? How would this impact both performance and fairness? Given that nearly all weights are binary, the use of random sampling for the buffer seems even more counterintuitive.

**Relation To Broader Scientific Literature:**

This paper integrates fairness into class incremental learning, addressing the issue of unfair catastrophic forgetting, which has been neglected in existing continual learning research.

**Theoretical Claims:**

The authors provide a theoretical proof connecting cross-entropy loss to EO. For other fairness measures, such as EER and DP, the authors rely on prior work to justify the use of cross-entropy loss as a proxy. While this reliance on established literature is reasonable, a more detailed theoretical or empirical justification for EER and DP would strengthen the claim and provide a more comprehensive understanding of how cross-entropy loss relates to these fairness metrics.

---

> ### Author Rebuttal · Authors · 2025-04-01
>
> Thanks for your thoughtful review.
> To save space, we use the following shorthand terms:
> * C&E: Claims And Evidence
> * E&A: Experimental Designs Or Analyses
> * M&E: Methods And Evaluation Criteria
> * OCS: Other Comments Or Suggestions
>
> >C&E1: Scope limited to class
>
> Please refer to [W1] of Reviewer qrn5.
> > C&E2: FaIRL performance
>
> We clarify that FaIRL's results appear worse than its original results because we unified the backbone across all baselines, which is different from the original model.
> In general, FaIRL's complex model structures not only lead to training instability, but also involve multiple loss terms with many hyperparameters. Moreover, FaIRL has the longest training time ([E&A3] of our response) among all baselines, and thus difficult to tune.
> >C&E3, W2: FSW with a fair post-processing
>
> We emphasize that FSW−𝜖-fair achieves the lowest fairness disparity among all baselines. We also clarify that directly comparing FSW with iCaRL−𝜖-fair is unfair, as 𝜖-fair is a post-hoc method applied after training. A more appropriate comparison is FSW−𝜖-fair versus iCaRL−𝜖-fair, where FSW−𝜖-fair shows superior DP disparity across all datasets.
> iCaRL's high accuracy on BiasBios likely stems from its prototypical classifier, structurally distinct from the fully-connected layer used by other methods. Notably, iCaRL’s accuracy surpasses even ‘Joint Training’, commonly considered the upper bound. (The prototypical classifier is not always superior, as we discuss in [E&A2] of Reviewer qrn5.) On the other hand, FSW achieves fairness and accuracy comparable to Joint Training on BiasBios, effectively mitigating unfair forgetting without compromising accuracy.
> >M&E, E&A2: Fairness measurement
>
> For the results of more fairness measures, please refer to [M&E1, E&A1] of Reviewer qrn5.
> We respectfully disagree that FSW should outperform fairness-agnostic baselines across additional non-targeted fairness metrics. Each fairness measure has its own purpose and applicability (see [M&E2] of Reviewer qrn5), and prior work [1] shows that satisfying multiple notions of fairness simultaneously is impossible.
>
> [1] Kleinberg et al., ‘Algorithmic Fairness’, AEA 2018
> >Theoretical Claims: Cross-entropy loss for EER and DP
>
> We extend the theoretical argument presented for EO (Sec. A.2) to EER and DP.
> * EER: $\ell(G_y)$ estimates '$-\log p(\hat{\text{y}}|\text{y}=y)$' and $\ell(G_{\mathbb{Y}})$ estimates '$-\log p(\hat{\text{y}})$'
> * DP: $\ell'(G_{y, z})$ estimates '$-\log p(\hat{\text{y}}|\text{z}=z)$' and $\ell'(G_y)$ estimates '$-\log p(\hat{\text{y}})$'
>
> >E&A1: Multiple sensitive attributes
>
> We agree that extending to multiple sensitive attributes is crucial for real-world scenarios, but it adds significant complexity in loss design and optimization. In the Future Work section, we propose an initial strategy to jointly minimize fairness disparities across attributes, thus addressing concerns like fairness gerrymandering.
> >E&A3: Runtime
>
> We compare the total runtime of all baselines and show that the overhead of FSW is not excessive. We do not compare with Joint Training as it accesses all the previous data, which is impractical in class-incremental learning.
> |Biased MNIST|Runtime|
> |-|-|
> |FSW (EO)|825s|
> |iCaRL|947s|
> |WA|193s|
> |CLAD|169s|
> |GSS|5632s|
> |OCS|2681s|
> |FaIRL|5988s|
>
> We also discuss how to reduce FSW's overhead in [W1, Q1] of Reviewer CGdC.
> >W1: Relationship between weight adjustment and fairness
>
> Please refer to [W1] of Reviewer qrn5.
> >OCS1: Gradient of all layers
>
> We use only last-layer gradients to reduce computational overhead, and it is empirically reasonable and supported by prior works mentioned in our paper. Nonetheless, we acknowledge that some information may be lost.
>
> > OCS2: FSW without considering fairness
>
> We compare FSW with and without fairness, and show that FSW with fairness mitigates unfair catastrophic forgetting. As a result, the two methods have comparable accuracy, while FSW has a larger fairness disparity.
>
> |Biased MNIST (wrt DP)|Acc.|DP Disp.|
> |-|-|-|
> |FSW| .9040 |**.0083**|
> |FSW (w/o fairness)|**.9046**|.0084|
>
> (The results for EO Disp. are similar)
> >Q1:  Buffer strategy
>
> We clarify that buffering preserves representative samples of previously learned groups, while sample weighting specifically targets mitigating unfair catastrophic forgetting during training.
> Although selecting high-weight samples for buffering may provide better current-task fairness, it can distort the original distribution of groups within the buffer. Modified distributions can lead not only to misclassification but also to difficulty in accurately determining fairness weights for future tasks, as the buffer works as a reference for both.
> >Q2: Sample with zero weight
>
> We clarify that the samples with zero weights are excluded from training.

---

### Official Review · Reviewer_M6HL · 2025-03-11

**Overall Recommendation:** 3

**Summary:**

This paper investigates fairness in class-incremental learning (CIL). The authors argue that when learning new knowledge negatively impacts old tasks whose gradients oppose the new one while minimally affecting other tasks, unfairness arises. To address this, the authors introduce three fairness measures: equal error rate (EER), equalized odds (EO), and demographic parity (DP). Based on these measures, they propose the fairness-aware sample weighting (FSW) approach. Experiments demonstrate that FSW effectively reduces unfairness in CIL scenarios.

### update after rebuttal
Thank you for the authors' careful response. All my concerns have been resolved, and I've increased my recommendation to weak accept.

**Claims And Evidence:**

The paper proposes three fairness measures: EER, EO, and DP. However, the authors did not conduct ablation studies to separately validate each measure’s individual effectiveness. I would like to see experimental results illustrating how removing each of these measures individually from the full FSW method impacts performance.

**Essential References Not Discussed:**

None

**Experimental Designs Or Analyses:**

One limitation in the experimental design is that the compared methods are not state-of-the-art methods known to perform best on MNIST. Therefore, the experiments do not sufficiently demonstrate the superiority of FSW.

**Methods And Evaluation Criteria:**

The proposed method contributes to reducing unfairness in class-incremental learning scenarios, and the evaluation datasets chosen are standard benchmarks widely used in this field.

**Other Comments Or Suggestions:**

None

**Other Strengths And Weaknesses:**

The strengths of this paper include detailed writing and explicit mathematical derivations. However, a key weakness is the insufficient ablation studies.

**Questions For Authors:**

1. According to the example in Figure 1(c), although opposition between $g_2^*$ and $g_1$ is reduced, the resulting gradient becomes nearly opposite to $g_0$. When the number of tasks grows, is it possible that reducing opposition to one task inadvertently increases opposition to others?
2. Both EO and DP introduce constraints based on sensitive groups within classes (the dimension $\mathbb{Z}$). However, since our task focuses solely on class-incremental, is this additional constraint on the $\mathbb{Z}$ dimension meaningful?
3. Why does the Error Rate metric only consider a single sensitive attribute—the class ($\mathbb{Y}$ dimension)—while the Positive Rate metrics (EO and DP) consider two dimensions ($\mathbb{Y}$ and $\mathbb{Z}$)?

**Relation To Broader Scientific Literature:**

This paper primarily relates to continual learning, particularly CIL. It might also apply to domain-incremental scenarios. Methodologically, this study emphasizes fairness and relates to certain approaches in continual semantic segmentation, such as “Fairness continual learning approach to semantic scene understanding in open-world environments (NeurIPS 2023).”

**Theoretical Claims:**

The theoretical claims made in this paper are supported by mathematical derivations.

---

> ### Author Rebuttal · Authors · 2025-04-01
>
> Thank you for your thoughtful review and constructive feedback.
> To save space, we use the following shorthand terms:
> * C&E: Claims And Evidence
> * E&A: Experimental Designs Or Analyses
> * RL: Relation To Broader Scientific Literature
> * M&E: Methods And Evaluation Criteria
>
> >C&E: Each measure's effectiveness
>
> We clarify that FSW is not designed to simultaneously optimize multiple fairness measures, but only **optimizes one provided by the user**. Hence, an ablation study for each measure's individual effectiveness is not applicable in our setup.
> Rather, we validate the effectiveness of FSW by independently optimizing for each metric and reporting performance accordingly.
> >E&A: Baselines
>
> We acknowledge that some state-of-the-art methods on MNIST may achieve higher performances. However, our choice of baselines was driven by practical considerations of computational cost and spatial complexity. For instance, while L2P [1] achieves strong performance, it leverages a pretrained vision transformer that already embeds extensive prior information, making it less directly comparable to FSW. Similarly, DER [2] trains a separate feature extractor for each task, which incurs additional storage overhead and complexity. In contrast, the baselines like iCaRL and WA are widely recognized as strong and representative baselines in many studies [3-5], offering a fair and practical comparison. Within these constraints, we believe our experiments sufficiently demonstrate the superiority of FSW. Moreover, like iCaRL and WA, other state-of-the-art methods that optimize solely for accuracy are likely to show a worse accuracy-fairness tradeoff than FSW.
> [1] Wang et al., ‘Learning to prompt for continual learning’, CVPR 2022
>
> [2] Yan et al., ‘Dynamically expandable representation for class incremental learning’, CVPR 2021
>
> [3] Zhou et al., ‘Class-Incremental Learning: A Survey’, IEEE TPAMI 2024
>
> [4] Jiangpeng He., ‘Gradient Reweighting: Towards Imbalanced Class-Incremental Learning’, CVPR 2024
>
> [5] Zhuang et al., ‘Class incremental learning with analytic learning for hyperspectral image classification’, Journal of the Franklin Institute 2024
> > RL: Extension on domain-incremental scenarios
>
> We clarify that FairCL [6] is already discussed in the related work of our paper. FairCL specifically targets the imbalanced class distribution based on the number of pixel counts of each class. In comparison, we support fairness more generally in class-incremental learning by satisfying multiple notions of group fairness for sensitive groups, including classes.
> While class-incremental learning has the highest difficulty among continual learning scenarios (task, domain, and class-incremental learning), FSW can indeed extend to domain-incremental scenarios.
>
> [6] Truong et al., ‘Fairness continual learning approach to semantic scene understanding in open-world environments’, NeurIPS 2023
> >Q1: Assurance of loss decrease
>
> You are correct that in some cases, reducing the gradient opposition for one group may increase it for another. This is a natural trade-off when balancing multiple group objectives in a shared parameter space. However, FSW is designed to **minimize the average classification loss across all sensitive groups.** While the loss of some groups may increase due to the interactions, the overall trend is a reduction of unfair forgetting by minimizing disparities across groups.
> Hence, our goal is not to perfectly preserve all the previously learned groups, since catastrophic forgetting is inevitable, but to prevent unfair forgetting while minimally reducing the accuracy of other groups.
> >Q2, 3: Necessity of both Y and Z for a fair class-incremental learning scenario
>
> Ensuring fairness in class-incremental learning is practically important, as unfair model predictions can undermine the trust and safety in human-related automated systems that use continual learning. Several metrics can be utilized to define fairness. First, EER is used as a target metric where a sensitive attribute (Z) is not explicitly provided. This scenario includes many commonly used class-incremental learning benchmarks where only the class label (Y) is available. In such cases, EER serves as a practical proxy for fairness under task imbalance, focusing on equal error across classes.
> On the other hand, when Z is available, fairness can also be evaluated using group-based metrics like EO and DP, which use both Y and Z to cover broader social norms.
> For the usage of each metric (EER, EO, and DP), please refer to the [M&E2] of reviewer qrn5.
>
> ⠀
>
> We will include the discussion and experimental results in our revised paper. We hope that our responses address your concerns. If we have successfully addressed your questions, we would really appreciate it if you consider raising your score. If you have any further concerns, we are happy to discuss and address them.

---

> > ### Comment · Reviewer_M6HL · 2025-04-02
> >
> > Thank you for your careful response. All my concerns have been resolved, and I've increased my recommendation to weak accept.

---

> > > ### Author Response · Authors · 2025-04-05
> > >
> > > We appreciate your participation in the discussion process and are happy that our response has addressed your concerns. We thank you for the valuable questions and feedback and will incorporate them into our paper.
> > >
> > > Warm regards,
> > >
> > > Authors

---

### Official Review · Reviewer_CGdC · 2025-03-11

**Overall Recommendation:** 3

**Summary:**

This submission proposes a fairness-aware sample weighting (FSW) algorithm to address unfair catastrophic forgetting in class-incremental learning. A key contribution of this work is the detailed analysis of the forgetting mechanism, which demonstrates that catastrophic forgetting occurs when the gradient vectors of current task data and sensitive groups have opposite directions, resulting in negative inner products. This theoretical insight provides a clear understanding of when unfair forgetting happens and serves as the foundation for the proposed FSW algorithm. By adjusting the sample weights to align the gradient vectors, the method effectively reduces the forgetting of underperforming groups while maintaining overall accuracy. The work is well-motivated, addressing a critical yet understudied problem in trustworthy AI.

**Claims And Evidence:**

The claims made in the submission are well-supported by clear and convincing evidence, as follows:
* The theoretical analysis of gradient vectors and the demonstration of negative inner products support the claim that certain directions of gradient updates can lead to unfair forgetting (Figure 1).
* The experimental results across various datasets (Table 2) show that FSW achieves better accuracy-fairness tradeoff compared to state-of-the-art methods, supporting the claim that FSW effectively mitigates unfair forgetting.
* The analysis (Figure 2) visually demonstrates that FSW achieves a superior balance between accuracy and fairness on both MNIST and Biased MNIST datasets. This supports the claim that FSW provides an optimal tradeoff in class-incremental learning scenarios.

**Essential References Not Discussed:**

The literature discussed by the authors appears to be sufficiently comprehensive and closely related to the topic. The cited works cover the necessary background and prior findings, providing a thorough context for their contributions.

**Experimental Designs Or Analyses:**

Yes, I have checked them. The soundness and validity of the experimental designs and analyses in this submission are well-constructed, such as the comprehensive evaluation on multiple datasets (Table 1, Figure 2), fairness and accuracy tradeoff analysis (Figure 2), and sample weighting analysis (Figure 3). It is commendable that the manuscript provides ample illustrative observations, such as Figures 2, 3, and 4, which add to its value.

**Methods And Evaluation Criteria:**

Yes, the manuscript employs reasonable methods and evaluation criteria. The authors use a variety of common datasets in class-incremental learning scenarios (MNIST, Fashion-MNIST, Biased MNIST, Drug Consumption, and BiasBios) and well-recognized metrics (accuracy, fairness measures such as Equal Error Rate (EER), Equalized Odds (EO), and Demographic Parity (DP) disparity). This provides a comprehensive assessment of the proposed method's performance in terms of both accuracy and fairness.

**Other Comments Or Suggestions:**

I find this work quite valuable. It fills a gap in the research of class - incremental learning by considering fairness. The proposed FSW algorithm provides a practical solution for real - world applications where fairness is crucial.
Overall, I am positive about this work. Given the identified weaknesses, I hope the authors can address them in future research or in a potential rebuttal.

**Other Strengths And Weaknesses:**

**Strengths,**
* The paper presents an interesting and practical problem of fair class-incremental learning. The proposed fairness - aware sample weighting (FSW) algorithm is a novel approach. It offers a new way to address the issue of unfair catastrophic forgetting, which can potentially inspire further research in the field of Trustworthy AI.
* The authors conduct a comprehensive theoretical analysis. They use gradient vectors to analyze unfair forgetting and formulate optimization problems for different fairness measures. This provides a solid theoretical foundation for their proposed method.
* A variety of experiments are carried out on multiple datasets. The experiments cover different types of data such as images, text, and tabular data. The results demonstrate the effectiveness of FSW in achieving a better accuracy - fairness trade - off compared to state - of - the - art methods.
* The paper is well - structured and clearly written. The authors first introduce the problem, then present related work, followed by the framework, experiments, and conclusions. They also discuss the limitations of existing methods and their own work's contributions clearly.

**Weaknesses,**
* Although the FSW algorithm shows good performance, its computational complexity is quadratic to the number of current task samples. For large - scale datasets, this could lead to high computational costs. The authors mention clustering as a potential solution, but it is not fully explored in the paper.
* The experiments mainly focus on a limited number of fairness measures and datasets. There could be other fairness measures or datasets where the performance of FSW might be different. A more extensive exploration of different scenarios would strengthen the generality of the results.
* The fairness - aware optimization problems are transformed into linear programming problems. However, the quality of the results may depend on the choice of the linear optimization solver. The authors use CPLEX, but they do not discuss the impact of other solvers on the performance of FSW.
* The use of first - order Taylor series approximation in the theoretical analysis may not be sufficient for large networks. In large - scale neural networks, the loss function may be highly non - linear, and the assumption of local linearity in small optimization steps might not hold as well. This could lead to inaccurate approximations and affect the validity of the theoretical results related to unfair forgetting and the performance of the FSW algorithm.
* The FSW algorithm proposed in the article aims to address the unfairness issues in the class distribution of sensitive data, but it does not fully consider the impact of data drift. When data drift occurs, the strategy of the FSW algorithm to adjust weights based on current task samples may not be flexible enough. This is because the algorithm relies on the gradient relationships between current and past data. Under data drift, the gradient relationships are complex and changeable, making it difficult for the FSW algorithm to adapt. As a result, it cannot adjust the weights in a timely manner to balance fairness and accuracy. Therefore, the effectiveness and adaptability of the FSW algorithm in dealing with the class distribution of sensitive data in data drift scenarios need to be further studied.

**Questions For Authors:**

Here are a few questions that interest me,
* How would the FSW algorithm perform in scenarios with extremely large datasets considering its quadratic computational complexity? Are there any other more efficient alternatives to clustering for reducing the computational overhead?
* Have the authors considered applying FSW to other types of fairness measures not explored in this paper? What challenges might they face in such applications?
* How sensitive is the performance of FSW to the choice of the linear optimization solver? Have the authors tested other solvers, and if so, what were the results?
* In the context of large - scale neural networks, how do you justify the use of first - order Taylor series approximation? Do you plan to explore higher - order approximations or other methods to improve the accuracy of the theoretical analysis?
* Since data drift can disrupt the gradient relationships that the FSW algorithm relies on, causing difficulties in balancing fairness and accuracy, what steps will you take to ensure that the algorithm can still effectively address the unfairness issues in sensitive data class distribution during data drift?

#after rebuttal# I will maintain my current raiting.

**Relation To Broader Scientific Literature:**

The submission provides a thorough discussion of the relevant literature. Prior to this study, most efforts in class-incremental learning focused on using gradient information to mitigate catastrophic forgetting. However, this work explores a new technical pathway by addressing fairness in class-incremental learning through sample weighting. The key contribution is the development of a fairness-aware sample weighting (FSW) algorithm, which is related to the broader concepts of fairness, optimization, and catastrophic forgetting in continual learning (CL). The authors discuss literature related to these concepts, including CL, fairness in machine learning, gradient-based methods, and optimization techniques.

**Theoretical Claims:**

Yes, I have reviewed them. The manuscript includes reasonable theoretical claims, such as the analysis of gradient vectors and their impact on catastrophic forgetting (Lemma 3.1, Theorem 3.3), and the formulation of fairness-aware sample weighting through linear programming (Section 3.2). Moreover, the authors provide empirical evidence to support their theoretical claims (Figure 2).

---

> ### Author Rebuttal · Authors · 2025-04-01
>
> Thank you for your thoughtful review and constructive feedback.
> >W1, Q1: Quadratic computational complexity
>
> Clustering current data and assigning cluster-wise weight reduce LP variables, lowering computational complexity. Mini-batch K-means[1] is efficient for extremely large datasets where the complexity is O(BKDI) (B: batch size, K: number of clusters, D: data dimension, I: number of iterations).
> Another approach is reducing the complexity of LP using a first-order method, which results in a complexity of O(ND) (N: number of current task samples, D: data dimension). CPLEX solves LP based on the simplex method [2], which is accurate but computationally intensive due to Hessian calculations. First-order methods scale better by relying solely on gradient information [3]. These two approaches—clustering and first-order optimization—are orthogonal and can be combined to further improve FSW's scalability on large datasets.
>
> [1] Sculley, ‘Web-scale k-means clustering’, WWW 2010
>
> [2] Bixby, ‘Solving real-world linear programs: A decade and more of progress’, Operations research 2002
>
> [3] Beck, ‘First-order methods in optimization’, SIAM 2017
> >W2, Q2: More fairness measures or datasets
>
> FSW is not limited to EER, EO, and DP, but **can extend to other group fairness metrics**, including PP (Predictive Parity) [4, 5]. As a demonstration, PP ensures equal precision across sensitive groups, i.e., $\Pr({\text{y}}=\hat{y}|\hat{\text{y}}=\hat{y},\text{z}=z_1)=\Pr({\text{y}}=\hat{y}|\hat{\text{y}}=\hat{y},\text{z}=z_2)$ for $\hat{y}\in\mathbb{Y}$ and $z_{1,2}\in\mathbb{Z}$. We define the cost function as $L_{PP}=\frac{1}{|\mathbb{Y}||\mathbb{Z}|}\sum_{\hat{y}\in\mathbb{Y},z\in\mathbb{Z}}|\tilde{\ell}(f_𝜃,G_{\hat{y},z})-\tilde{\ell}(f_𝜃,G_{\hat{y}})|$. This allows FSW to optimize both accuracy and PP, as with other metrics.
> Additional experiments on Biased MNIST show that FSW improves PP while maintaining comparable accuracy. Moreover, since group fairness emphasizes aligning predictions across demographic distributions [5], FSW's group-specific weighting approach is broadly applicable.
> |Biased MNIST (wrt PP)|Acc.|PP Disp.|
> |-|-|-|
> |W/o FSW|**.9101**|.1078|
> |FSW |.9042|**.1032**|
>
> We also note that it is not easy to find suitable datasets for class-incremental learning involving multiclass classification with sensitive attributes. Many fairness datasets are designed for binary classification (e.g., Adult, COMPAS) or have too few classes for class-incremental learning (e.g., MEPS, Communities and Crime).
>
> [4] Dieterich et al., ‘COMPAS risk scales: Demonstrating accuracy equity and predictive parity’, Northpointe Inc. 2016
>
> [5] Verma et al., ‘Fairness definitions explained’, FairWare 2018
> >W3, Q3: Optimization solver
>
> We conduct additional experiments comparing CPLEX with another LP solver, HiGHS [6], and observe comparable accuracy-fairness results, indicating FSW’s robustness to the solver.
> While we can also use other solvers, we use CPLEX as it offers **strong solution stability and high speed** in solving large-scale problems with numerous variables and constraints. CPLEX also provides comprehensive tools for in-depth solution analyses and is thus widely used [7-9].
> |Biased MNIST (wrt EO)| Acc.|EO Disp.|
> |-|-|-|
> |FSW (using CPLEX)|**.9095**|.1187|
> |FSW (using HiGHS)|.9074|**.1172**|
>
> |Biased MNIST (wrt DP)| Acc.|DP Disp.|
> |-|-|-|
> |FSW (using CPLEX)|**.9040**|**.0083**|
> |FSW (using HiGHS)|.9024|.0091|
>
> [6] Huangfu et al., ‘Parallelizing the dual revised simplex method’, Mathematical Programming Computation 2018
>
> [7] Sridhar et al., ‘An approximate, efficient LP solver for LP rounding’, NIPS 2013
>
> [8] Gearhart et al., ‘Comparison of open-source linear programming solvers’, 2013
>
> [9] Zhang et al., ‘iflipper: Label flipping for individual fairness’, SIGMOD 2023
> > Q4: Higher-order approximation for Taylor series
>
> We can extend the approximated loss to a higher-order Taylor approximation to improve theoretical accuracy, but it makes the optimization **non-convex and NP-hard**. We omit the proof due to the character limit, but will show it in our final response. Nevertheless, exploring alternatives like convex relaxations or tractable approximations of higher-order terms is a promising direction to enhance theoretical fidelity.
> > W4, Q5: Effect of data drift
>
> Since FSW relies on gradients derived from both the data and the model, it can still compute adaptive weights if the data drift is not severe. However, if the data drift is severe enough to shift the decision boundary, FSW can be combined with existing robust training methods to ensure both fairness and robustness.
>
> ⠀
>
> We thank you for your constructive feedback, and please let us know if any of your concerns are not fully addressed. We are always happy to be engaged with you for further discussions.

---

> > ### Comment · Reviewer_CGdC · 2025-04-05
> >
> > Thank you for the response. I shall maintain the current raiting.

---

> > > ### Author Response · Authors · 2025-04-05
> > >
> > > As promised in our initial response, we provide proof of the non-convexity and NP-hardness of employing a higher-order Taylor approximation.
> > >
> > > > Proof of Non-Convexity and NP-Hardness
> > >
> > > Using a second-order Taylor expansion around the current model parameters $\theta$, the approximated loss becomes:
> > >
> > > $\tilde{\ell}(G) = \ell(G) - \nabla_\theta \ell(G)^\top \mathbf{x} + \frac{1}{2} \mathbf{x}^\top H_\theta(G) \mathbf{x}$
> > >
> > > where $H_\theta(G)$ is the Hessian (assumed to be PSD; we also prove a case when it is not), and we define $\mathbf{x}:= \frac{\eta}{|T|} \sum_{d_i \in T} \mathbf{w}^i \nabla_\theta \ell(d_i)$.
> > > This function is convex in $\mathbf{x}$ and thus convex in $\mathbf{w}$.
> > >
> > > To encourage fairness, we penalize the difference between two group losses:
> > >
> > > $L_{\text{fair}}(\mathbf{w}) := \left| \tilde{\ell}(G_1) - \tilde{\ell}(G_2) \right| = \left| c + b^\top \mathbf{x} + \frac{1}{2} \mathbf{x}^\top Q \mathbf{x} \right|$
> > >
> > > where $c$ and $b$ are appropriate constants and $Q = H_\theta(G_1) - H_\theta(G_2)$.
> > >
> > > Introducing a slack variable $t$, we rewrite the fairness loss constraint as:
> > >
> > > $- t \leq c + b^\top \mathbf{x} + \frac{1}{2} \mathbf{x}^\top Q \mathbf{x} \leq t$
> > >
> > > This yields a Quadratically Constrained Quadratic Program (QCQP) in $\mathbf{w}$.
> > >
> > > The constraint involves a quadratic form with matrix $Q = H_\theta(G_1) - H_\theta(G_2)$. In general, $Q$ is indefinite unless $H_\theta(G_1) = H_\theta(G_2)$. Hence, at least one constraint is non-convex.
> > >
> > > It is known that a QCQP, even with a single non-PSD constraint matrix, is non-convex and NP-hard. [1]
> > >
> > > Although $\tilde{\ell}(G)$ is convex in $\mathbf{w}$, the fairness term introduces non-convexity. Thus, the overall optimization problem is non-convex and NP-hard.
> > >
> > > Furthermore, if $H_\theta(G)$ is not PSD, then $\tilde{\ell}(G)$ itself becomes non-convex in $\mathbf{w}$, which also leads to an NP-hard optimization problem.
> > >
> > >
> > > [1] Pardalos et al., ``Quadratic programming with one negative eigenvalue is NP-hard,’’ Journal of Global optimization, 1991.
> > >
> > > ⠀
> > >
> > > We appreciate your participation in the discussion process and are happy that our response has addressed your concerns. We thank you for the valuable questions and feedback and will incorporate them into our paper.
> > >
> > > Warm regards,
> > >
> > > Authors

---

### Official Review · Reviewer_qrn5 · 2025-03-17

**Overall Recommendation:** 3

**Summary:**

This paper addresses the fairness issue in class-incremental learning and proposes the Fairness-aware Sample Weighting (FSW) algorithm. The authors theoretically analyze the causes of unfair forgetting and optimize gradient direction by adjusting training sample weights to improve fairness. Furthermore, FSW formulates group fairness metrics as a linear programming problem, with experimental results demonstrating its superiority over existing methods across multiple datasets.\
##update after rebuttal \
Thank you for the authors' thoughtful responses. After reviewing their rebuttal and considering the insights from other reviewers, I find that most of my initial concerns have been sufficiently addressed, though a few issues still remain. That said, I believe these remaining points could be resolved in future iterations of the work. Given that other reviewers have also raised important observations worth considering, I will retain my original scores for now.

**Claims And Evidence:**

Most of the claims in this paper are well demonstrated. However, I still have some concerns: \
The paper assumes that data replay is the sole approach to mitigating forgetting, without considering more advanced continual learning methods such as parameter isolation and adaptive knowledge distillation. If more advanced continual learning techniques can more effectively retain previous knowledge, is it still necessary to specifically design FSW for sample weighting? In other words, the selection of baselines in the paper primarily focuses on certain methods in current incremental learning and fairness research, yet fails to compare against other potential strategies that could mitigate unfair forgetting, which may undermine the persuasiveness of the results.

**Essential References Not Discussed:**

N/A

**Experimental Designs Or Analyses:**

I have examined the experimental design and analysis of this paper. The experimental design is comprehensive, incorporating a real dataset and extensive experimental analysis. However, the experiment also has certain limitations. \
1.The primary objective of the paper is to demonstrate that the proposed method can balance accuracy and fairness in class-incremental learning. However, the experimental employs different metrics to evaluate accuracy and fairness, making the results less intuitive. It is recommended to use a unified and intuitive metric to more clearly highlight the effectiveness of the proposed method.\
2.In the accuracy and fairness results section, it is stated that the iCaRL generally achieves high accuracy while maintaining a low EER disparity. However, it also exhibits a high EO disparity. This conclusion lacks quantitative evidence and fails to clearly demonstrate the effectiveness of the proposed method compared to iCaRL.

**Methods And Evaluation Criteria:**

The authors theoretically analyze the causes of unfair forgetting and optimize gradient direction by adjusting training sample weights to improve fairness. However, there are some issues that should be further resolved: \
In the section "sample weighting for unfairness mitigation", the paper analyzes and discusses three fairness evaluation metrics. However, in the experiments section, only one of these metrics is selected for comparison. It is recommended that the paper include a discussion on the relationships and applicability of the three metrics, clearly defining the purpose of each metric. This would enhance the clarity and persuasiveness of the comparative results.

**Other Comments Or Suggestions:**

The paper involves a significant number of symbols and formula derivations. It is recommended to present the meanings of different symbols in a list, if possible, to enhance the readability of the paper.

**Other Strengths And Weaknesses:**

Strengths:\
    1.This paper proposes a sample weighting method to address the issue of unfairness in class-incremental tasks, which holds significant implications for research in the field of fair artificial intelligence. Additionally, it provides an intuitive demonstration of the causes of unfair forgetting.\
    2.The experiments in this paper are comprehensive and the analysis is detailed, providing substantial evidence for the effectiveness of the proposed approach. Moreover, different visualizations are used to intuitively present the experimental results.\
Weaknesses:\
    1.As an optimization method, how can the relationship between the weights optimized by FSW and fairness be explained? Currently, the paper lacks an intuitive explanation of the impact of weight adjustments, which may affect the interpretability and credibility of its practical application.\
    2.The paper proposes to mitigate unfair forgetting primarily through sample weighting, but it does not sufficiently discuss potential new issues that may arise from this method, such as the decline in the learning ability of new classes due to excessive adjustment. Should corresponding countermeasures be developed to address over-adjustment?

**Questions For Authors:**

Please refer to the concerns above.

**Relation To Broader Scientific Literature:**

This paper enhances fairness in class-incremental tasks through a sample weighting approach. While prior literature has addressed similar issues, this work introduces a novel solution paradigm that ensures fairness without compromising accuracy.

**Theoretical Claims:**

This paper presents four key theoretical proofs. I have reviewed the logic and validity of these proofs and offer the following suggestion: \
In the proof of Lemma A.1, the Taylor expansion is employed, but only a first-order approximation is used, neglecting higher-order terms. Would it be possible to provide additional mathematical analysis or empirical validation to demonstrate that ignoring higher-order terms does not significantly affect the conclusion?

---

> ### Author Rebuttal · Authors · 2025-04-01
>
> Thanks for your thoughtful review.
> To save space, we use the following shorthand terms:
> * C&E: Claims And Evidence
> * E&A: Experimental Designs Or Analyses
> * M&E: Methods And Evaluation Criteria
> * OCS: Other Comments Or Suggestions
>
> >C&E: More Baselines
>
> While methods like parameter isolation and adaptive knowledge distillation mitigate catastrophic forgetting in general, they do not explicitly address forgetting across sensitive groups, potentially causing unfair forgetting. Parameter isolation preserves task-specific parameters but does not distinguish sensitive groups. Similarly, adaptive knowledge distillation lacks mechanisms to identify and retain network components beneficial to sensitive groups. The core contribution of FSW is its capability to **separately address different groups through sample weighting.** Future work could be sensitive group-specialized parameter isolation or adaptive knowledge distillation. Additionally, we further justify in [E&A] of Reviewer M6HL why our experiments sufficiently demonstrate the superiority of FSW.
> >M&E1, E&A1: Clarity of metrics in the experiments
>
> We extend our Ablation Study to include accuracy and more fairness metrics results. When background color is used as the sensitive attribute (EO and DP), FSW optimized for one fairness metric also partially mitigates the disparity of other metrics, though its effectiveness is highest for the targeted metric. In contrast, FSW does not effectively reduce EER disparity, where the sensitive attribute is defined as the class label.
> |Biased MNIST|Acc.|EO Disp.|DP Disp.|EER Disp.|
> |-|-|-|-|-|
> |W/o FSW|**.9100**|.1260|.0086|.0317|
> |FSW (wrt EO)|.9095|**.1187**|.0084|.0322|
> |FSW (wrt DP)|.9040|.1229|**.0083**|**.0285**|
>
> Additionally, defining a unified metric for accuracy and fairness is challenging, as their relative importance varies by application. Instead, we visualize the accuracy-fairness tradeoff curves (Sec. B.9), where FSW consistently occupies the optimal bottom-right region, demonstrating Pareto-optimal performance.
> >M&E2: Applicability of the metrics
>
> EER targets fairness in predictive quality, aiming for equitable accuracy or error across different prediction outcomes, whereas EO and DP are standard group fairness metrics used when protected attributes exist. EO is preferred when ensuring equal model performance across groups or when true/false positives are equally important. DP is preferred for mitigating data bias or achieving ‘blind fairness’ by equalizing output distributions. Additionally, in [Q2, 3] of Reviewer M6HL, we explain why we chose EER, EO, and DP as our metrics.
> >Theoretical Claims: Taylor expansion error analysis
>
> To address your concern, we empirically evaluate the approximation error from using only the first-order Taylor expansion, **showing that the error diminishes and converges to zero with increasing epochs (Fig. 6).**
> Additionally, first-order approximations are widely used in continual learning, assuming local linearity where catastrophic forgetting mainly occurs in the first-order term [1, 2]. While higher-order terms can be included, doing so makes the optimization non-convex and NP-hard. A detailed proof is provided in [Q4] of reviewer CGdC.
>
> [1] Lopez-Paz et al., ‘Gradient Episodic Memory for Continual Learning', NeurIPS 2017
>
> [2] Aljundi et al., ‘Gradient based sample selection for online continual learning', NeurIPS 2019
> >E&A2: Behavior of iCaRL compared with FSW
>
> As detailed in Sec. B.9, FSW achieves a superior accuracy-fairness tradeoff compared to baselines, including iCaRL. iCaRL notably suffers from high fairness disparity, particularly when the sensitive attribute is not class-based, due to its reliance on well-separated feature embeddings. The t-SNE visualization (Fig. 5) on Biased MNIST shows misaligned samples due to background color biases, negatively affecting iCaRL’s accuracy. Quantitatively, Tables 2, 5, 6, and 7 show iCaRL’s consistently higher EO and DP disparities, confirming FSW’s robustness under less discriminative representations.
> >W1: Relationship between weight adjustment and fairness
>
> FSW’s core idea is to dynamically assign weights to mitigate unfair catastrophic forgetting, as illustrated in Fig. 1 of our paper. Forgetting is not uniform (like class 1 in Fig. 1), leading to fairness disparities. Fig. 1 may give the impression that FSW is limited to classes, but by considering a class to be a group, we can apply the same idea to mitigate unfair catastrophic forgetting for sensitive groups.
> >W2: Learning ability of new classes
>
> We clarify that our training objective uses the classification loss $L_{acc}$ (Eqs. (2)-(4)) for the current task data. To balance fairness and accuracy, FSW uses a hyperparameter λ on $L_{acc}$ to prevent excessive adjustments that hinder the learning of new classes.
> >OCS: Readability
>
> We will summarize the symbol definitions into a table in our revision.
>
> ⠀
>
> Please let us know if any of your concerns are not fully addressed.

---

### Decision · Program_Chairs · 2025-05-01

**Decision:**

Reject

**Comment:**

This paper focuses on achieving accuracy on sensitive classes in class-incremental learning setting. The authors theoretically show that a naive training framework will induce unfair catastrophic forgetting for different groups and class combinations, and identify a primary cause and propose a fix that learns a weighting of the training samples by solving an optimization problem that minimizes the overall losses of sensitive groups while minimizing disparities among them.

Reviewers liked the paper’s setting and appreciated the explanations about why the problem happens. Reviewers were enthusiastic but had many suggestions to improve the work and important questions about the trade-offs and computational complexity that were not able to be sufficiently addressed by the authors' responses, and the reviewers were unable to argue that the novelty or significance were strong enough for ICML acceptance.

We hope the authors will find the reviews constructive and help them make a stronger report on this promising research.